# ADVERSARIAL GRAPH NEURAL NETWORK BENCH-MARKS: TOWARDS PRACTICAL AND FAIR EVALUATION

## ABSTRACT

Adversarial learning and the robustness of Graph Neural Networks (GNNs) are topics of widespread interest in the machine learning community, as documented by the number of adversarial attacks and defenses designed for these purposes. While a rigorous evaluation of these adversarial methods is necessary to understand the robustness of GNNs in real-world applications, we posit that many works in the literature do not share the same experimental settings, leading to ambiguous and potentially contradictory scientific conclusions.

In this benchmark, we advocate for standardized, rigorous evaluation practices in adversarial GNN research. We perform a comprehensive re-evaluation of seven widely used attacks and eight recent defenses under both poisoning and evasion scenarios, across six popular graph datasets. Our study spans over 437,000 experiments conducted within a unified framework.

We observe substantial differences in adversarial attack performance when evaluated under a fair and robust procedure. Our findings reveal that previously overlooked factors, such as target node selection and the training process of the attacked model, have a profound impact on attack effectiveness, to the extent of completely distorting performance insights. These results underscore the urgent need for a standardized evaluation framework in adversarial graph machine learning.

## 1 INTRODUCTION

Applying machine learning to graph-structured data, such as financial transaction networks, social graphs, and molecular structures, requires models that can effectively embed non-Euclidean relationships. Graph Neural Networks (GNNs), introduced by Scarselli et al. (2009) and Micheli (2009), have become foundational tools for this purpose. Over the past decade, GNNs have achieved strong performance across domains, but their vulnerability to adversarial attacks has raised growing concerns. A series of recent works (Zügner et al., 2018; Li et al., 2023; Xu et al., 2019a; Geisler et al., 2021) demonstrate that even minor perturbations to the input graph can significantly degrade GNN performance.

As attack strategies have proliferated, inconsistencies in evaluation protocols have emerged as a serious obstacle to scientific progress. Many studies report substantial gains using differing experimental setups, making results difficult to compare and conclusions potentially misleading.
The reproducibility crisis in machine learning has highlighted the importance of standardized, rigorous empirical evaluations (Lipton & Steinhardt, 2019). In adversarial graph learning, Mujkanovic et al. (2022) warn that the graph community has yet to absorb the "bitter lesson" from the vision community, where overlooking adaptive attacks and evaluation rigor once led to a flood of unreliable results.

In this work, we identify several recurring issues in current evaluations. First, GNNs are often trained using attack-specific hyperparameters or fixed data splits, biasing results. Second, new attack models are frequently tested under more favorable conditions than their baselines. Third, evaluations commonly select target nodes in a way that underrepresents high-degree nodes, which are typically more resistant to attacks (see Figure 8). As a result, reported improvements may reflect favorable setups rather than true advances in method design.
To mitigate these issues, we propose a standardized, robust evaluation framework for adversarial attacks and defenses on GNNs. We re-evaluate several widely used gray and white-box attacks to expose how different experimental setups can lead to inconsistent or overstated findings. While a

comprehensive re-evaluation of all attacks is infeasible, our focused effort aims to establish stronger evaluation practices for the community.

To better contextualize performance claims, we also introduce *a naive yet surprisingly good baseline*, $L^1$D-RND, which achieves competitive results at minimal computational cost. Its success reinforces the need for basic sanity checks when proposing complex new methods.
By demonstrating the limited scalability of existing attacks and their declining effectiveness on high-degree nodes, our work highlights overlooked challenges in adversarial graph learning. We hope to encourage the development of more robust and scalable attack and defense strategies.

**Disclaimer.** This work advocates for rigorous evaluation practices. It is not intended to rank attacks or discredit prior contributions but to enable more reliable and reproducible comparisons across future studies.

## 2 RELATED WORK

**Adversarial Attacks.** Recent studies on adversarial attacks on graph data have developed optimal strategies to minimally perturb the graph (controlled by a budget parameter) while achieving the highest impact on a GNN's classification performance. Among the first methods is Nettack (Zügner et al., 2018), a gradient-based adversarial attack strategy that generates slight perturbations on graph structure and node features. Upon the success of Nettack, a variety of novel adversarial attack strategies have been proposed (Chen et al., 2018; Geisler et al., 2021). The majority of adversarial attacks proposed in the early stage only focus on small-scale datasets, typically consisting of less than 5000 nodes, which are typically impractical in real-world applications of GNNs. Only extracting a much smaller subgraph centered at the target nodes, Li et al. (2020a) proposed SGA as a scalable adversarial strategy. PR-BCD, another approach to adversarial attacks at scale by Geisler et al. (2021), adopts the Randomized Block Coordinate Descent (Nesterov, 2012) for solving large-scale optimization problems to find optimal perturbations. Meanwhile, in a recent study, GOttack (Alom et al., 2025) uses graph structures by targeting topological equivalence groups and exploiting their influence in gradient-based adversarial models.

**Evaluation procedures.** We follow the good practice of Errica et al. (2020) and Shchur et al. (2018). Particularly, both works standardize the evaluation procedures and promote a reproducible experimental environment with a rigorous *model selection* and *assessment framework*, but in two different contexts. Errica et al. (2020) focuses on graph classification tasks while Shchur et al. (2018)'s work is primarily on node classification. In addition, Shchur et al. (2018) have shown that the train/validation/test split of choice used in evaluation significantly impacts the performance ranking, thus drawing community attention to the necessity of using different splits in the evaluation procedure. Differentiating from them, which focus on designing rigorous evaluation frameworks for GNN models, we propose a robust evaluation procedure to prevent over-optimistic and biased estimates of the true performance of adversarial attack strategies.

The Graph Robustness Benchmark (GRB) (Zheng et al., 2021) was introduced a few years ago, and it mainly focuses on global evasion attacks. However, the GRB does not consider three valuable scenarios: (i) targeted attacks, (ii) poisoning scenarios, and (iii) the distinction between homophilic and heterophilic graphs. Our benchmark addresses these limitations by incorporating both targeted evasion and poisoning attacks, while explicitly evaluating performance on homophilic and heterophilic graphs, with victim models trained in each scenario.

## 3 PRELIMINARIES

Let $\mathcal{G} = (\mathcal{V}, \mathcal{E}, \mathbf{X})$ denote a graph, where $\mathcal{V}$ is the set of $N$ nodes, $\mathcal{E} \subseteq \{(v, w) \mid v, w \in \mathcal{V}\}$ is the set of directed edges, and $\mathbf{X} = \{\mathbf{x}_0, \mathbf{x}_1, \ldots, \mathbf{x}_{N-1}\}$ is the set of node feature vectors. Each $\mathbf{x}_i \in \mathbb{R}^M$ encodes the $M$-dimensional attributes of node $v_i$. The graph structure is represented by an adjacency matrix $\mathbf{A} \in \{0, 1\}^{N \times N}$, where $\mathbf{A}_{ij} = 1$ if $(v_i, v_j) \in \mathcal{E}$, and 0 otherwise. Each node $v_i$ has an associated label vector $\mathbf{y}_i \in \{0, 1\}^{|\mathcal{C}|}$ indicating its membership in one of $|\mathcal{C}|$ classes, forming the label matrix $\mathbf{Y} \in \{0, 1\}^{N \times |\mathcal{C}|}$.

**Semi-supervised Node Classification.** We focus on node classification in a semi-supervised setting, where labels are available only for a subset of nodes. Let $\mathcal{V}_L \subset \mathcal{V}$ denote the set of labeled nodes with known labels $\mathbf{Y}^L$, and $\mathcal{V}_U = \mathcal{V} \setminus \mathcal{V}_L$ the set of unlabeled nodes. The goal is to learn a function $g : \mathcal{G}, \mathbf{Y}^L \to \mathbf{Y}^U$ that predicts a class probability distribution for each node in $\mathcal{V}_U$. The predicted label $\hat{y}_v$ for a node $v \in \mathcal{V}_U$ corresponds to the class with the highest predicted probability in $g(v)$.

**Node Classification Margin.** For a node $v$ with ground truth label $y$, the classification margin $M_v$ measures the confidence of the model in the correct class. It is defined as the difference between the model's output score for the true class and the highest score assigned to any incorrect class (Zügner et al., 2018):

$$M_v = g(v)_y - \max_{c \in \mathcal{C},\, c \neq y} g(v)_c \tag{1}$$

A small or negative margin indicates that the prediction is uncertain or incorrect, making such nodes more susceptible to adversarial perturbation.

**Risk Assessment.** Risk assessment refers to the empirical evaluation of model performance across multiple random splits (Errica et al., 2020). Given $K$ random splits of $\mathcal{V}$ into disjoint subsets $\mathcal{V}_{\text{train}}$, $\mathcal{V}_{\text{valid}}$, and $\mathcal{V}_{\text{test}}$, the model is trained on $\mathcal{V}_{\text{train}}$ and tuned on $\mathcal{V}_{\text{valid}}$. For each split $k$, the best hyper-parameter configuration is selected based solely on validation performance. The empirical risk is then estimated by averaging the test performance across the $K$ splits.

**Model Selection.** Model selection aims to identify the hyper-parameter configuration that yields the highest validation accuracy. However, validation accuracy is often a biased estimator of generalization performance (Errica et al., 2020; Cawley & Talbot, 2010). Overreliance on validation performance can lead to overfitting and inflated expectations. In adversarial GNN literature, model selection and final evaluation are often conflated, undermining fair comparisons across attack strategies. Proper separation between model selection and risk assessment is essential to avoid misleading conclusions.

## 4 GRAPH ADVERSARIAL ATTACKS

Adversarial attacks on graphs aim to perturb either the structure or features of a graph $\mathcal{G} = (\mathbf{A}, \mathbf{X})$ in order to degrade the performance of a GNN. We refer to the targeted model as the **victim** model. The attack modifies $\mathcal{G}$ into a perturbed version $\mathcal{G}' = (\mathbf{A}', \mathbf{X}')$, leading the victim to misclassify selected nodes.

**Attacker's Capacity.** The adversarial attack can introduce perturbations to data either in the inference or training phases. In the **evasion** setting, the victim model trains on clean graph data $\mathcal{G}$ to perform inference on the perturbed data $\mathcal{G}'$. In the **poisoning** setting, adversarial attacks create a modified graph $\mathcal{G}'$, which is then used to train a model.

**Perturbation Type.** We perturb $\mathcal{G}$ within a given budget $\Delta$ by adding or removing edges from $\mathcal{E}$. Formally, we can write

$$\sum_u \sum_v |\mathbf{A}_{uv} - \mathbf{A}'_{uv}| \leq \Delta \tag{2}$$

**Attacker's Knowledge.** Attacks differ in the information available to the adversary. In **black-box** settings, the attacker lacks access to model parameters and labels. **White-box** attacks assume full access to both, a strong but often unrealistic assumption. In **gray-box** settings, the attacker can access the training data and labels, allowing them to train a **surrogate** model that approximates the victim. We adopt the gray-box setting, as it balances realism with the ability to diagnose vulnerabilities. Unlike prior work that uses fixed surrogates, we also evaluate **adaptive** attacks where perturbations are directly optimized against defended victim models, simulating stronger adversaries. Note that our evaluation pipeline is modular and extensible to all attack types.

**Attacker's Target.** We focus on **targeted** attacks, where a chosen subset of nodes $\mathcal{V}_T \subseteq \mathcal{V}_{test}$ are perturbed to induce misclassification, as they are often harder to detect in real systems.

**Victim Models.** We define two classes of victim models: **vanilla** GNNs (Bacciu et al., 2020), which are not trained with adversarial robustness in mind, and **defended** GNNs, which incorporate explicit defense mechanisms. Attacks against vanilla models define the baseline vulnerability, while defended scenarios test the effectiveness of robustness interventions. We emphasize that defense approaches generally operate without any prior knowledge of specific attacks.

## 4.1 ATTACK MODELS AND PITFALLS OF EVALUATION

Many attack evaluations in the literature suffer from inconsistent experimental setups, limiting fair comparison. Details of evaluation pitfalls for specific attacks are discussed in Appendix E; here we formalize criteria for rigorous assessment.

**Target Node Selection.** Most prior works follow the Nettack (Zügner et al., 2018) strategy, selecting (i) the 10 nodes with the highest margin of classification, indicating evident correctness; (ii) the 10 nodes with the lowest margin (still correctly classified); (iii) 20 additional nodes randomly chosen. This strategy may underrepresent high-degree nodes, which are harder to attack due to their richer neighborhood context (Figure 8). This bias inflates attack performance and skews conclusions.

**Evaluation Criteria.** A high-quality evaluation should satisfy the following: (i) the victim model has undergone a model selection process, as it usually happens in real-world scenarios; (ii) results are averaged over $K$ random splits with standard deviations and public splits; (iii) target nodes include diverse structural types; and (iv) evaluations include both vanilla and defended victims. Our benchmark adheres to all of these conditions.

**Attack Models.** We benchmark seven widely cited attack methods, selected based on peer-review status, architectural diversity, and citation count. These are Nettack (Zügner et al., 2018), FGA (Chen et al., 2018), SGA (Li et al., 2020a), GOttack (Alom et al., 2025), PR-BCD (Geisler et al., 2021), and PGD (Xu et al., 2019a). Full summaries and surrogate configurations appear in Appendix Section J.

**Victim Models.** Vanilla victim models are three standard GNNs: GCN (Kipf & Welling, 2017), GSAGE (Hamilton et al., 2017), and GIN (Xu et al., 2019b), each using a single aggregation function, and a fourth vanilla model, PNA (Corso et al., 2020), which combines multiple aggregation operations. We also evaluate nine defended victim models, selected according to the taxonomy in Appendix Table 13, with selection criteria detailed in Appendix G.

**Adaptive Attacks.** Adaptive attacks are designed with full awareness of the defense, producing stronger and more targeted perturbations. We evaluate PR-BCD in both its fixed-surrogate (PR-BCD (NA)) and adaptive variants. Though non-adaptive PR-BCD may underestimate its true capability, we include it due to its scalability, popularity, and baseline strength (later results in Tables 27 and 28 will show marginal differences between the variants). The scope of this work is to benchmark adversarial attacks and defenses in a practical setting, where neither attackers nor defenders have access to the opponent's backbone model or strategy. Consequently, evaluating defenses against fully adaptive attacks specifically crafted to circumvent their core mechanisms falls outside the focus of this study. Nevertheless, we acknowledge that adaptive evaluation can provide a more accurate and reliable lower bound on robustness (Mujkanovic et al., 2022), and that stronger, defense-tailored adaptive attacks exists (Dong et al., 2025).

**Naïve Baseline.** We introduce $L^1$D-RND, a simple yet effective baseline attack. Instead of using gradients or learned surrogates, $L^1$D-RND perturbs the graph by modifying edges connected to nodes selected using their degree and features. Despite its simplicity and low computational cost, it achieves surprisingly strong results, underscoring the importance of including naïve baselines to contextualize claimed improvements. Algorithm 3 and implementation details are provided in Appendix I.

## 4.2 RISK ASSESSMENT IN ADVERSARIAL EVALUATION

Unifying the good practices of Errica et al. (2020) and Shchur et al. (2018), the pseudo-algorithm of our proposed adversarial attacks evaluation is provided in Algorithm 1.

We first obtain $K$ different random splits from datasets, (Line 3). The victim model's hyperparameters are first tuned on the $i$-th split's training set, and the best victim model *for that split* is chosen based on the performance on the validation set (Line 6).

The model selection process relies solely on the training and validation sets to ensure an unbiased risk estimation. It is noteworthy that model selection of all models is performed on clean data (data without perturbation). Given the predictions of the best victim model, we sample a subset of target test nodes $\mathcal{V}_T$ that have been correctly classified (Line 9). An adversarial example on a given target node is considered a successful attack if it causes the victim model to flip its prediction about the target node.

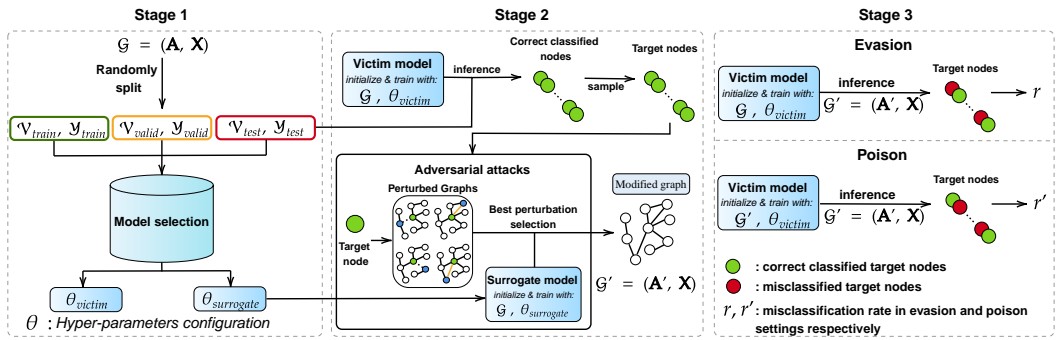

Figure 1: Overview of our risk assessment framework for adversarial GNN evaluation.

The adversarial attack performance is evaluated based on the misclassification rate on a specific budget $\Delta$, averaged over $K$ different splits and $R$ risk assessment runs for each split. Figure 1 visualizes the overall proposed evaluation pipeline. As the hyper-parameter configurations of victim models are carefully selected through the model selection process, we also perform model selection on surrogate models used in adversarial attacks to ensure that the process is realistic. The model selection process on victim models and surrogate models is kept on the same hyper-parameter grids.

# 5 EXPERIMENTS

We conduct extensive experiments to re-evaluate adversarial attacks under the standardized framework described in Section 4.2.

**Experimental Setup.** We adopt a transductive, semi-supervised node classification setting. For the evaluation procedure defined in Section 4.2, we set $K = 5$ and $R = 3$. Following common practice (Zügner et al., 2018; Alom et al., 2025), the $\mathcal{V}_{train}/\mathcal{V}_{valid}/\mathcal{V}_{test}$ ratio is set to 10/10/80.

We implement early stopping with the patience parameter $n$, where training stops if $n$ epochs have passed without improvement on the validation set. Importantly, the same data split $F_i$ (Line 3) is shared across different models to ensure a fair comparison.

We perform model selection for all victim and surrogate models, in both vanilla and defended scenarios, based on their performance on the validation sets. For each split, we evaluate adversarial attacks equipped with surrogate models. We report the average misclassification rate of adversarial attacks on vanilla and defended models across initialization seeds and splits, and report the percentage of nodes misclassified by the model. We fixed the same model as a surrogate model for each adversarial method (Table 16) for all evaluation settings, regardless of the choice of victim models, as attackers may not always know the classifier's architecture prior to performing the attack in practice.

---

**Algorithm 1** Adversarial attack/defense evaluation

1: **Input:** Dataset $\mathcal{D}$, Configs $\Theta$, Attack method **Attack**, Budget $\Delta$, Splits $K$, Runs $R$
2: **Output:** Avg. misclassification rates in both evasion and poison settings
3: Create $K$ train/val/test splits $F_1, .., F_K$ from $\mathcal{D}$
4: **for** $i = 1, ..., K$ **do**
5:     $\mathcal{V}_{train}^i, \mathcal{V}_{valid}^i, \mathcal{V}_{test}^i = F_i$
6:     $\theta_{best}^i$ = select($\Theta, \mathcal{V}_{train}^i, \mathcal{V}_{valid}^i, \mathcal{D}$) // Alg. 2
7:     **for** $r = 1, ..., R$ **do**
8:         $f$ = train($\theta_{best}^i, \mathcal{V}_{train}^i, \mathcal{V}_{valid}^i, \mathcal{D}$)
9:         $\mathcal{V}_T$ = node_select($f, \mathcal{V}_{test}^i, \mathcal{D}$)
10:         **for** $v$ in $\mathcal{V}_T$ **do**
11:             $\mathcal{D}'$ = **Attack**$(v, \mathcal{D}, \Delta)$
12:             $s_{v,r}^i = 1$ if $f(v) = y_v$, else 0 otherwise // evasion
13:             $f'$ = train($\theta_{best}^i, \mathcal{V}_{train}^i, \mathcal{V}_{valid}^i, \mathcal{D}'$) // retrain
14:             $s_{v,r}^{',i} = 1$ if $f'(v) = y_v$, else 0 // poison
15:         reset **Attack**
16:         **end for**
17:     **end for**
18: **end for**
19: success rate = $\frac{\sum_{i=1}^K \sum_r^R \sum_t^{\mathcal{T}} s_{t,r}^i}{K \times R \times |\mathcal{T}|}$ // evasion
20: success rate$'$ = $\frac{\sum_{i=1}^K \sum_r^R \sum_t^{\mathcal{T}} s_{t,r}^{',i}}{K \times R \times |\mathcal{T}|}$ // poison
21: **Return:** success rate, success rate$'$

---

**Hyper-parameters.** Model selection varies hyperparameters, including the number of layers, embedding dimensions, learning rate, dropout, and weight decay, based on ranges provided in original publications. Additional model-specific parameters (e.g., batch size, aggregation type) are included as needed. Full details are in Appendix C.

Table 1: Descriptive statistics of datasets.

| Type | Dataset | Nodes | Edges | Features | Labels |
|---|---|---|---|---|---|
| **Homophilic** | CORA | $2,708$ | $5,069$ | $1,432$ | 7 |
| | CITESEER | $3,327$ | $3,668$ | $3,703$ | 6 |
| | PUBMED | $19,717$ | $44,325$ | $500$ | 3 |
| **Heterophilic** | CHAMELEON | $2,277$ | $36,101$ | $3,132$ | 5 |
| | SQUIRREL | $5,201$ | $217,073$ | $3,148$ | 5 |
| **Large scale** | OGB-ARXIV | $169,343$ | $1,166,243$ | $128$ | 40 |

Table 2: **Homophily Results.** Evaluating adversarial attacks with budget $\Delta = 1$ in both evasion and poison settings on GCN (vanilla attack) and GNNGuard (defended attack). NA indicates a non-adaptive variant.

| | Victim | Attack Model for Evasion | | | | | | | Attack Model for Poisoning | | | | | | |
|---|---|---|---|---|---|---|---|---|---|---|---|---|---|---|---|
| | | L¹D-RND | FGA | NETTACK | PGD | PR-BCD (NA) | SGA | GOttack | L¹D-RND | FGA | NETTACK | PGD | PR-BCD (NA) | SGA | GOttack |
| CORA | GCN | 13.20 ± 0.04 | 27.87 ± 0.04 | 29.60 ± 0.05 | 29.33 ± 0.04 | **32.13 ± 0.04** | 26.27 ± 0.04 | 28.53 ± 0.04 | 15.47 ± 0.04 | 30.00 ± 0.06 | **33.47 ± 0.04** | 31.73 ± 0.04 | 32.80 ± 0.06 | 29.33 ± 0.04 | 33.33 ± 0.07 |
| CORA | GNNGuard | 6.27 ± 4.13 | 6.67 ± 3.44 | 6.80 ± 4.77 | 6.80 ± 2.60 | 8.13 ± 3.34 | **8.40 ± 4.29** | 8.40 ± 4.97 | 6.93 ± 4.40 | 7.47 ± 3.96 | 7.47 ± 5.37 | 6.93 ± 3.01 | 8.93 ± 3.99 | 9.60 ± 3.64 | **10.27 ± 6.54** |
| CITESEER | GCN | 15.20 ± 0.04 | 25.47 ± 0.04 | 28.13 ± 0.07 | 25.47 ± 0.05 | **34.53 ± 0.07** | 23.47 ± 0.03 | 25.60 ± 0.04 | 16.27 ± 0.04 | 31.87 ± 0.07 | **36.40 ± 0.07** | 30.80 ± 0.07 | 34.27 ± 0.06 | 25.20 ± 0.04 | 34.27 ± 0.08 |
| CITESEER | GNNGuard | **4.67 ± 3.68** | 3.33 ± 3.18 | 4.67 ± 2.35 | 3.07 ± 2.49 | 3.07 ± 2.12 | 4.00 ± 2.73 | 4.67 ± 2.89 | 4.80 ± 3.84 | 4.40 ± 3.31 | **6.00 ± 2.93** | 3.20 ± 2.11 | 3.20 ± 1.97 | 4.80 ± 3.00 | 4.80 ± 2.70 |
| PUBMED | GCN | 10.93 ± 0.03 | 35.60 ± 0.03 | 33.73 ± 0.03 | 34.13 ± 0.04 | 29.60 ± 0.03 | 34.27 ± 0.04 | **35.87 ± 0.03** | 9.73 ± 0.03 | 35.33 ± 0.03 | 34.13 ± 0.05 | 33.60 ± 0.03 | 29.60 ± 0.03 | 34.13 ± 0.04 | **35.60 ± 0.03** |
| PUBMED | GNNGuard | **6.53 ± 4.63** | 3.60 ± 1.88 | 2.93 ± 1.67 | 2.53 ± 1.60 | 4.27 ± 2.25 | 3.47 ± 1.92 | 3.07 ± 2.25 | **6.53 ± 4.93** | 4.80 ± 3.19 | 4.40 ± 2.53 | 4.27 ± 2.12 | 5.60 ± 4.08 | 4.93 ± 3.20 | 4.93 ± 3.10 7.87 |

Table 3: **Heterophily Results.** Evaluating adversarial attacks with budget $\Delta = 1$ in both evasion and poison settings on GCN (vanilla attack) and RUNG (defended attack). NA indicates a non-adaptive variant.

| | Victim | Attack Model for Evasion | | | | | | | Attack Model for Poisoning | | | | | | |
|---|---|---|---|---|---|---|---|---|---|---|---|---|---|---|---|
| | | L¹D-RND | FGA | NETTACK | PGD | PR-BCD (NA) | SGA | GOttack | L¹D-RND | FGA | NETTACK | PGD | PR-BCD (NA) | SGA | GOttack |
| SQUIRREL | GCN | 24.93 ± 33.09 | 62.40 ± 14.64 | 1.87 ± 3.34 | 47.73 ± 10.25 | **69.87 ± 10.76** | 52.00 ± 6.19 | 13.33 ± 4.64 | 34.67 ± 27.36 | 63.87 ± 12.25 | 2.80 ± 2.24 | 52.27 ± 8.21 | **70.27 ± 11.16** | 53.47 ± 5.97 | 13.60 ± 3.79 |
| SQUIRREL | RUNG | 2.13 ± 1.77 | 1.87 ± 2.88 | 0.27 ± 1.03 | 2.00 ± 1.85 | 1.73 ± 2.49 | **2.67 ± 3.68** | 0.93 ± 2.25 | 11.33 ± 8.64 | **20.53 ± 9.69** | 6.53 ± 4.44 | 17.33 ± 7.81 | 15.07 ± 6.18 | 18.40 ± 9.33 | 6.27 ± 5.18 |
| CHAMELEON | GCN | 21.87 ± 28.89 | 62.40 ± 7.72 | 3.07 ± 2.60 | 44.00 ± 20.95 | 58.00 ± 18.53 | 45.47 ± 10.38 | 23.47 ± 8.16 | 35.60 ± 22.31 | **66.40 ± 8.25** | 7.87 ± 4.98 | 53.47 ± 17.98 | 64.80 ± 14.69 | 51.47 ± 9.12 | 27.07 ± 8.48 |
| CHAMELEON | RUNG | **2.27 ± 3.28** | 0.93 ± 1.83 | 0.27 ± 0.70 | 1.87 ± 4.69 | 0.80 ± 2.24 | 2.13 ± 4.31 | 0.67 ± 1.23 | 12.00 ± 8.88 | **16.40 ± 7.53** | 7.33 ± 2.89 | 13.73 ± 6.41 | 11.73 ± 5.90 | 14.27 ± 6.76 | 10.80 ± 4.89 |

**Target Node Selection.** For each experiment, we evaluate on 50 target nodes selected to ensure diversity in classification margin and structural role: ii) 10 correctly classified nodes with the lowest degree, iii) 10 correctly classified nodes with the highest margin, iv) 10 nodes with the lowest margin (but still correctly classified) and v) 10 randomly chosen nodes.

**Datasets.** We evaluate on six datasets: three homophilous graphs (CORA, CITESEER, PUBMED (Yang et al., 2016)), two heterophilous graphs (SQUIRREL, CHAMELEON (Rozember-czki et al., 2021)), and one large-scale benchmark from OGB (Hu et al., 2020b). Many adversarial and defense methods do not scale well to large graphs, and we highlight such limitations where relevant (see Appendix D).

**Computational Environment.** Experiments were run using Python 3.8.19 and PyTorch 2.3.0 on a Linux cluster with Intel Xeon Gold 6338 CPUs (128 cores), 251 GB RAM, and NVIDIA RTX A40 GPUs with 44 GB memory. GNNs were implemented using PyTorch Geometric 2.5.3, and we reused code from DeepRobust (Li et al., 2020b), GreatX (Wu et al., 2022), and author-provided repositories for defense methods not in those libraries.

**Reproducibility.** We release all code, dataset splits, and model selection hyperparameters to support reproducible benchmarking with minimal overhead. Code is available at: `https://anonymous.4open.science/r/Adversarial-Benchmark`.

# 6 RESULTS AND DISCUSSION

This section provides an in-depth discussion of our results. We discuss vanilla models' attacks in Section 6.1 and defense models' attacks in Section 6.2. Notably, higher misclassification rates reflect more effective attack models. Time and GPU cost results are detailed in the Appendix F.3 due to limited space.

**Computational considerations.** Our experiments include up to 437,075 training runs (see Appendix F for a breakdown). In some cases, model selection or attacks on a single split exceeded 120 hours, making full experiments infeasible. We capped training time at 120 hours; results exceeding this are marked as OOR (Out of Resource).

## 6.1 VANILLA EVASION AND POISONING ATTACKS

We evaluate seven adversarial attack models under evasion and poisoning settings on vanilla GNNs across homophily and heterophily datasets. Appendix Tables 18, 19, and 20 report complete results. For conciseness, we also provide reduced summaries in Tables 2 and 3, and Figure 3.

**Homophily datasets.** PR-BCD (NA) has the highest misclassification rates (i.e., best attack model) in Table 18 for a budget of $\Delta = 1$ in 4 out of 12 evasion scenarios (spanning three datasets and four victim models). The remaining cases are distributed among other methods, with $L^1$D-RND surprisingly yielding the highest misclassification rates in PUBMED and CITESEER when PNA is the victim model. When the budget is increased ($\Delta = 2, \ldots, 5$), Nettack demonstrates superior performance in 36 out of 48 cases in Table 18. Unlike the low-budget case, PR-BCD (NA) achieves the best results only in CITESEER and CORA for GCN. In scenarios where GCN is the victim model, PR-BCD (NA), which uses GCN as the surrogate model, delivers competitive performance, surpassing Nettack in 10 out of 15 settings; these are the $\Delta = 1, \ldots, 5$ budgets in CORA and CITESEER in Table 18. However, PR-BCD (NA) shows limited adversarial effectiveness on victim models that differ significantly from the surrogate ones. For example, PR-BCD (NA) no longer outperforms FGA on GIN, GSAGE and PNA in evasion attacks. In Table 18, the baseline $L^1$D-RND exhibits the lowest performance based on average rank on homophily datasets (i.e., 7 out of 7 attack models). Notably, unlike the attack models, the baseline does not achieve high misclassification rates with increasing budgets.

As shown in the lower rows of Table 18, poisoning attacks are significantly more effective than evasion attacks. For $\Delta = 1$, the best-attack model, Nettack, achieves a $4.72\%$ relative increase, rising from an average of $27.76\%$ in evasion to $32.48\%$ in poisoning across three datasets and four victim models. Even the baseline, $L^1$D-RND, experiences a $8.9\%$ improvement in poisoning attacks. Nettack remains the best-performing model when ranks are averaged over all five budgets, with an overall rank of 1.65 across three datasets and four models. FGA follows as the second-best model, with an average rank of 3.23.

Among three homophily datasets, PUBMED has the lowest misclassification rates for $\Delta = 5$, whereas attack models reach 70% in CORA and CITESEER in Table 18. With CORA and CITESEER, even the $L^1$D-RND baseline makes considerable gains in misclassification with increasing budgets.

In addition, attacks provide a critical lens to evaluate the robustness of victim models under adversarial conditions. As Table 18 shows, **GraphSAGE is the most resilient victim model in both evasion and poisoning attacks**. In evasion, Nettack yields the lowest average misclassification rate of $44.8\%$ against GraphSAGE across 15 budgets (three datasets and five budgets). GIN follows with an average misclassification rate of $48.5\%$. In poisoning, Nettack has an average of $47.97\%$ misclassification rate on GraphSAGE models in three datasets across all budgets; other victim models have misclassification rates in $[53.4\%, 57.4\%]$.

**Heterophily datasets.** As shown in Table 19 and 20, under budget 1, the average misclassification rate across seven attacks on four non-defense models is 33.96% for heterophily datasets and 27.78% for homophily datasets, while the average misclassification rate is 52.21% for homophily datasets and 46.86% for heterophily datasets under budget 5. This suggests that the first **perturbation has a greater adversarial impact in heterophily settings**. However, **with increasing perturbation budgets, attacks tend to yield larger gains on homophily datasets**. Nettack demonstrates the highest effectiveness on homophily datasets with an average rank of 1.64, but its performance significantly drops on heterophily datasets, where it ranks 5.48, the second worst. Interestingly, another evaluation shows that FGA achieves an average rank of 3.10 on homophily datasets (second-best), but rises to 1.92 on heterophily datasets, making it the top-performing attack in that setting. Similarly, the naïve random attack $L^1$D-RND shows the opposite trend; it performs surprisingly well on heterophily datasets with an average rank of 2.33, but performs the worst, with an average rank of 6.27, on homophily datasets among the seven adversarial attack methods.

**Large-scale dataset.** On the moderately sized OGB-ARXIV dataset, which contains fewer than 200K nodes and is still considered small by industry standards, only three attack methods ($L^1$D-RND, PR-BCD (NA), and SGA) and two victim models (GCN and GSAGE) could be fully evaluated within the 120-hour compute limit. As shown in Table 4, SGA consistently achieves the highest misclassification rates across budgets 1 to 5 in both evasion and poisoning settings. **This result underscores a critical limitation: most existing adversarial attacks are not scalable enough to be**

**applied even to modestly large graphs, raising concerns about their practicality in real-world deployments.**

Table 4: Misclassification rate (↑) on OGB-ARXIV with budget $\Delta = 1$ to 5 in both evasion and poison setting on GCN and GSAGE.

| | Attack | GCN | | | | | GSAGE | | | | |
|---|---|---|---|---|---|---|---|---|---|---|---|
| | | 1 | 2 | 3 | 4 | 5 | 1 | 2 | 3 | 4 | 5 |
| Evasion | L$^1$D-RND | $16.00 \pm 2.00$ | $30.67 \pm 7.02$ | $36.00 \pm 2.00$ | $38.00 \pm 2.00$ | $36.67 \pm 3.06$ | $16.67 \pm 8.08$ | $27.33 \pm 7.02$ | $\underline{33.33 \pm 7.57}$ | $\underline{29.33 \pm 9.45}$ | $34.67 \pm 7.02$ |
| | PR-BCD (NA) | $\underline{23.33 \pm 3.06}$ | $\underline{34.00 \pm 3.46}$ | $38.00 \pm 2.00$ | $40.67 \pm 3.06$ | $39.33 \pm 1.15$ | $20.67 \pm 1.15$ | $24.67 \pm 1.15$ | $22.00 \pm 3.46$ | $24.67 \pm 3.06$ | $28.00 \pm 2.00$ |
| | SGA | $\mathbf{36.67 \pm 5.03}$ | $\mathbf{48.67 \pm 7.02}$ | $\mathbf{56.00 \pm 2.00}$ | $\mathbf{57.33 \pm 1.15}$ | $\mathbf{58.67 \pm 1.15}$ | $\mathbf{40.67 \pm 2.31}$ | $\mathbf{56.00 \pm 10.00}$ | $\mathbf{63.33 \pm 4.16}$ | $\mathbf{72.00 \pm 7.21}$ | $\mathbf{71.33 \pm 5.77}$ |
| Poison | L$^1$D-RND | $17.33 \pm 4.16$ | $32.00 \pm 6.00$ | $\underline{38.00 \pm 2.00}$ | $39.33 \pm 1.15$ | $38.67 \pm 2.31$ | $17.33 \pm 5.03$ | $\underline{26.00 \pm 3.46}$ | $\underline{30.00 \pm 14.00}$ | $31.33 \pm 6.43$ | $34.67 \pm 7.02$ |
| | PR-BCD (NA) | $22.00 \pm 5.29$ | $34.67 \pm 3.06$ | $36.67 \pm 1.15$ | $\underline{40.67 \pm 5.03}$ | $39.33 \pm 1.15$ | $\underline{17.33 \pm 3.06}$ | $24.00 \pm 2.00$ | $19.33 \pm 3.06$ | $26.00 \pm 2.00$ | $26.00 \pm 2.00$ |
| | SGA | $\mathbf{36.67 \pm 2.31}$ | $\mathbf{48.67 \pm 8.08}$ | $\mathbf{56.00 \pm 2.00}$ | $\mathbf{57.33 \pm 1.15}$ | $\mathbf{58.67 \pm 1.15}$ | $\mathbf{40.00 \pm 3.46}$ | $\mathbf{60.00 \pm 8.00}$ | $\mathbf{61.33 \pm 5.77}$ | $\mathbf{70.67 \pm 5.03}$ | $\mathbf{74.67 \pm 8.08}$ |

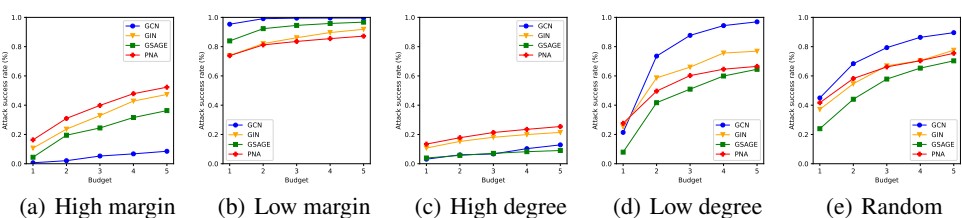

| (a) High margin | (b) Low margin | (c) High degree | (d) Low degree | (e) Random |
|---|---|---|---|---|

Figure 2: Average misclassification rate for different node categories of four non-defense models caused by seven adversarial attacks on three homophily datasets in the poison setting.

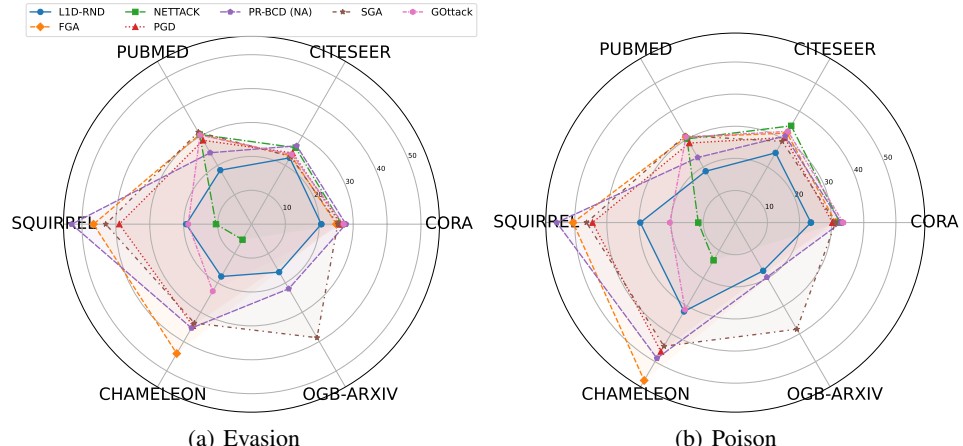

| (a) Evasion | (b) Poison |
|---|---|

Figure 3: **Vanilla Results.** Average mis-classification rate (%) of adversarial attacks across all vanilla models under budget 1.

## 6.2 Defended Evasion and Poisoning Attacks

The results for evasion and poisoning attacks are shown in Appendix Tables 23, 22 for homophily datasets and Tables 24, 26 for heterophily datasets. We also show reduced results in Table 2 and Table 3, lower rows, and Figure 4.

**Homophily datasets.** Appendix Tables 23 and 22 demonstrate that defense models substantially reduce misclassification rates, with poisoning attacks being generally easier to defend against than evasion. For example, Nettack's average misclassification rate drops from $49.1\%$ to $32.2\%$ under evasion, and from $53.6\%$ to $23.56\%$ under poisoning when defenses are applied. Nettack still ranks highest in 8 out of 15 defended poisoning scenarios.

Among defense methods, GNNGuard is the most effective: at $\Delta = 1$, it achieves an average misclassification rate (i.e., best defense) of just $5.01\%$ for evasion and $5.92\%$ for poisoning across three datasets and seven attacks. GRAND consistently ranks as the second-best defense. In contrast,

the FGA attack model, despite its strong performance in the vanilla setting, performs poorly against all defended models.

The performance of the $L^1$D-RND baseline is noteworthy: this simple, naive attack achieves the highest misclassification rate in 19 out of 45 defended evasion settings and 18 out of 45 defended poisoning settings (across three datasets, three victim models, and five budgets). **Its surprisingly strong performance, despite lacking any optimization or model-specific tuning, calls into question the actual gains offered by several state-of-the-art adversarial attack methods.**

To highlight key trends, we focus on the best-performing defense, GNNGuard, and its performance against vanilla attacks on the widely used GCN model, as summarized in Table 2.

At lower budgets (e.g., $\Delta = 1$), which represent realistic perturbation scenarios such as the addition or removal of a single edge, Nettack does not outperform any other model in evasion attacks. When defenses are applied, Nettack's evasion effectiveness often falls below that of PR-BCD (NA). However, in poisoning attacks, Nettack remains strong, with GOttack emerging as the second-best method.

Overall, **Table 2 highlights the robustness of GNNGuard, which substantially reduces the effectiveness of advanced attacks in both evasion and poisoning settings, often lowering their impact to the level of the naive $L^1$D-RND baseline.**

**Heterophily datasets.** As shown in Tables 24 and 26, defenses are more effective on heterophily datasets than on homophily datasets. Across budgets $\Delta = 1$ to 5, average misclassification rates on homophily datasets are $[20.11\%, 40.82\%]$, while the corresponding rates on heterophily datasets are substantially lower: $[16.08\%, 23.33\%]$.

This discrepancy suggests that **heterophily graphs are more resistant to adversarial perturbations**, likely due to weaker local homogeneity, which reduces the impact of structural changes. Under defense, attack methods achieve abysmally low rates on heterophily datasets compared to homophily datasets. Notably, FGA becomes the top-performing attack on heterophily datasets (rank 1.92), while Nettack's performance degrades sharply, dropping from rank 2.08 to 6.10, the worst among all methods. The best defense model, RUNG (Hou et al., 2024), reduces most attack rates to 2%. This underscores that **novel approaches are needed in heterophily settings**, marking this as an open and underexplored research area.

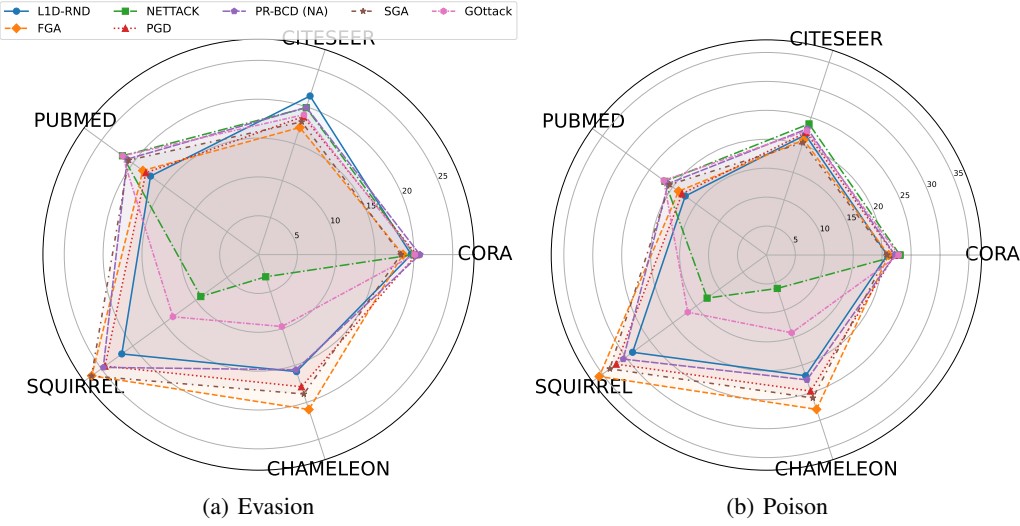

(a) Evasion                 (b) Poison

Figure 4: **Defended Results.** Average mis-classification rate (%) of adversarial attacks across all defended models under budget 1.

**The Impact of Target Node Selection.** Target node selection plays a critical role in evaluating adversarial attacks. In the literature, Nettack's strategy, selecting 10 high-margin nodes, 10 low-margin nodes, and 20 randomly chosen ones (totaling 40), has become a de facto standard. However, this approach ignores structural properties such as node degree, which substantially influence attack suc-

cess. Appendix Table 29 shows that using node degree as a selection criterion results in substantially lower attack success compared to margin-based or random selection, even in vanilla (undefended) settings. Figure 8 shows that all attack methods perform poorly on these nodes, while low-degree and low-margin nodes remain vulnerable. High-degree nodes exhibit alarmingly low misclassification rates, ranging from only $0.05$ to $0.28$, despite the absence of any defense. Crucially, because most attacks were developed and benchmarked on small graphs with low average degree (e.g., CORA), this vulnerability remained undetected, largely due to scalability limitations that prevent testing on larger, high-degree networks. This suggests that **evaluations ignoring node degree systematically overstate both the effectiveness of attack models and the fragility of GNNs and raises serious concerns about the applicability of current attack models to real-world networks**, where average node degrees are much higher (Rossi & Ahmed, 2025). Our results reveal that low-degree nodes tend to be substantially more vulnerable, whereas high-degree nodes exhibit stronger inherent robustness under adversarial attacks. This suggests that future defense mechanisms may benefit from prioritizing protection for low-degree nodes rather than applying a uniform strategy across the entire graph.

**The Impact of Victim Model Selection.** In practical deployments, victim models are selected based on performance over training and validation sets. However, many adversarial GNN studies evaluate attacks on fixed, non-optimized model configurations, ignoring this critical step. Our results in Table 30 show that incorporating model selection into evaluations can significantly alter attack outcomes. On CORA and CITESEER, victim models chosen via model selection are generally more vulnerable: for example, SGA's misclassification rate on GIN in the poisoning setting differs by an average of $15.27\%$ across budgets. In contrast, on PUBMED, model selection sometimes leads to more robust victim models. This is particularly evident for GSAGE, where attacks are less effective on tuned models than on fixed ones.

These findings highlight a critical inconsistency: the perceived effectiveness of **adversarial attacks depends not only on the attack method but also on whether the victim model is realistically selected**. Evaluations that omit this step may either overstate or understate the vulnerability of GNNs, leading to misleading conclusions about attack strength.

**Limitations.** Our benchmark focuses on static graphs, which is the predominant setting in the adversarial GNN literature. While attacks on dynamic graphs and continuous-time embeddings are also important, they remain largely unexplored across existing benchmarks. We view these as natural and valuable extensions of our work rather than omissions, and anticipate that our framework can provide the foundation for evaluating such scenarios in the future.

## 7    CONCLUSION

We have conducted a large-scale evaluation of adversarial attacks and defenses on GNNs, revealing that conclusions from prior work often do not hold under fair and rigorous settings. While Nettack has remained a strong performer, the unexpectedly competitive results of our naive baseline, $L^1$D-RND, challenge assumptions about the progress made in adversarial graph learning. PR-BCD and FGA are scalable options. Our analysis has shown that dataset properties, target node selection, and victim model configuration significantly affect attack success, yet have been inconsistently addressed in past evaluations. Our findings highlight the need for standardized, practical benchmarks that reflect real-world constraints and model selection practices. By exposing gaps in current evaluation protocols, we have laid the groundwork for more reliable assessments of adversarial robustness in graph learning. We hope this work prevents the repetition of past methodological pitfalls and encourages more transparent and scalable evaluations moving forward.

## REPRODUCIBILITY STATEMENT

We release a complete anonymized codebase to ensure full reproducibility at `https://anonymous.4open.science/r/Adversarial-Benchmark`. All experiments are run with fixed random seeds, and model hyperparameters are obtained from performing model selection on set of all possible hyperparameters provided in Table 17. Additional details on compute resources and experimental setup are described in Section 5.

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
