# Appendix

## A LLM USAGE

We acknowledge the use of LLMs to aid or polish writing. We confirm that research motivation, methodology, idea and content comes from us, the authors, while the LLM's role was limited to refining clarity, grammar, style, and LaTeX formatting.

## B LIMITATION AND FUTURE WORK

While our benchmark explores a comprehensive range of hyper-parameters, we acknowledge that model weight initialization can also influence robustness outcomes. Ennadir et al. (2024b) highlights that different initializations may lead to noticeably different robustness levels. Due to the computational cost, over 437,000 additional experiments, our current benchmark utilized the initialization used by the original papers and does not perform hyper-parameter tuning on initialization. We therefore report the specific initialization scheme used for each model, leaving a full exploration of initialization robustness for future work.

Although this work focuses on topological attacks, our experimental pipeline and open-source code have been extended to support feature-based attacks. We therefore conduct feature-attack variants of FGA, NETTACK, PGD, PR-BCD, and SGA on GCN across homophily datasets. The results show that NETTACK and FGA consistently yield the highest misclassification rates in both evasion and poisoning settings, while SGA and PGD perform substantially weaker on PUBMED. We also observe the same trend as in structural attacks: high-degree nodes remain significantly more robust (11.92% average success rate) than low-degree nodes (94.85%). Despite these results, a full benchmark of feature-based attacks is beyond the scope of the current paper, and we acknowledge this as an important direction and leave it as future work, building upon our setting.

Our benchmark does not include an evaluation of certified robustness. While certification offers valuable theoretical guarantees, most defense methods in our study do not provide certification modules or compatible implementations, making it infeasible to run a fair and standardized certification comparison. We therefore consider certification an important but orthogonal direction, and we leave a systematic benchmark of certified defenses to future work.

## C HYPER-PARAMETERS

Table 17 presents the hyperparameters for various models. Model selections include the number of convolutional layers, the learning rate, and the early stopping criterion (based on either validation accuracy or validation loss) for all models. In addition, model-specific parameters such as regularization terms, dropout rates, and other configurations were chosen as appropriate for each model.

Moreover, we outline the hyperparameters for the model's specific parameters, as follows.

**RobustGCN**: The hyperparameters for the RGCN model are configured as follows: $\gamma = 0.5$, $\beta_1 = 1$, and $\beta_2 = 1.5$ across all datasets. The dropout rate $\in \{0.5, 0.6\}$, while the Adam optimizer is used for optimization with a learning rate $\in \{0.001, 0.01, 0.005\}$. The model is trained for a maximum of 1000 epochs, with early stopping applied based on the performance on the validation set.

**GARNET**: We set the hyperparameters as follows: the number of eigenpairs used for spectral embedding, $r \in \{50, 500\}$; the number of nearest neighbors used for base graph construction, $k \in \{50, 30, 10\}$; and the edge pruning threshold, $\gamma$, set to 0.0003. Moreover, we explore GARNET configurations where node features are either included or excluded to understand the role of node attributes. Similarly, for the construction of the nearest-neighbor graph, we assess both weighted and unweighted approaches. We evaluated the effect of normalizing versus not normalizing the adjacency matrix on stabilization and scaling in graph computations.

**GRAND**: We set the hyperparameters for the GRAND model as follows: the node drop rate is set to 0.5, the temperature parameter, $T \in \{0.1, 0.5, 0.3\}$, the regularization coefficient, $\lambda \in \{1.0, 0.5, 0.7\}$, and the order of graph convolutions, order $\in \{2, 4, 8\}$. Moreover, we explore GRAND configurations

by adjusting the number of samples, sample $\in \{2, 3, 4\}$, to examine the impact of sample size on learning.

Given a model, all possible choices of all hyperparameters form a set of hyperparameter configurations. We perform model selection using Algorithm 2 to select the configuration having the best performance on the validation dataset, $\mathcal{V}_{valid}$.

---

**Algorithm 2** Model Selection (select)

---

1: **Input:** Set of configurations $\Theta$, Dataset $\mathcal{D}$,
   Train indexes $\mathcal{V}_{train}$, Validation indexes $\mathcal{V}_{valid}$
2: **Output:** $\theta_{best}$: the best hyper-parameter configuration
3: $p_\theta = \emptyset$
4: **for** $\theta$ in $\Theta$ **do**
5:    $f = \text{train}(\theta, \mathcal{V}_{train}, \mathcal{V}_{test}, \mathcal{D})$
6:    performance = validate($\theta, \mathcal{V}_{valid}, \mathcal{D}$)
7:    $p_\theta = p_\theta \cup$ performance
8: **end for**
9: $\theta_{best} = \arg\max_\theta p_\theta$
10: **Return:** $\theta_{best}$

---

**Initializer.** Initialization plays a non-trivial role in determining the stability and robustness of GNN models. Consistent with findings in recent literature, different initializations may lead to different robustness levels, even under identical training settings. While our benchmark uses commonly adopted default initializers for each architecture, we recognize that further investigation into initialization sensitivity may reveal additional insights into model and defense behavior.

# D   DATASETS

We conducted experiments on six widely used node classification datasets, with their statistics provided in Table 1. Specifically, we use three homophilic graphs: CORA (Yang et al., 2016), CITESEER (Yang et al., 2016), and PUBMED (Yang et al., 2016) datasets are citation networks characterized by undirected edges and binary features, where nodes represent publications and edges correspond to citation links. We also include two heterophilic datasets CHAMELEON (Rozemberczki et al., 2019) and SQUIRREL (Rozemberczki et al., 2019) which are Wikipedia-based page-to-page networks focused on specific topics. In these datasets, nodes represent web pages, edges indicate mutual hyperlinks between pages, and node features are derived from several informative nouns found within the corresponding Wikipedia content. Moreover, we also consider one large-scale graph dataset OGB-ARXIV (Hu et al., 2020b), which is a directed graph representing the citation network among all Computer Science papers on arXiv. In this dataset, each node corresponds to an arXiv paper, and each directed edge indicates that one paper cites another.

## D.1   RELATED WORK

Adversarial robustness in graph neural networks has been actively studied since the first structural perturbation methods were introduced. Zügner et al. Zügner et al. (2018) presented Nettack, one of the earliest attacks on graph data, which demonstrated the vulnerability of GNNs to edge manipulations and inspired a large body of follow-up work.

Benchmarking efforts have also appeared. The Graph Robustness Benchmark Zheng et al. (2021) highlighted the need for reproducibility and scalability, but focused mainly on global evasion attacks without targeted or poisoning settings. More recent studies such as EGALA Hong & Hsieh and DistTack Zhang et al. (2024) explored scalability by approximating gradients or leveraging distributed training, yet these efforts remained tied to validating specific new attacks. Our work instead provides a systematic quantification of scalability across a wider range of attack families, datasets, and homophily regimes.

On the defense side, preprocessing-based strategies such as Jaccard filtering and SVD remain popular. Self-guided Robust Graph Structure Refinement In et al. (2024) exemplifies recent progress toward

adaptive and automated preprocessing defenses. We include GCN-Jaccard in our benchmark and note that such defenses complement the broader suite of defense mechanisms we evaluate.

Beyond edge manipulations, researchers have broadened adversarial settings. Zhu et al. Zhu et al. (2022) showed theoretically and empirically that heterophily strongly affects GNN robustness, aligning with our empirical results on homophilic versus heterophilic graphs. Restricted black-box attacks such as SheAttack Lei et al. and unnoticeable graph injection attacks Chen et al. (2022) further extend the threat model, illustrating how stealth and injection-based perturbations introduce challenges beyond simple edge flips.

While prior works have studied scalability Hong & Hsieh; Zhang et al. (2024), homophily effects Zhu et al. (2022), preprocessing defenses In et al. (2024), and new adversarial models Lei et al.; Chen et al. (2022), they do so in isolation. Our benchmark unifies these dimensions into a single reproducible framework, spanning 470K+ experiments, and provides a comprehensive evaluation protocol that complements and extends the scope of existing studies.

# E    ADVERSARIAL ATTACKS EVALUATION PITFALLS

**Nettack.** The adversarial impact on victim models was reported with an average of over five random initiations/splits. However, the results do not include standard deviation, and different train/validation/test splits are not publicly available, which makes it difficult to re-produce the evaluation experiments. The evaluation of Nettack is limited to a single victim model, GCN, with fixed hyperparameters; hence, it is unclear if model selection has been performed. In addition, the authors evaluate Nettack using a target node selection strategy that does not encompass all classes of nodes, particularly those with high degrees. This limitation results in an incomplete assessment of Nettack's impact on high-degree target nodes.

**FGA.** The authors evaluate FGA on multiple victim models, such as GCN (Kipf & Welling, 2017), GraRep (Cao et al., 2015), GraphCAN (Wang et al., 2018), trained and evaluated on at least a random train/validation/test split. However, whether the averaged results are reported from different random splits and whether model selection is performed for each victim model is unclear. The adversarial impact of FGA is assessed on target nodes selected randomly based on their labels. Importantly, the evaluation code has not been released, making it difficult to reproduce the results.

**SGA.** SGA is evaluated on multiple victim models such as GCN (Kipf & Welling, 2017), SGC (Wu et al., 2019a), GraphSAGE (Hamilton et al., 2017), etc,. However, similarly to FGA and Nettack, it is unclear whether the assessment and model selection are conducted over different random splits of training/validation/test sets. Moreover, the adversarial effect of SGA is evaluated on randomly selected target nodes, which take neither high-margin and low-margin nodes as used in Nettack nor nodes with high degrees.

**PR-BCD.** The article reports results obtained from the average over three random/splits and follows a target node selection strategy similar to Nettack according to the code. The evaluation process omitted model selection for each split; hence, PR-BCD is evaluated on different scalable victim models with fixed hyper-parameters for all splits. Importantly, the authors did not compare PR-BCD performance with popular baselines such as Nettack and FGA, even on small datasets, as the authors focus on robustness and adversarial performance on large graphs.

**PGD.** Similar to Nettack, PGD's performance is evaluated and reported with the mean and standard deviation of five different random splits on a single GCN classifier. However, data splits are not publicly released, making it impossible to reproduce the result, and it is not clear if GCN hyper-parameters are fixed in advance for every split or obtained from model selection for each split.

**GOttack.** GOttack is evaluated on three backbone GNNs (GCN, GIN, and GraphSAGE) and four defense models (RGCN, GCN-Jaccard, GCN-SVD, and MedianGCN), using fixed hyperparameters. It follows Nettack's target node selection strategy, which does not account for all classes of nodes, especially high-degree nodes. Additionally, the experiments were not conducted over different random splits of training/validation/test sets. This limits the evaluation of GOttack's effectiveness.

### E.1 Addressing Common Questions on Pitfalls

**Have attacks such as Nettack, FGA, PR-BCD, and GOttack not already been studied extensively?**

While these attacks have been evaluated in earlier work, prior studies were typically ad-hoc and limited: using fixed hyperparameters, single data splits, or narrow subsets of target nodes. Our benchmark corrects these shortcomings by conducting over 437,000 experiments across homophilic, heterophilic, and large-scale datasets, while incorporating proper model selection and multiple random splits. This provides the first systematic, large-scale, and reproducible comparison of adversarial GNN attacks.

**Were structural node properties, such as margin or degree, not already analyzed before?**

Some papers have presented anecdotal evidence on node properties, such as degree or margin, often restricted to small datasets like CORA (Zheng et al., 2021; Zhu et al., 2022). These analyses were not comprehensive or reproducible, as they relied on binning or one-off case studies. In contrast, our benchmark systematically shows that node degree strongly influences attack success rates across multiple datasets. Appendix Table 29 further confirms that degree-based target selection drastically reduces attack effectiveness, a pattern overlooked in prior work due to scalability and methodological limitations.

**Have victim model configurations and model selection not already been considered?**

Most previous evaluations fixed hyperparameters and skipped model selection, a limitation also noted in earlier benchmarks (Zheng et al., 2021). Our results demonstrate that incorporating realistic victim model selection can shift attack outcomes by more than 15% depending on the dataset and budget. This shows that omitting model selection produces misleading conclusions about attack strength. Our benchmark closes this gap by embedding model selection systematically into the evaluation.

## F   Detailed Results on Adversarial Attack Evaluation

**Experiment breakdown.**   Each instance involving training a victim model is counted as one experiment. Specifically, a total of 353,565 experiments are conducted for model selection across 14 models, including surrogate, vanilla, and defense models, on homophily, heterophily, and large-scale datasets. In contrast, 83,565 experiments are required to evaluate seven attack methods on all vanilla and defense models across all datasets. The breakdown of experiments for model selection and attack evaluation is presented in Table 5 and Table 6, respectively. In total, this study involves 437,075 experiments.

### F.1   Vanilla Evasion and Poisoning Attacks

**Homophily datasets.** In this section, we present the experimental results. Table 18 illustrates the success rates of six adversarial attacks in four vanilla models (GCN, GIN, GSAGE and PNA) in three homophily datasets under evasion and poisoning attack settings.

For the CORA dataset in the evasion attack setting, Nettack achieves the best performance in 12 of the 20 tasks, including 4 out of 5 tasks for each of the GIN, GSAGE and PNA models. However, in the GCN model, Nettack delivers the second-best results. In contrast, PR-BCD (NA) performs best in 5 of 5 tasks on the GCN model, although it only secures 7 of 20 tasks overall in the evasion setting. Interestingly, the proposed baseline $L^1$D-RND, achieves the highest score on the PNA model for budget $\Delta = 1$. Likewise, in the poisoning attack setting for the same dataset, Nettack demonstrates the best performance, securing 14 out of 20 tasks. Meanwhile, PR-BCD (NA), GOttack, SGA, FGA and PGD achieve the best scores on at least one task out of 20. For a budget $\Delta = 1$, the best-performing model, Nettack, achieves a 22.13% relative increase on the CORA dataset, rising from an average attack success rate of 26.65% in evasion to 32.56% in poisoning across four victim models.

On the CITESEER dataset in the evasion attack setting, Nettack achieves the best performance in 13 out of 20 tasks. Similarly to the CORA dataset, Nettack delivers the second-best results on the GCN model and the proposed baseline, $L^1$D-RND, achieves the highest score in the PNA model

Table 5: Break up of the numbers of experiments required for performing model selection for surrogate models, vanilla and defense models on homophily, heterophily and large-scale datasets. There are three homophily datasets, each of which has five training/validation/test splits. Each heterophily dataset, CHAMELEON and SQUIRREL, has also five training/validation/test splits. Due to high time complexity, we only consider one training/validation/testing split for large-scale dataset, OGB-ARXIV, provided by Hu et al. (2020a)

| Model | # combinations | homophily datasets ($3 \times 5$) | CHAMELEON ($\times 5$) | SQUIRREL ($\times 5$) | OGB-ARXIV ($\times 1$) | Total |
|---|---|---|---|---|---|---|
| GCN | 2187 | 32805 | 10935 | 10935 | 2187 | |
| GIN | 2187 | 32805 | 10935 | NA | NA | |
| GSAGE | 2187 | 32805 | 10935 | 10935 | 2187 | |
| PNA | 729 | 10935 | NA | NA | NA | |
| GCN-surrogate | 243 | 3645 | 1215 | 1215 | 243 | |
| SGC | 243 | 3645 | 1215 | 1215 | 243 | |
| GNNGuard | 48 | 720 | 240 | 240 | NA | |
| GRAND | 4374 | 65610 | 21870 | 21870 | NA | |
| ElasticGNN | 1296 | 19440 | 6480 | 6480 | NA | |
| RobustGCN | 324 | 4860 | 1620 | 1620 | NA | |
| GCORN | 27 | 405 | 135 | 135 | NA | |
| RUNG | 216 | 3240 | 1080 | 1080 | NA | |
| GCN-GARNET | 48 | 720 | 240 | 240 | NA | |
| GCN-Jaccard | 6 | 90 | NA | NA | NA | |
| **Total** | **14,115** | **211,725** | **66,900** | **55,965** | **4,860** | **353,565** |

Table 6: Break up of the number of experiments required for evaluating attacks on vanilla and defense models on homophily, heterophily and large-scale datasets

| Table index | # rows | # cols | # cells | Expected experiments | OOR | # Actual experiments |
|---|---|---|---|---|---|---|
| Table 4 | 6 | 10 | 60 | 180 | 0 | 180 |
| Table 18 | 56 | 15 | 840 | 12,600 | 0 | 12,600 |
| Table 19 | 28 | 5 | 140 | 2,100 | 0 | 2,100 |
| Table 20 | 42 | 5 | 210 | 3,150 | 0 | 3,150 |
| Table 23 | 56 | 15 | 840 | 12,600 | 0 | 12,600 |
| Table 22 | 56 | 15 | 840 | 12,600 | 70 | 12,530 |
| Table 24 | 28 | 10 | 280 | 4,200 | 0 | 4,200 |
| Table 26 | 56 | 10 | 560 | 8,400 | 0 | 8,400 |
| Table 27 | 4 | 15 | 60 | 900 | 0 | 900 |
| Table 28 | 4 | 10 | 40 | 600 | 0 | 600 |
| Table 29 | 56 | 15 | 840 | 12,600 | 0 | 12,600 |
| Table 30 | 56 | 15 | 840 | 12,600 | 0 | 12,600 |
| Table 31 | 14 | 5 | 70 | 1,050 | 0 | 1,050 |
| | | | | **Total:** | | **83,510** |

for a budget $\Delta = 1$. In contrast, PR-BCD (NA) achieves the best performance in all tasks in the GCN model, although it secures only 5 of the total 20 tasks in the evasion setting. Likewise, in the poisoning attack setting, Nettack demonstrates the best performance, securing 14 out of 20 tasks. Moreover, PR-BCD (NA) and FGA each score the best on 3 out of 20 tasks. However, Nettack achieves a 33.48% relative increase on the CITESEER dataset, rising from 26.06% in evasion to 34.79% in poisoning across four victim models for a budget $\Delta = 1$.

Likewise, on the PUBMED dataset, Nettack achieves the best performance in 11 out of 20 tasks in both the evasion and poisoning attack settings. In contrast, GOttack delivers the best performance in 5 out of 20 tasks in the evasion setting and 3 out of 20 tasks in the poisoning setting. Notably, the proposed baseline, $L^1$D-RND, secures the highest score on the PNA model for a budget of $\Delta = 1$ in both settings. Additionally, FGA achieves the best scores on 2 out of 20 tasks in the evasion setting and 3 out of 20 tasks in the poisoning setting. Furthermore, Nettack exhibits a 3.01% relative decrease on the PUBMED dataset, dropping from 31.03% in evasion to 30.09% in poisoning across four victim models for a single budget. On average, the misclassification rate of all attacks on GCN on the PUBMED dataset in both evasion and poison settings at budget $\Delta = 1$ drops by 22.59% when

high-degree nodes are selected as target nodes. Still, on average, a node degree-based selection of target nodes decreases the misclassification rate of adversarial attacks by $9.8\%$.

**Heterophily datasets.** This section presents the experimental results. Table 19 and 20 show the success rates of six adversarial attacks on three vanilla models (GCN, GIN, and GSAGE) across two heterophilic datasets under both evasion and poisoning attack settings.

For the SQUIRREL dataset under the evasion attack setting, PR-BCD (NA) achieves the highest performance in 5 out of 10 tasks, including 4 out of 5 tasks for the GCN model. FGA ranks second, performing best in 4 out of 10 tasks, with 4 out of 5 for the GSAGE model, and also securing the second-best results in 4 out of 5 tasks for the GCN model. Under the poisoning attack setting for the same dataset, both PR-BCD (NA) and FGA achieve top performance, each leading in 4 out of 10 tasks. In contrast, Nettack, GOttack, SGA, and PGD demonstrate the lowest success rates in both evasion and poisoning scenarios. For a budget $\Delta = 1$, the best-performing model, PR-BCD (NA), achieves a $4.76\%$ relative increase on the SQUIRREL dataset, rising from an average attack success rate of $53.07\%$ in evasion to $55.6\%$ in poisoning across three victim models.

Similarly, on the CHAMELEON dataset, the proposed naive $L^1$D-RND achieves the best performance in 7 out of 15 tasks under the evasion attack setting. In comparison, FGA and PR-BCD (NA) both demonstrate the second-best performance, each excelling in 4 out of 15 tasks. However, under the poisoning attack setting, FGA outperforms all other methods, achieving the best results in 10 out of 15 tasks, while PR-BCD (NA) and the proposed $L^1$D-RND achieve top results in 3 and 2 tasks, respectively. Consistent with observations on other heterophily datasets, Nettack, GOttack, SGA, and PGD show the lowest performance across both evasion and poisoning settings. However, PR-BCD (NA) demonstrates a $37.94\%$ relative improvement on the CHAMELEON dataset, increasing from $35.37\%$ in the evasion setting to $48.8\%$ in the poisoning setting across three victim models at a budget of $\Delta = 1$.

**Large-scale datasets.** Table 4 presents the success rates of three adversarial attacks on two vanilla models (GCN and GSAGE) using a large-scale dataset under both evasion and poisoning attack settings. On the OGB-ARXIV dataset, SGA achieves the best performance, outperforming others in all 10 out of 10 tasks across both settings. In comparison, PR-BCD (NA) secures the second-best performance in 7 out of 10 tasks under the evasion setting. Similarly, under the poisoning setting, both PR-BCD (NA) and the proposed $L^1$D-RND model achieve the second-best results in 5 out of 10 tasks.

## F.2 Defense Evasion and Poisoning Attacks

**Homophily datasets.** In Tables 23 and 22, we present the results of six adversarial attacks on 8 defense models, across three homophily datasets in both evasion and poisoning attack settings.

On ElasticGNN, the average performance of adversarial attacks is $23.35\%$ under a budget of 1 and $43.88\%$ under a budget of 5. Among all evaluated attacks, FGA exhibits the highest adversarial impact on ElasticGNN, achieving an average rank of 2.47 across both evasion and poisoning settings, followed closely by Nettack with an average rank of 2.57.

Conversely, while FGA performs best on ElasticGNN, it demonstrates surprisingly poor performance on GCN-GARNET, with an average rank of 5.27, ranking as the second-worst among the seven evaluated attacks. In contrast, $L^1$D-RND attains the second-best average rank of 2.47. This result indicates that the relative effectiveness of adversarial attacks varies significantly depending on the target defense model, underscoring the necessity of evaluating attacks across diverse defense strategies. On GCN-GARNET, the average attack performance is $18.89\%$ for budget 1 and $45.15\%$ for budget 5.

For GCN-Jaccard, the average attack performance is $24.76\%$ under budget 1 and $50.89\%$ under budget 5. Similar to ElasticGNN, GCN-Jaccard effectively defense against the $L^1$D-RND baseline attack, which consistently ranks lowest (average rank of 7.00 in both evasion and poison scenarios). In contrast, Nettack and GOttack emerge as the most effective attacks, with average ranks of 1.96 and 2.40, respectively.

GNNGuard demonstrates exceptional robustness against all adversarial attacks, with average misclassification rate of $5.46\%$, $9.31\%$, $12.22\%$, $15.19\%$, and $17.61\%$ from budget 1 through 5, respectively.

Notably, attack effectiveness varies significantly across datasets. In particular, SGA performs best in 6 out of 10 tasks on the CORA dataset, while Nettack dominates on CITESEER in 9 out of 10 tasks), and PR-BCD (NA) leads on PUBMED in 8 out of 10 tasks. These observations further emphasize the importance of evaluating adversarial methods across varied datasets and defense models.

On GRAND, the average attack performance is 20.33% and 40.92% for budgets 1 and 5, respectively. Our simple baseline attack $L^1$D-RND achieves superior performance across all homophily datasets on GRAND, with an average rank of 1.13 and best results in 26 out of 30 tasks, outperforming even gradient-based attack methods and demonstrating significantly higher computational efficiency. This raises concerns regarding the effectiveness on defense models, especially GRAND, of existing attack methods against simple baselines. RobustGCN exhibits the weakest robustness among all defenses, with average adversarial performances of 28.76% for budget 1 and 55.57% on budget 5. Nettack again proves most effective on this model, with an average rank of 2.16 and top performance in 11 out of 30 tasks.

Among recent defense models, GCORN shows average adversarial performance of 21.10% for budget 1 and 34.88% for budget 5, while RUNG achieves 18.75% and 33.21%, respectively, ranking as the second most robust model. On both GCORN and RUNG, Nettack remains the top-performing attack (average rank 1.90), followed surprisingly by $L^1$D-RND.

In summary, Nettack consistently ranks as the most effective adversarial attack across 8 defense models on homophily datasets, with an overall average rank of 2.09. GOttack follows as the second-best method with an average rank of 3.83. Notably, our $L^1$D-RND performs on bar or even surpasses gradient-based methods on GCN-GARNET, GCORN, RUNG, and especially GRAND.

**Heterophily datasets.** Tables 24 and 26 present the success rates of seven adversarial attacks against six defense models in two heterophilic datasets, in both the evasion and poisoning attack settings. In the setting of evasion attack, the FGA demonstrates the best performance, achieving success in 12 and 11 of the 30 tasks on the SQUIRREL and CHAMELEON datasets, respectively. Similarly, in the context of the poisoning attack, FGA continues to perform well, securing 17 and 14 of the tasks 30 in the datasets SQUIRREL and CHAMELEON, respectively. However, the best-performing model, FGA, shows a significant performance gain in the SQUIRREL dataset, with a 62.61% relative increase in the success rate that increases from 18.17% under evasion to 29.55% under poisoning attacks in all defense models. Likewise, for a budget of $\Delta = 1$, FGA achieves a relative improvement 48.16% on the data set CHAMELEON, increasing its average attack success rate from 17.75% in the evasion setting to 26.04% in the poisoning setting. Interestingly, the proposed naive approach $L^1$D-RND achieves the second-best results with 9 out of 30 successful attacks on both datasets in the evasion setting. In the poisoning setting, the proposed model $L^1$D-RND secures 5 and 11 of the 30 tasks in the SQUIRREL and CHAMELEON datasets, respectively. On the other hand, SGA achieves moderate performance, scoring 8 and 6 in the evasion setting on the SQUIRREL dataset, and 7 and 1 on the CHAMELEON dataset in the evasion and poisoning settings, respectively. Notably, similar to their performance on vanilla models, Nettack, GOttack, and PGD consistently exhibit lower success rates across both heterophilic datasets. It is important to note that, in the CHAMELEON dataset under the GNNGuard defense, all attack models, including the best performing model, FGA, produced surprising results. The misclassification rate remained at zero, even as the budget increased, with no observable changes. This occurred because GNNGuard pruned all the edges in this dataset.

We also present the misclassification rates of defense and vanilla models under different adversarial attacks across three homophily and two heterophily datasets, as shown in Figure 5 and Figure 6 for the evasion and poisoning settings, respectively.

**Understanding defense success and failure across graph structures.** Tables 7 and 8 provide deeper insight into why certain defenses succeed or fail under different graph structures. Similarity-based pruning methods such as GCN-Jaccard and GNNGuard collapse under heterophily because neighboring nodes are inherently dissimilar. For instance, on CHAMELEON, all attack models produce a misclassification rate of 0.00 under GNNGuard as the attack budget increases, since GNNGuard prunes nearly all edges in heterophilous graphs (refer Table 24). Training-based defenses that rely on feature smoothness, such as GRAND, also deteriorate sharply under heterophily. In the evasion setting, GRAND drops from 19.05% on homophily graphs to 3.33% on heterophily graphs, a 471.40% relative decrease. In the poisoning setting, it falls from 21.63% to 7.17%, corresponding to a 201.56% relative decrease, revealing its sensitivity to structure–feature mismatch. Defenses like

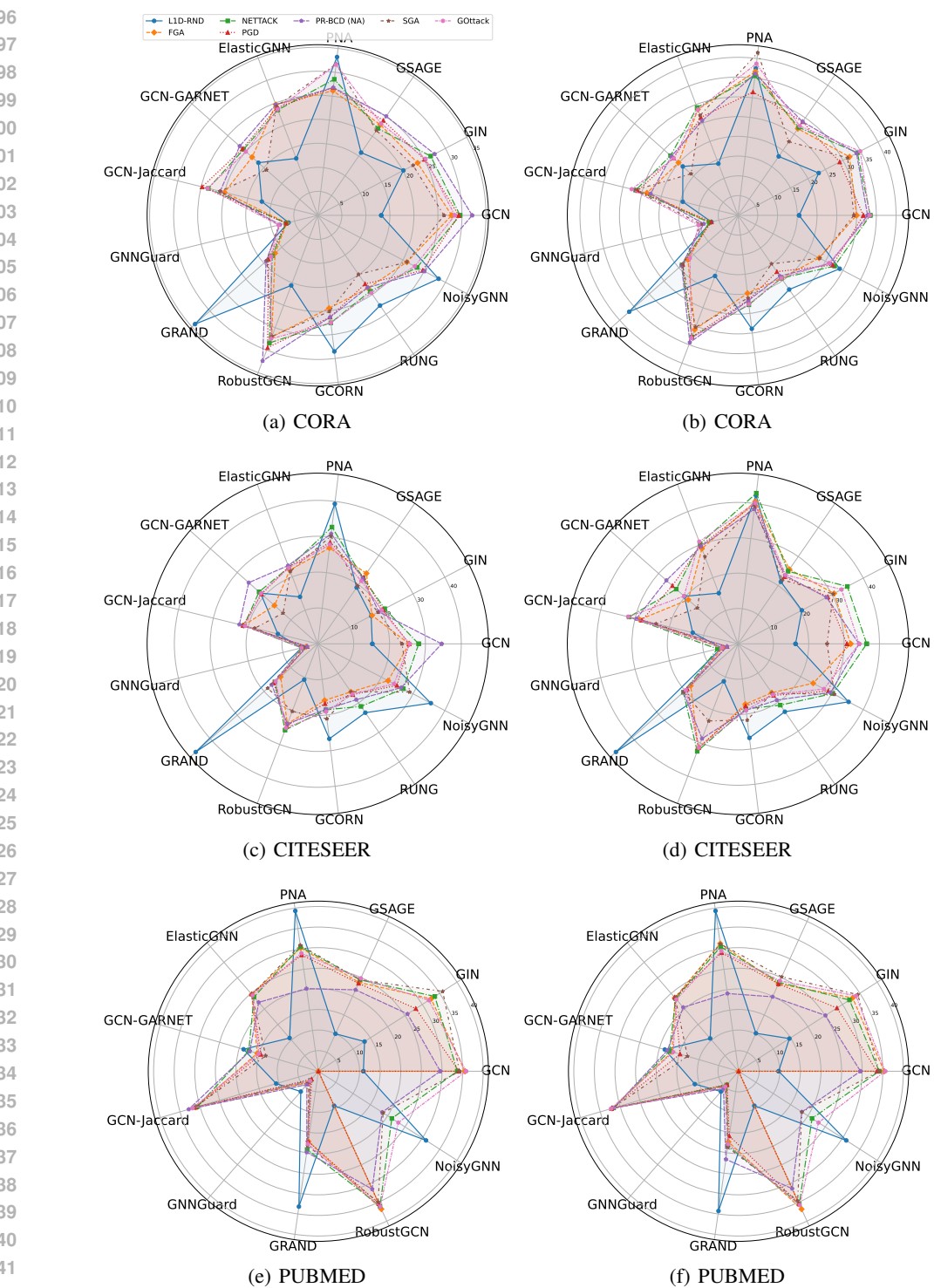

Figure 5: Misclassification rates of defense and non-defense models under different adversarial attacks (budget 1) across three homophily datasets. Left column: evasion setting; Right column: poison setting.

RUNG and GCORN also become unstable when homophily is low. RUNG decreased from 18.63% to 1.47% in evasion (1169.72% decline) and from 18.88% to 12.98% in poisoning (45.49% decrease). GCORN also drops from 20.88% to 16.70% in evasion (25.03% decline) and 21.32% to 18.20% in

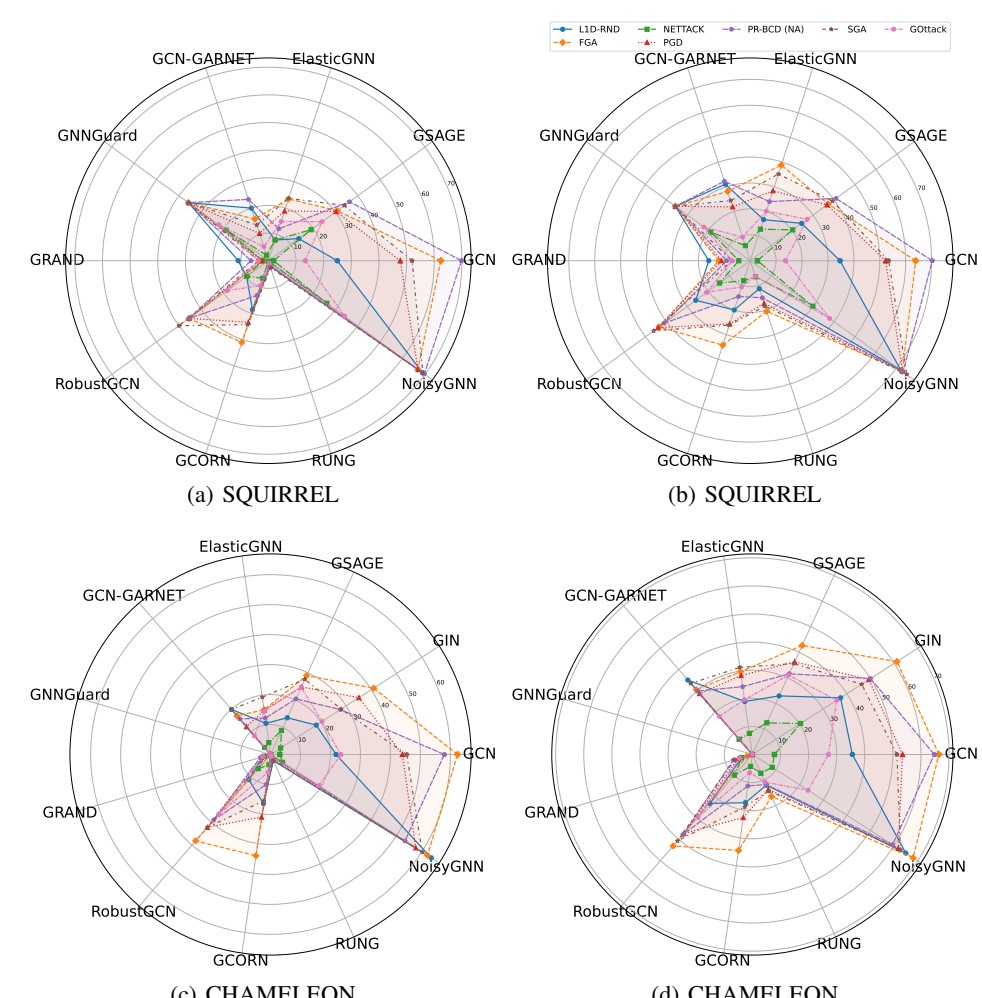

(a) SQUIRREL  (b) SQUIRREL

(c) CHAMELEON  (d) CHAMELEON

Figure 6: Misclassification rates of defense and non-defense models under different adversarial attacks (budget 1) across two heterophily datasets and one large-scale dataset. Left column: evasion setting; Right column: poison setting.

poisoning (17.16% decrease), highlighting their struggles to generalize beyond homophilous settings. In contrast, defenses that make fewer homophily assumptions such as RobustGCN, ElasticGNN, and to some extent GARNET exhibit more stable behavior across diverse datasets. In the poisoning setting, GARNET improves from 19.47% to 24.02%, yielding an 18.96% gain. RobustGCN remains steady, shifting only slightly from 26.71% to 25.66% in evasion and increasing from 29.52% to 32.65% in poisoning (9.59% gain). Meanwhile, ElasticGNN, although experiencing some degradation from 21.99% to 14.27% in evasion and 24.71% to 24.02% in poisoning (2.88% decline) still performs more reliably than heavily homophily-dependent defenses. These patterns collectively confirm that defenses grounded in rigid smoothing or homophily assumptions tend to fail on heterophily graphs, whereas structurally flexible defenses remain more resilient.

## F.3 COMPUTATIONAL COSTS OF ADVERSARIAL ATTACK

Although Nettack delivers the best performance, it is significantly more computationally expensive than GOttack, SGA, and PR-BCD (NA). Table 11 shows that GOttack is the fastest among gradient-based adversarial attack methods on smaller datasets, such as CORA and CITESEER, while SGA outperforms others on the PUBMED dataset due to its focus on the local information of target nodes. While PR-BCD (NA) is not the fastest method on the datasets studied, its time complexity is

Table 7: Selected defenses and their average success rates (evasion setting) under budget 1 across five datasets. GCN-Jaccard excluded for SQUIRREL and CHAMELEON because of an implementation error. Higher is better. Best performance in **Bold**, second best is underline.

| Taxonomy | | | Selected Defense | CORA | CITESEER | PUBMED | SQUIRREL | CHAMELEON |
|---|---|---|---|---|---|---|---|---|
| Improving graph | Unsupervised | | Jaccard-GCN* | 20.81 ± 3.94 | 19.58 ± 3.53 | 28.49 ± 7.30 | – | – |
| | Supervised | | GARNET | 18.91 ± 2.50 | 20.01 ± 3.87 | 16.03 ± 1.82 | 12.95 ± 7.05 | 13.73 ± 5.80 |
| Improving training | | | GRAND | 16.26 ± 7.33 | 20.47 ± 10.29 | 20.40 ± 5.30 | 4.38 ± 3.00 | 2.28 ± 1.03 |
| | | | GCORN | 22.26 ± 2.76 | 19.48 ± 3.30 | OOR | 18.11 ± 8.14 | 15.27 ± 9.58 |
| | | | NoisyGNN | 23.73 ± 2.42 | **26.97 ± 4.05** | 16.05 ± 10.88 | **57.38 ± 17.40** | **45.86 ± 21.98** |
| Improving architecture | Adaptively weighting edges | Rule-based | GNNGuard | 7.358 ± 0.84 | 3.9 ± 0.70 | 3.77 ± 1.23 | 31.36 ± 6.95 | 0.00 ± 0.00 |
| | | Probabilistic | RobustGCN | **26.76 ± 4.88** | 21.96 ± 4.89 | **31.40 ± 9.20** | 26.30 ± 12.28 | 25.00 ± 10.93 |
| | | Robust agg. | ElasticGNN | 22.47 ± 4.02 | 21.33 ± 3.04 | 22.17 ± 4.77 | 15.56 ± 6.22 | 12.97 ± 4.44 |
| | | | RUNG | 18.64 ± 2.21 | 18.61 ± 2.51 | OOR | 1.65 ± 0.74 | 1.27 ± 0.73 |

Table 8: Selected defenses and their average success rates (poison setting) under budget 1 across five datasets. GCN-Jaccard excluded for SQUIRREL and CHAMELEON because of an implementation error. Higher is better. Best performance in **Bold**, second best is underline.

| Taxonomy | | | Selected Defense | CORA | CITESEER | PUBMED | SQUIRREL | CHAMELEON |
|---|---|---|---|---|---|---|---|---|
| Improving graph | Unsupervised | | Jaccard-GCN* | 24.03 ± 4.26 | 26.72 ± 6.06 | 28.95 ± 7.30 | – | – |
| | Supervised | | GARNET | 20.51 ± 2.32 | 21.51 ± 3.65 | 16.45 ± 1.83 | 21.92 ± 9.53 | 26.11 ± 9.24 |
| Improving training | | | GRAND | 20.78 ± 6.61 | 23.54 ± 9.26 | 20.55 ± 5.82 | 9.52 ± 3.59 | 4.81 ± 1.58 |
| | | | GCORN | 22.66 ± 2.71 | 19.97 ± 3.08 | OOR | 19.90 ± 8.70 | 16.49 ± 9.59 |
| | | | NoisyGNN | 26.07 ± 2.00 | **29.46 ± 3.18** | 15.98 ± 10.86 | **61.77 ± 17.79** | **49.80 ± 21.90** |
| Improving architecture | Adaptively weighting edges | Rule-based | GNNGuard | 8.22 ± 1.25 | 4.45 ± 0.91 | 5.06 ± 0.71 | 31.36 ± 6.95 | 0.00 ± 0.00 |
| | | Probabilistic | RobustGCN | **30.15 ± 5.76** | 27.19 ± 7.10 | **31.19 ± 9.10** | 33.71 ± 11.92 | 31.58 ± 10.94 |
| | | Robust agg. | ElasticGNN | 25.67 ± 4.91 | 27.27 ± 5.05 | 21.18 ± 4.46 | 25.14 ± 8.85 | 22.89 ± 7.64 |
| | | | RUNG | 18.81 ± 2.20 | 18.95 ± 2.27 | OOR | 13.63 ± 5.29 | 12.32 ± 2.67 |

independent of both the number of nodes and edges, making it advantageous for large-scale datasets like OGB-arxiv (Hu et al., 2020b). In any case, the **reliance on results from only three datasets, as is common among the studied models, presents a significant limitation and raises concerns about the robustness and generalizability of the field's findings**. The authors are not aware of any other machine learning domain where the majority of results rely on such a limited number of datasets.

**Time complexity breakdown.** To better understand the computational cost of different adversarial attack methods, we break attack algorithms into three main stages according to their implementations provided by DeepRobustLi et al. (2020b): **Surrogate**, **Pre-attack** and **Attacks**. **Surrogate** stage includes the time required to train and set up the surrogate used in the attack. **Pre-attack** involves operations: computing logits of target nodes and normalizing the adjacency matrix in NETTACK and GOttack; performing project gradient descent training and projected randomized block coordinate descent in PGD and PR-BCD, respectively; retrieving subgraphs in SGA. Finally, in **Attacks** phases, attack algorithms find the best edges to flip according to surrogate loss of all potential edges in NETTACK and selected potential edges in GOttack, while PR-BCD and PGD perform a Bernoulli sample to select the optimal edge to flip.

Table 11 shows the time complexity breakdown of three main stages of adversarial attacks for a budget of 1 across three datasets (CORA, PUBMED, and OGB-ARXIV). Across all datasets, $L^1$D-RND is the fastest method, achieving close to zero seconds in all three stages. Its efficiency is due to its simple random edge selection strategy, which avoids surrogate training and iterative optimization entirely. Benefiting from fast gradient-based updates, FGA spends significantly less time on **Pre-attack** and **Attack** stages on CORA and PUBMED datasets and achieves the second fastest on CORA and PUBMED. However, FGA need to maintain a dense adjacency matrix to compute the gradient, which requires more than 100GiB on OGB-ARXIV, causing Out-Of-Memory (OOM). On CORA, PUBMED and OGB-ARXIV, SGA is established to be one of the most efficient attacks, as it limits the pre-computation to small subgraphs, resulting in minimal in **Pre-attak** and **Attack** stages. In contrast, PGD's runtime on CORA and PUBMED is dominated by project gradient descent training

Table 9: Average total attack time on 50 target nodes (in Seconds). Smaller is better. Best time in **Bold**, second best is underline.

| | | $\Delta \rightarrow$ | 1 | 2 | 3 | 4 | 5 |
|---|---|---|---|---|---|---|---|
| CORA | Global | L$^1$D-RND | **0.17 ± 0.08** | **0.29 ± 0.06** | **0.49 ± 0.11** | **0.65 ± 0.10** | **0.92 ± 0.13** |
| | | FGA | 0.36 ± 0.03 | 0.67 ± 0.06 | 0.98 ± 0.10 | 1.30 ± 0.12 | 1.57 ± 0.17 |
| | | NETTACK | 19.72 ± 1.46 | 29.75 ± 1.00 | 41.01 ± 2.11 | 50.90 ± 2.51 | 62.80 ± 2.06 |
| | | PGD (NA) | 110.86 ± 7.63 | 104.17 ± 8.61 | 95.58 ± 10.93 | 95.83 ± 5.90 | 95.09 ± 14.17 |
| | | PR-BCD (NA) | 159.17 ± 26.05 | 158.33 ± 19.49 | 163.96 ± 20.16 | 164.16 ± 17.39 | 163.26 ± 21.55 |
| | | GOttack | 8.10 ± 1.03 | 11.09 ± 0.28 | 14.57 ± 0.32 | 17.73 ± 0.55 | 21.43 ± 0.83 |
| | Local | SGA | 16.90 ± 0.37 | 17.73 ± 0.43 | 18.44 ± 0.39 | 18.99 ± 0.46 | 19.23 ± 0.62 |
| CITESEER | Global | L$^1$D-RND | **0.15 ± 0.04** | **0.29 ± 0.12** | **0.36 ± 0.09** | **0.37 ± 0.09** | **0.42 ± 0.07** |
| | | FGA | 0.25 ± 0.04 | 0.36 ± 0.08 | 0.50 ± 0.06 | 0.56 ± 0.03 | 0.58 ± 0.04 |
| | | NETTACK | 22.84 ± 1.37 | 29.27 ± 0.91 | 35.83 ± 1.24 | 42.80 ± 1.27 | 49.48 ± 1.62 |
| | | PGD (NA) | 134.79 ± 6.95 | 129.90 ± 8.68 | 126.59 ± 8.77 | 127.51 ± 9.37 | 121.21 ± 10.59 |
| | | PR-BCD (NA) | 209.56 ± 40.58 | 215.76 ± 45.17 | 224.09 ± 43.40 | 199.46 ± 36.82 | 195.49 ± 34.81 |
| | | GOttack | 6.39 ± 1.08 | 8.18 ± 0.17 | 10.41 ± 0.23 | 12.50 ± 0.34 | 14.66 ± 0.36 |
| | Local | SGA | 17.39 ± 0.56 | 17.97 ± 0.49 | 19.02 ± 0.52 | 19.13 ± 0.60 | 19.46 ± 0.60 |
| PUBMED | Global | L$^1$D-RND | **1.81 ± 0.54** | **1.72 ± 0.42** | **1.77 ± 0.44** | **1.75 ± 0.37** | **1.64 ± 0.45** |
| | | FGA | 13.68 ± 0.29 | 27.72 ± 0.48 | 39.25 ± 3.37 | 47.43 ± 0.70 | 59.28 ± 0.87 |
| | | NETTACK | 423.84 ± 18.17 | 765.05 ± 29.30 | 1095.95 ± 31.13 | 1426.95 ± 42.03 | 1753.58 ± 67.42 |
| | | PGD (NA) | 4114.65 ± 182.27 | 3922.62 ± 159.91 | 3814.07 ± 112.16 | 3721.93 ± 108.08 | 3689.32 ± 103.25 |
| | | PR-BCD (NA) | 173.83 ± 6.62 | 177.46 ± 7.42 | 173.28 ± 3.54 | 171.80 ± 8.22 | 173.10 ± 7.56 |
| | | GOttack | 153.77 ± 3.32 | 252.04 ± 6.47 | 355.97 ± 10.38 | 452.34 ± 10.73 | 553.61 ± 13.92 |
| | Local | SGA | 18.09 ± 0.40 | 17.71 ± 0.63 | 19.00 ± 0.68 | 19.85 ± 0.93 | 19.66 ± 0.99 |

Table 10: GPU memory usage in MB of attacks on budget 1. Smaller is better. Best GPU memory usage is in **bold**, and the second best is in underline.

| Method | CHAMELEON | CORA | CITESEER | SQUIRREL | PUBMED |
|---|---|---|---|---|---|
| SGA | 90.824 | 105.619 | 92.143 | 92.312 | 92.537 |
| NETTACK | 65.747 | 64.909 | 66.041 | 64.258 | 62.898 |
| FGA | 335.235 | 331.022 | 413.564 | 328.964 | 328.773 |
| PGD (NA) | 421.835 | 417.622 | 499.784 | 415.564 | 415.374 |
| PRBCD (NA) | 196.014 | 195.169 | 159.254 | 131.659 | 157.704 |
| L$^1$D-RND | **39.568** | **39.553** | **38.901** | **37.825** | **38.703** |
| GOttack | 64.984 | 63.315 | 65.275 | 62.536 | 62.510 |

in **Pre-attack** stage, accounting for 96% of total runtime on PUBMED. In addition, PGD encounters the same memory constraints as FGA on OGB-ARXIV. Instead of project gradient descent, PR-BCD utilize projected randomized block coordinate descent, which has been shown to reduce the dominant effect of **Pre-attack** stage to 55% on PUBMED. Methods such as NETTACK and GOttack are significantly slower, particularly on larger datasets. **Pre-attack** and **Attack** stages dominate the runtime in NETTACK due to performing optimization to select the best edge to flip over all potential edges. GOttack's **Pre-attack** and **Attack** stages' runtime has shown improvements over NETTACK due to reducing the search space by wisely filtering potential edges.

Table 10 reports the GPU memory usage (in MB) of different attack methods under budget 1 across five benchmark datasets. As expected, gradient-based methods such as PGD (NA) and FGA consume substantially more memory, with peak usage above 400 MB on CiteSeer and PubMed. PRBCD (NA) reduces the cost relative to PGD but still requires more than 130 MB across datasets. In contrast, traditional structure-based methods like NETTACK and GOttack are much lighter, staying around 60–66 MB, with GOttack consistently achieving the second lowest footprint. The best efficiency is obtained by L$^1$D-RND, which uses less than 40 MB on all datasets. These results confirm that L$^1$D-RND provides the most memory-efficient attack, while GOttack is a competitive second choice.

## F.4 COMPUTATIONAL COSTS OF DEFENSE

Table 12 reports the average total training time (in seconds) for various defense models across five benchmark datasets. Overall, NoisyGNN consistently achieves the fastest training times on most datasets, with particularly low values on CORA, PUBMED, and SQUIRREL. Its efficiency can be attributed to its lightweight architecture and minimal computational overhead, which allows it to scale well even on larger graphs. The second fastest model is generally RobustGCN, which

Table 11: Time break down of adversarial attacks on budget 1 (in Seconds). Smaller is better. Best time in **Bold**, second best is underline.

| | Method | Surrogate | Pre-attack | Attack |
|---|---|---|---|---|
| **CORA** | $L^1$D-RND | **0.00000 ± 0.00000** | **0.00006 ± 0.00000** | **0.00213 ± 0.00245** |
| | FGA | 0.73066 ± 0.00001 | 0.00011 ± 0.00000 | 0.00531 ± 0.00013 |
| | NETTACK | 1.23075 ± 0.00011 | 0.16365 ± 0.06042 | 0.20430 ± 0.14573 |
| | PGD (NA) | 0.98343 ± 0.00003 | 3.69364 ± 0.82624 | 0.04686 ± 0.00793 |
| | PRBCD (NA) | 1.43114 ± 0.00000 | 4.31843 ± 0.16030 | 0.06399 ± 0.01213 |
| | SGA | 1.62636 ± 0.00005 | 0.32616 ± 0.05827 | 0.00323 ± 0.00016 |
| | GOttack | 1.16423 ± 0.00007 | 0.08035 ± 0.05181 | 0.07113 ± 0.04261 |
| **PUBMED** | $L^1$D-RND | **0.00000 ± 0.00000** | **0.00007 ± 0.00000** | **0.00387 ± 0.00429** |
| | FGA | 1.86278 ± 0.00002 | 0.00045 ± 0.00022 | 0.27881 ± 0.00710 |
| | NETTACK | 4.45833 ± 0.00026 | 1.43834 ± 0.04844 | 6.53812 ± 5.08244 |
| | PGD (NA) | 2.75710 ± 0.00006 | 155.74790 ± 32.54714 | 2.74592 ± 0.33127 |
| | PRBCD (NA) | 2.57097 ± 0.00000 | 3.29684 ± 0.42744 | 0.06547 ± 0.01111 |
| | SGA | 10.47330 ± 0.00007 | 0.33145 ± 0.05469 | 0.00329 ± 0.00013 |
| | GOttack | 1.91390 ± 0.00005 | 0.91947 ± 0.05263 | 2.03039 ± 1.37443 |
| **OGB-ARXIV** | $L^1$D-RND | **0.00000 ± 0.00000** | **0.00010 ± 0.00001** | **0.13041 ± 0.23359** |
| | FGA | OOM | OOM | OOM |
| | NETTACK | 12.54029 ± 0.28053 | 362.42469 ± 39.61532 | 894.00637 ± 8.04044 |
| | PGD (NA) | OOM | OOM | OOM |
| | PRBCD (NA) | 7.86793 ± 0.00000 | 12.93282 ± 0.14348 | 0.34241 ± 0.01064 |
| | SGA | 3.85193 ± 0.00870 | 0.50283 ± 0.20965 | 0.01772 ± 0.01850 |
| | GOttack | 12.80845 ± 0.87578 | 326.28377 ± 27.37981 | 288.88327 ± 7.75647 |

Table 12: Average total time to train defense models (in Seconds). Smaller is better. Best time in **Bold**, second best is underline.

| Defense | CORA | CITESEER | PUBMED | SQUIRREL | CHAMELEON |
|---|---|---|---|---|---|
| ElasticGNN | 5.03 ± 2.78 | 3.92 ± 2.03 | 8.57 ± 5.27 | 10.67 ± 2.79 | 9.72 ± 5.05 |
| GCN-GARNET | 2.56 ± 2.51 | 3.87 ± 3.66 | 25.81 ± 9.39 | 3.45 ± 2.14 | 3.79 ± 1.56 |
| GCN-Jaccard | 1.24 ± 0.68 | 1.58 ± 0.79 | - | - | - |
| GNNGuard | 1.91 ± 0.47 | 1.62 ± 0.59 | 4.39 ± 0.84 | 5.42 ± 0.63 | 2.37 ± 0.54 |
| GRAND | 6.46 ± 3.00 | 7.50 ± 2.39 | 93.48 ± 40.46 | 28.94 ± 23.13 | 5.42 ± 6.92 |
| RobustGCN | 3.95 ± 2.99 | **0.87 ± 0.12** | 5.54 ± 1.67 | 2.24 ± 0.56 | **1.09 ± 0.22** |
| GCORN | 54.99 ± 8.24 | 29.04 ± 7.89 | 4605.11 ± 727.34 | 130.53 ± 134.07 | 15.40 ± 5.34 |
| RUNG | 130.52 ± 5.34 | 106.62 ± 10.01 | 4265.07 ± 2279.45 | 46.06 ± 20.67 | 7.69 ± 2.78 |
| NoisyGNN | **0.77 ± 0.28** | 1.29 ± 0.49 | **2.13 ± 0.50** | **1.78 ± 1.35** | 1.70 ± 0.68 |

performs competitively on CITESEER, PUBMED, and CHAMELEON, likely due to its optimized graph convolution operations that reduce redundant computations. In contrast, RUNG is the slowest model by a wide margin, especially on large datasets such as PUBMED and CITESEER. This significant slowdown is likely caused by its complex training procedure and extensive use of robust aggregation mechanisms, which introduce high computational cost. Other models such as GCORN and GRAND also show high training times on large graphs, highlighting the trade-off between defense sophistication and training efficiency.

# G DEFENSE MODELS

**Selection criteria.** In addition to the vanilla attack scenario, we also consider evaluating adversarial attack methods against defense models (defense attack scenario). However, to avoid the significant number of experiences needed to evaluate each adversarial attack to all defense models existing in the literature, we select defense models used to evaluate adversarial attacks, we first select six defense models, one from each taxonomy described in Table 13 to cover the entire spectrum of defense techniques. In this work, we select defense models according to the following criteria: i) highly cited defenses published at renowned venues; ii) publicly available codes, recent; iii) recently published. Following Mujkanovic et al.'s work, we exclude defense models from "Robust training" taxonomy from our study, since they require knowing clean graph $\mathcal{G}$, which is typically not available in the poison attack setting.

Table 13: Categorization of selected defenses, adopted and and extended from Günnemann (2022); Mujkanovic et al. (2022).

| | Taxonomy | Selected Defenses | Other Defenses |
|---|---|---|---|
| Improving graph | Unsupervised | Jaccard-GCN (Wu et al., 2019b) | (Entezari et al., 2020; Duan et al., 2020) (Xiao et al., 2021; Zhang et al., 2019; 2021) |
| | Supervised | GARNET (Deng et al., 2022) | (Jin et al., 2020; Xu et al., 2021; Tao et al., 2021) (Zhang et al., 2022; Jin et al., 2023) |
| Improving training | | GRAND (Feng et al., 2020) GCORN (Abbahaddou et al., 2024) NoisyGNN (Ennadir et al., 2024a) | (Feng et al., 2020; Elinas et al., 2020; Jin et al., 2019) (You et al., 2020; Zheng et al., 2020; Zhuang & Hasan, 2022) (Chen et al., 2021a; Deng et al., 2023; Feng et al., 2021b) (Hu et al., 2021; Xu et al., 2022; 2019a) |
| Improving architecture | Adaptively weighting edges — Rule-based | GNNGuard (Zhang & Zitnik, 2020) | (Jin et al., 2021; Liu et al., 2021a; Zhang & Lu, 2020) |
| | Probabilistic | RobustGCN (Zhu et al., 2019) | (Chen et al., 2020; Ioannidis & Giannakis, 2019; Feng et al., 2021a) Ioannidis et al. (2020); Luo et al. (2021) |
| | Robust agg. | ElasticGNN (Liu et al., 2021b) RUNG (Hou et al., 2024) | (Chen et al., 2021b; Geisler et al., 2021) |

**Jaccard-GCN.** Prior to train GCN, adjacency matrix is first preprocessed by pruning edges of pairs of nodes that have Jaccard coefficient of the binarized features $J_{ij} = \frac{\mathbf{X}_i \mathbf{X}_j}{min\{\mathbf{X}_i + \mathbf{X}_j, 1\}}$ exceed a threshold $\epsilon$.

**GARNET.** Similar to Jaccard-GCN, GARNET is a spectral method that preprocesses data to enhance the robustness of GNN models. GARNET first leverages weighted spectral embedding to construct a base graph that utilizes the top few dominant singular components of $\mathbf{A}$ to restore its important graph spectrum. This is not only robust to adversarial attacks but also preserves important structural information for GNNs training. Secondly, GARNEET refines the base graph by pruning uncritical edges according to a probabilistic graphical model.

**GRAND.** GRAND begins by introducing a random propagation strategy to augment the graph data. Following that, GRAND leverages consistency regularization to optimize the prediction consistency of unlabeled nodes across different data augmentations.

**GCORN.** A theoretical concept of expected robustness in the context of attributed graphs and an upper bound of the expected robustness are first defined by Abbahaddou et al.. Leveraging the established upper bound on the expected robustness, GCORN utilizes the orthonormalization of weight matrices to control the robustness of GCNs against adversarial attacks.

**GNNGuard.** The robustness of GNNGUARD is attributed to two novel components: the neighbor importance estimation and the layer-wise graph memory. GNNGUARD estimates an importance weight for every edge $e_{uv}$ to quantify how relevant node $u$ is to another node $v$ according to the hypothesis that similar nodes are more likely to interact than dissimilar nodes. Updating edges with low importance weights corresponds to pruning incritical edges, since those edges will likely be ignored by the GNN. However, neighbor importance estimation and edge pruning change the graph structure between adjacent GNN layers, which could potentially destabilize GNN training. To allow for robust estimation of importance weights and smooth evolution of edge pruning, GRAND utilizes layer-wise graph memory at each GNN layer, keeping partial memory of the pruned graph structure from the previous layer.

**RobustGCN.** To improve GCN robustness, instead of representing nodes as vectors, RobustGCN utilizes Gaussian distributions as the hidden representations to absorb perturbations. Moreover, it uses a variance-based attention mechanism that assigns weights to the node neighbourhoods according to their variances when performing convolutions.

**ElasticGNN.** ElasticGNN utilizes a novel and general message passing scheme that works based on $\mathcal{L}_1$ and $\mathcal{L}_2$-based graph smoothing. This message passing algorithm is not only friendly to back-propagation training but also achieves the desired smoothing properties with a theoretical convergence guarantee. Locally adaptive smoothness makes Elastic GNNs more robust to adversarial attacks on graph structure. This is because the attack tends to connect nodes with different labels, which fuzzes the cluster structure in the graph. But LeasticGNN can tolerate large node differences along these wrong edges, and maintain the smoothness along correct edges.

**RUNG.** RUNG improves robustness against adversarial attacks by introducing a robust and unbiased graph signal estimator that mitigates the estimation bias present in traditional $\mathcal{L}_1$-based methods. It employs the Minimax Concave Penalty (MCP) to downweight or prune suspicious edges with large feature differences, reducing the impact of adversarial perturbations. The Quasi-Newton Iteratively

Reweighted Least Squares (QN-IRLS) algorithm efficiently solves the non-convex optimization problem, ensuring stable convergence and enabling the model to maintain clean accuracy while defending against attacks.

# H    IMPORTANCE OF NODE DEGREE AS CRITERIA TO SELECT TARGET NODES AND MODEL SELECTION IN ADVERSARIAL ATTACK EVALUATION

In this section, we highlight the importance of node degree and model selection in evaluating adversarial attacks. Table 30 provides a comprehensive comparison of the differences in performance of adversarial attacks when evaluating with and without model selection, while Table 29 shows complete differences caused by considering degree as target node selection. A high level summary of Table 30 and Table 29 is provided by Figure 7. In general, high-degree nodes are more challenging to attack by existing adversarial methods, resulting in significant increases in the performance of all adversarial attacks when we exclude high-degree nodes in evaluation, especially on the PUBMED dataset (Figure 7(c)). In addition, Figure 7 also indicates that evaluating adversarial attacks without performing model selections on victim and surrogate models, which is typically done in practice, results in differences in adversarial attack performance.

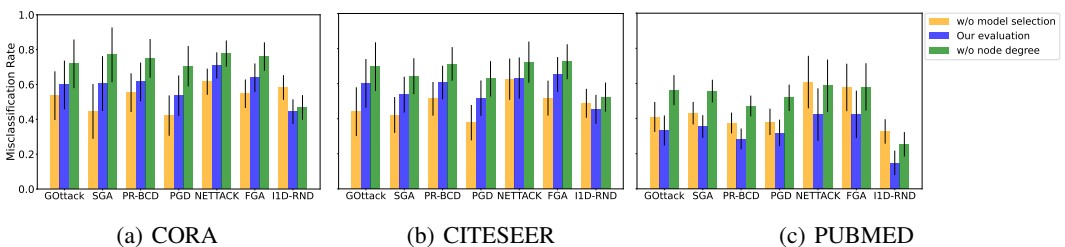

|     (a) CORA     |     (b) CITESEER     |     (c) PUBMED     |

Figure 7: Performance of seven adversarial attacks on GSAGE on CORA datasets in poison setting. The figures show the comparison of adversarial attacks' performance obtained from our proposed evaluation procedure, our evaluation without model selection and our evaluation without node degree as a criteria to select target nodes.

Table 14 highlights a clear dependence of adversarial vulnerability on node degree. For both Cora and Citeseer, nodes with lower degrees exhibit markedly higher average misclassification rates under the top three attacks, confirming that sparsely connected nodes are the most susceptible to perturbations. In Cora, degree-1 nodes reach nearly 0.9 average misclassification, while nodes of degree 14 or 15 show negligible rates. A similar pattern holds for Citeseer, where the rate drops from 0.84 at degree 1 to almost zero by degree 14. Interestingly, intermediate degrees (such as 10–12) show fluctuations rather than a strictly monotonic decline, suggesting that local structural context and model bias interact with degree in nontrivial ways. Overall, these results reinforce the intuition that robustness is strongly linked to connectivity, with low-degree nodes remaining a persistent weak point across models and datasets.

**Model Selection.** Model selection is an important step performed prior to deploying GNNs in real-world application. Table 30 indicates that failing to perform model selection in adversarial attacks could prevent adversarial GNN practitioners from exploring the impact of them in practice. The difference between evaluating adversarial attacks with and without model selections ranges from $-16.40\%$ to $19.07\%$ with $5.19\%$ average absolute difference across adversarial attacks. Performing model selection in evaluation could potentially change the average rank of adversarial attacks. Especially in the setting of GIN and GSAGE, in both evasion and poison settings, the performance of $L^1$D-RND attacks significantly drops by $9.15\%$ on average; thus increasing the average rank of $L^1$D-RND by $2.74$ on average.

**Node degree.** Table 29 shows that considering node degree as a target node selection criterion instead of randomly selecting results in a significant drop in adversarial attack performance in most of the cases, with $9.80\%$ average decrease. Especially in the setting of GCN on PUBMED dataset on budget 3 in both settings, the difference in PR-BCD (NA) performance between evaluating with

Table 14: Top 15 node degrees and their average misclassification rates on vanilla models (GCN, GIN, GraphSAGE, PNA) under top-3 adversarial attacks (Nettack, FGA, PRBCD) in the evasion setting.

| Node Degree | Cora (Avg. Misclassification %) | Citeseer (Avg. Misclassification %) |
|:---:|:---:|:---:|
| 1 | 0.88±0.13 | 0.84±0.13 |
| 2 | 0.84±0.15 | 0.83±0.09 |
| 3 | 0.79±0.12 | 0.71±0.14 |
| 4 | 0.65±0.20 | 0.60±0.08 |
| 5 | 0.61±0.16 | 0.47±0.15 |
| 6 | 0.58±0.19 | 0.51±0.11 |
| 7 | 0.50±0.28 | 0.31±0.14 |
| 8 | 0.50±0.14 | 0.26±0.18 |
| 9 | 0.29±0.27 | 0.12±0.14 |
| 10 | 0.31±0.29 | 0.35±0.30 |
| 11 | 0.42±0.47 | 0.19±0.30 |
| 12 | 0.47±0.41 | 0.19±0.21 |
| 13 | 0.33±0.44 | 0.08±0.29 |
| 14 | 0.06±0.16 | 0.00±0.00 |
| 15 | 0.18±0.24 | 0.01±0.04 |

and without degree as target node selection criteria is notably significant, $26.3\%$ in evasion and $25.97\%$. Furthermore, the difference of adversarial attack performances in the PUBMED dataset is $-14.969\%$ on average, while that of those in CORA and CITESEER is $-6.93\%$ and $-7.51\%$ on average, respectively.

# I  $L^1$D-RND: A SIMPLE YET EFFECTIVE BASELINE FOR ADVERSARIAL ATTACKS

This section provides further information about the naive baseline $L^1$D-RND. The algorithm and intuition behind $L^1$D-RND are discussed in Section I, while Section I highlights the improvement of $L^1$D-RND over random attack (RND).

**Attack strategy and motivation.** $L^1$D-RND attack follow the strategy described in Algorithm 3. Given a target node $v_t$, the adjacency matrix $\mathbf{A}$, and node features $\mathbf{X}$, $L^1$D-RND randomly decides to add or to remove an edge (Line 7). In the case of adding an edge, $L^1$D-RND first randomly sampled a set of vertices $\mathcal{V}'_{\text{candidate}}$ that are not neighbours of the target node $v_t$ (Line 10). $L^1$D-RND then adds an edge between the target node $v_t$ and $u$, where $u$ is the node with the highest degree in $\mathcal{V}'_{\text{candidate}}$ (Line 13). We select the highest node from the node's sample $\mathcal{V}'_{\text{candidate}}$ to avoid always selecting a node with the highest degree from the whole graph. The adversarial impact of edges added by $L^1$D-RND is attributed to two features: stochasticity and node degree. In particular, sampling a subset of nodes enables stochastic noise to be introduced to the target node $v_t$, and selecting nodes with the highest degree enables the node that can aggregate the most messages from its neighbours. In the case of removing an edge, $L^1$D-RND considers removing an edge between target node $v_t$ and its neighbours, excluding new neighbours added by $L^1$D-RND (Line 18). Similar to the case of adding an edge, $L^1$D-RND then sampled a set of vertices $\mathcal{V}'_{\text{candidate}}$ from its neighbours. Finally, $L^1$D-RND removes an edge between target node $v_t$ and $u$, where $u$ has the highest influence score that is computed in Algorithm 4. For each node, $u$, the influence score is defined by computing the $\mathcal{L}_1$ norm of $u$ and its neighbour node features. This influence score provides an approximation of the contribution of messages of node $u$ to the representation of $u$'s neighbour in the message aggregation stage.

**Improvement over random attack (RND).** This section highlights the improvement of $\mathcal{L}_1$ norm and node degree in $L^1$D-RND over random attack (RND). Table 31 shows that $L^1$D-RND outperforms RND by an average of $5.15\%$ in $84\%$ settings. Especially, $L^1$D-RND performs significantly better by an average of $13.4\%$ than RND in all settings on GRAND.

---

**Algorithm 3** $L^1$D-RND $(v_t, \Delta, \mathcal{G}, r)$

---

1: **Input:** Target node $v_t$, modified budget $\Delta$,
    Graph $\mathcal{G} = (\mathbf{A}, \mathbf{X})$, sample ratio $r$
2: **Output:** Modified graph $\mathcal{G}' = (\mathbf{A}', \mathbf{X})$
3: $\mathbf{A}' = \mathbf{A}$
4: $\mathcal{N}_{v_t}^{\text{new}} = []$
5: **while** $|\mathbf{A} - \mathbf{A}'| < \Delta$ **do**
6:    $\mathcal{N}_{v_t} = \text{neighbor}(v_t, \mathbf{A}')$
7:    $x \leftarrow \text{rand}()$ // Random number between 0 and 1
8:    **if** $x > 0.5$ **then**
9:       // Add an edge
10:      $\mathcal{V}_{\text{candidate}} = [0, 1, 2, \ldots, \text{len}(\mathbf{A})] - \mathcal{N}_{v_t}$ // Exclude neighbours nodes
11:      $k = \lfloor r \times |\mathcal{V}_{\text{candidate}}| \rfloor$
12:      $\mathcal{V}'_{\text{candidate}} \sim \text{Uniform}(\mathcal{V}_{\text{candidate}}, k)$
13:      $e^* = (v_t, u) \leftarrow \underset{u \in \mathcal{V}'_{\text{candidate}}}{\arg \max} \text{degree}(u, \mathbf{A}')$
14:      $\mathbf{A}' = \mathbf{A}' + e^*$
15:      $\mathcal{N}_{v_t}^{\text{new}}.\text{append}(u)$
16:    **else**
17:      // Remove an edge
18:      $\mathcal{N}_{v_t}^- = \mathcal{N}_{v_t} - \mathcal{N}_{v_t}^{\text{new}}$
19:      $k = \lfloor r \times |\mathcal{N}_{v_t}^-| \rfloor$
20:      $\mathcal{V}_{\text{candidate}} \sim \text{Uniform}(\mathcal{N}_{v_t}^-, k)$
21:      $e^* = (v_t, u) \leftarrow \underset{u \in \mathcal{V}_{\text{candidate}}}{\arg \max} s_{\text{influence}}(u, \mathbf{A}', \mathbf{X})$
22:      $\mathbf{A}' = \mathbf{A}' - e^*$
23:    **end if**
24: **end while**

---

**Algorithm 4** $s_{\text{influence}}(u, \mathbf{A}, \mathbf{X})$

---

1: **Input:** Node $u$, Adjacency matrix $\mathbf{A}$, Node feature $\mathbf{X}$
2: **Output:** Influence score $s_{\text{influence}}$
3: $\mathcal{N}_u = \text{neighbor}(u, \mathbf{A}) + u$
4: $s_{\text{influence}} = L_1\text{norm}(\mathbf{X}[\mathcal{N}_u, :])$

---

While many benchmark datasets use sparse, bag-of-words-like node features (e.g., Cora, Citeseer, Pubmed), we test the generality of $L^1$D-RND by evaluating it on OGB-Arxiv, a large-scale graph with continuous node embeddings derived from raw text. Even in this setting, $L^1$D-RND outperforms PR-BCD (NA) in 8 out of 10 cases on GraphSAGE and remains competitive on GCN (Tables 16–19, Support results). These results suggest that even basic perturbation strategies can be effective under practical constraints, especially when stronger attacks lose efficacy due to defense adaptations or poor transferability. We advocate including simple baselines like $L^1$D-RND in future evaluations to ensure that observed robustness is not overstated by comparisons limited only to complex attacks.

**Time cost.** Table 11 has shown superior speed of our simple $L^1$D-RND attacks compared to iterative or gradient-based methods. 1RND demonstrates remarkable efficiency in adversarial attacks, as evidenced by its consistently low and stable execution times across budgets. With execution times ranging from just $16.90 \pm 0.37$ seconds to $19.23 \pm 0.62$ seconds, $L^1$D-RND orders of magnitude faster than remaining gradient-based attacks, especially PGD.

**Factors Driving the Effectiveness of $L^1$D-RND.** Our hypothesis is that the success of $L^1$D-RND attack, especially against defended models, arises from two main factors:

First is bias toward high-degree nodes. Even though $L^1$D-RND is simple, its edge selection is not purely random. On the addition side, it connects the target node to a high-degree node (line 13 in Alg.3). On the removal side, it prunes based on an $L_1$-norm-based influence measure (Alg.4), which correlates with local feature aggregation. This introduces a structure-aware perturbation mechanism,

despite the absence of gradient-based optimization. Our ablation shows that both components, when used independently, are sufficient to create harmful structural noise.

The second factor is the mismatch between attack complexity and defense generalization. Existing defenses often assume strong adversarial priors (e.g., feature gradients, link importance). These may fail to generalize against structurally simple, diverse attacks. $L^1$D-RND (because it doesn't follow conventional optimization) creates perturbations that may fall outside the expected adversarial space, allowing it to bypass some defenses.

The ablation table supports this: $L^1$D-RND-Add Only and $L^1$D-RND-Remove Only consistently outperform the fully randomized baseline and sometimes even the full $L^1$D-RND. In particular, Add Only is dominant in many settings, suggesting that the influence of hub-attachment alone can severely distort message-passing dynamics.

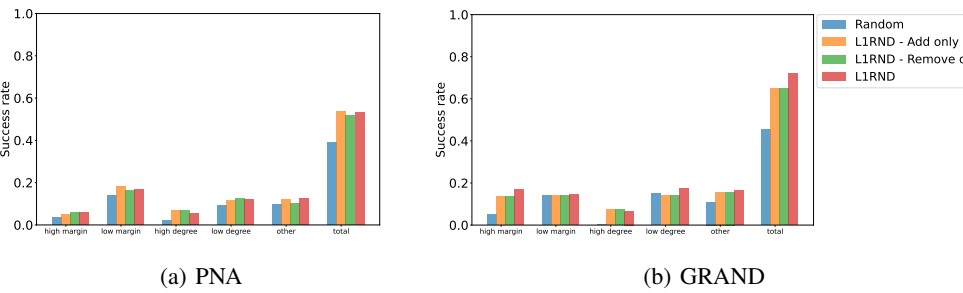

(a) PNA                                (b) GRAND

Figure 8: Ablation study on $L^1$D-RND. The effect of other two variants of $L^1$D-RND: $L^1$D-RND that add edge only (degree affect) and $L^1$D-RND that remove edge only ($L^1$ norm affect) on different class of target nodes: High/low margin, High/low degree and random on budget 5 of evasion setting.

## J  ADVERSARIAL ATTACK METHODS

In this section, we summarize the state-of-the-art adversarial attack techniques considered in our benchmark. These attacks vary in their use of surrogate models, optimization strategies, and access assumptions, offering a diverse testbed for evaluating GNN robustness.

Table 15: Adversarial attacks and victim models used to evaluate the attack method in the corresponding paper.

| ADVERSARIAL ATTACKS | VICTIM MODELS |
|---|---|
| NETTACK | GCN, DEEPWALK, CLN |
| FGA | GCN, GRAREP, GRAPHGAN, DEEPWALK, NODE2VEC, LINE |
| PGD | GCN |
| PR-BCD | GCN, SOFT MEDIAN GDC, SOFT MEDIAN PPRGO, SOFT MEDOID GDC, GDC, PPRGO, RGCN, GCN-JACCARD, GCN-SVD |
| SGA | GCN, SGC, GAT, GRAPHSAGE, CLUSTER-GCN |
| GOTTACK | GCN, GIN, GRAPHSAGE, RGCN, GCN-JACCARD, GCN-SVD, MEDI-ANGCN |

**Nettack.** One of the earliest comprehensive attack frameworks on graph data. Nettack perturbs both graph structure and node features using surrogate gradient information from a trained GCN. Specifically, it applies discrete combinatorial search to structure perturbations, while using gradient-based optimization for feature perturbations. Nettack operates in a targeted, transfer-based setting and has been widely adopted as a baseline for local attacks.

**FGA.** The Fast Gradient Attack generates adversarial examples by computing gradients with respect to the adjacency matrix and feature matrix of a GCN surrogate. Although originally proposed for

unsupervised embeddings (e.g., DeepWalk, LINE), it is also used in GNN contexts. FGA operates in a white-box setting with access to model gradients.

**PGD.** Projected Gradient Descent (PGD) attacks optimize graph perturbations using first-order methods and have been proposed both for attack and for adversarial training. In our benchmark, we use PGD in a transfer setting via a GCN surrogate. However, it is important to note that PGD can also be deployed adaptively in white-box settings, where its full strength is realized by directly attacking the defense model. Adaptive variants are left for future expansion of our benchmark.

**PR-BCD.** This scalable attack framework employs a block coordinate descent (BCD) strategy to optimize perturbations efficiently. It supports both targeted and global settings and introduces custom surrogate losses. In contrast to prior work that evaluated PR-BCD in a transfer setting, we evaluate it both as a transfer-based attack using a fixed GCN surrogate, and in an adaptive, white-box setting where it directly attacks the defended model. This inclusion addresses recent calls for adaptive robustness evaluation.

Table 16: Adversarial attacks and corresponding surrogate models.

| ADVERSARIAL ATTACKS | SURROGATE MODEL |
|---|---|
| NETTACK | GCN |
| FGA | GCN |
| PGD | GCN |
| PR-BCD | GCN |
| SGA | SGC |
| GOTTACK | GCN |

**SGA.** The Simplified Gradient-based Attack reduces computational overhead by restricting optimization to smaller subgraphs. It uses SGC (a simplified GCN variant) as its surrogate and targets structural vulnerabilities efficiently. While not adaptive to defenses, SGA offers a lightweight attack option for benchmarking.

**GOttack.** GOttack introduces a topological approach to adversarial attacks using graph orbits. Instead of relying on gradients, it learns structural patterns to perturb, making it model-agnostic and efficient. GOttack demonstrates strong performance in transfer-based settings, especially on graph datasets where symmetry-based features are predictive.

Table 15 summarizes the original victim models used in each attack's evaluation. Table 16 lists the surrogate models employed during attack generation. While many attacks rely on GCNs as surrogates, we highlight the distinction between transfer-based attacks (e.g., Nettack, FGA, GOttack) and our inclusion of adaptive PR-BCD as a white-box method directly targeting defended models. This allows our benchmark to more realistically assess model robustness under both practical and worst-case threat assumptions.

Table 17: Hyper-parameters used for model selection.

| Models | Layers | Hidden units | Learning rate | Batch size | Drop out | Weight decay | Epochs | Patience | Optimizer | Scheduler | Neighbor aggregation | Initializer |
|---|---|---|---|---|---|---|---|---|---|---|---|---|
| GCN (victim) | 2
3
4 | 32
64
128 | 0.001
0.01
0.1 | see
hidden
units | 0.5
0.6
0.7 | $5 \times 10^{-4}$
$10^{-3}$
0.01 | 1000 | 50 | ADAM | - | sum
mean
max | Glorot (Xavier) |
| GCN (surrogate) | 2
3
4 | 32
64
128 | 0.001
0.01
0.1 | - | 0.5
0.6
0.7 | $5 \times 10^{-4}$
$10^{-3}$
0.01 | 1000 | 50 | ADAM | - | sum | Glorot (Xavier) |
| GIN | 3
4
5 | 64
128
256 | 0.001
0.01
0.1 | 16
32
64 | 0.2
0.5
0.8 | $1 \times 10^{-4}$
$5 \times 10^{-4}$
$10^{-3}$ | 1000 | 50 | ADAM | Step-LR
(step: 50,
gamma: 0.5) | sum
mean
max | Kaiming (He) |
| GSAGE | 2
3
4 | 64
128
256 | 0.001
0.01
0.1 | 16
32
64 | 0.2
0.5
0.8 | $1 \times 10^{-4}$
$5 \times 10^{-4}$
$10^{-3}$ | 1000 | 50 | ADAM | - | sum
mean
max | Kaiming (He) |
| PNA | 3
4
5 | 64
128
256 | 0.001
0.01
0.1 | 16
32
64 | 0.5
0.6
0.7 | $5 \times 10^{-4}$
$1 \times 10^{-4}$
$10^{-3}$ | 10000 | 1000 | ADAM | - | add | Kaiming (He) |
| GNNGuard | 2
3 | 32
64
128 | 0.01
0.005 | - | 0.5
0.6 | $5 \times 10^{-4}$
$10^{-3}$ | 1000 | 50 | ADAM | - | mean | Glorot (Xavier) |
| GRAND | 2 | 16
32
64 | 0.001
0.01
0.005 | - | 0.5
0.6 | $5 \times 10^{-4}$
$1 \times 10^{-4}$
$10^{-3}$ | 1000 | 50 | ADAM | - | sum | Kaiming (He) |
| ElasticGNN | 2
3 | 64
128 | 0.01
0.005
0.05 | - | 0.5
0.6 | $5 \times 10^{-4}$
$5 \times 10^{-5}$
$5 \times 10^{-6}$ | 1000 | 50 | ADAM | - | mean | Kaiming (He) |
| RobustGCN | 2 | 16
32
64 | 0.001
0.01
0.005 | - | 0.5
0.6 | $5 \times 10^{-4}$
$1 \times 10^{-4}$
$10^{-3}$ | 1000 | 50 | ADAM | - | sum
mean | Glorot (Xavier) |
| SGC | 2
3
4 | 32
64
128 | 0.001
0.01
0.1 | - | 0.5
0.6
0.7 | $5 \times 10^{-4}$
$10^{-3}$
0.01 | 1000 | 50 | ADAM | - | sum | Glorot (Xavier) |
| GCORN | 2 | 32
64
128 | 0.001
0.01
0.1 | - | 0.5
0.6
0.7 | - | 1000 | 50 | ADAM | - | - | Orthogonal Initialization |
| RUNG | 2
3
4 | 32
64 | 0.001
0.01 | - | 0.5
0.6 | - | 1000 | 50 | ADAM | - | - | Kaiming (He) |
| NoisyGNN | 2 | 16
64
128
512 | 0.001
0.01
0.1 | - | 0.5
0.6
0.7 | - | 300 | 50 | ADAM | - | - | Glorot (Xavier) |

Table 18: **Vanilla Homophily Results.** Evaluating six adversarial attacks on four vanilla attack victim models - Miss-classification rate (%). Higher is better. Best performance in **bold**, second best underlined.

| | Dataset | CORA | | | | | CITESEER | | | | | PUBMED | | | | | |
|---|---|---|---|---|---|---|---|---|---|---|---|---|---|---|---|---|---|
| | Δ → | 1 | 2 | 3 | 4 | 5 | 1 | 2 | 3 | 4 | 5 | 1 | 2 | 3 | 4 | 5 | Avg. Rank |
| **Evasion** GCN | L¹D-RND | 13.20 ± 4.33 | 22.53 ± 6.95 | 29.20 ± 5.60 | 32.93 ± 6.67 | 32.13 ± 5.04 | 15.20 ± 4.20 | 28.67 ± 12.39 | 36.53 ± 9.09 | 37.73 ± 9.44 | 42.13 ± 7.11 | 10.93 ± 3.01 | 15.47 ± 4.87 | 17.20 ± 6.58 | 18.80 ± 5.28 | 23.60 ± 9.63 | 7.00 |
| | FGA | 27.87 ± 3.74 | 46.13 ± 7.03 | 55.73 ± 4.33 | 60.93 ± 5.06 | 64.67 ± 4.76 | 25.47 ± 4.10 | 36.00 ± 8.18 | 50.27 ± 5.90 | 56.53 ± 2.97 | 58.27 ± 4.27 | 35.60 ± 3.14 | 47.60 ± 5.77 | 55.33 ± 4.12 | 58.40 ± 2.29 | 58.00 ± 3.02 | 3.40 |
| | NETTACK | 29.60 ± 5.19 | 49.60 ± 5.77 | 56.80 ± 4.52 | 62.13 ± 4.17 | 63.73 ± 3.99 | 28.13 ± 7.15 | 45.33 ± 9.06 | 49.87 ± 9.61 | 58.93 ± 5.85 | 59.73 ± 3.92 | 33.73 ± 3.37 | 55.73 ± 4.33 | 57.47 ± 2.20 | 58.67 ± 2.09 | 58.67 ± 2.23 | 2.20 |
| | PGD | 29.33 ± 4.12 | 48.00 ± 5.24 | 54.93 ± 3.28 | 59.60 ± 4.36 | 59.20 ± 5.49 | 25.47 ± 4.87 | 40.27 ± 7.55 | 48.13 ± 7.23 | 52.27 ± 7.28 | 55.47 ± 5.15 | 34.13 ± 3.66 | 40.67 ± 3.98 | 40.93 ± 5.18 | 44.80 ± 5.75 | 44.93 ± 6.27 | 4.93 |
| | PR-BCD (NA) | **32.13 ± 3.50** | **53.87 ± 4.44** | **59.20 ± 2.81** | **63.20 ± 3.36** | **65.73 ± 3.92** | **34.53 ± 6.82** | **54.67 ± 5.05** | **60.40 ± 4.42** | **64.67 ± 5.16** | **65.87 ± 4.17** | 29.60 ± 3.04 | 36.53 ± 3.25 | 45.73 ± 3.92 | 51.47 ± 4.56 | 54.13 ± 3.25 | 2.47 |
| | SGA | 26.27 ± 3.69 | 44.67 ± 7.58 | 55.33 ± 4.88 | 58.13 ± 3.81 | 60.67 ± 5.00 | 23.47 ± 2.77 | 35.87 ± 6.07 | 47.47 ± 4.31 | 51.07 ± 4.95 | 53.73 ± 3.99 | 34.27 ± 4.27 | 42.13 ± 6.30 | 51.33 ± 7.08 | 54.27 ± 6.71 | 54.80 ± 4.46 | 5.13 |
| | GOttack | 28.53 ± 4.37 | 49.33 ± 6.83 | 56.13 ± 5.26 | 60.53 ± 6.12 | 62.13 ± 4.87 | 25.60 ± 4.22 | 39.33 ± 9.03 | 51.60 ± 6.10 | 57.33 ± 4.82 | 56.80 ± 7.40 | **35.87 ± 3.42** | **48.00 ± 6.46** | 56.13 ± 3.58 | 58.13 ± 2.77 | **59.60 ± 2.16** | 2.87 |
| GIN | L¹D-RND | 20.13 ± 7.35 | 37.07 ± 15.27 | 42.67 ± 8.80 | 43.20 ± 10.05 | 43.73 ± 12.02 | 16.80 ± 10.87 | 33.47 ± 15.76 | 35.73 ± 14.94 | 38.53 ± 13.17 | 43.33 ± 12.57 | 13.33 ± 5.64 | 23.07 ± 12.89 | 26.67 ± 12.75 | 22.40 ± 10.26 | 22.27 ± 11.54 | 6.60 |
| | FGA | 23.47 ± 4.63 | 31.07 ± 7.09 | 42.40 ± 9.77 | 50.67 ± 9.76 | 55.47 ± 7.31 | 17.07 ± 7.70 | 34.27 ± 10.33 | 42.13 ± 10.29 | 48.53 ± 9.96 | 55.33 ± 9.99 | 32.53 ± 21.89 | 44.53 ± 22.54 | 45.33 ± 23.31 | **54.67 ± 24.86** | 55.47 ± 27.60 | 3.60 |
| | NETTACK | **26.53 ± 6.91** | **45.20 ± 7.04** | **53.87 ± 7.03** | **59.20 ± 6.45** | **65.20 ± 6.88** | **21.07 ± 8.38** | **42.13 ± 8.30** | **51.60 ± 11.11** | **60.27 ± 10.61** | **62.53 ± 10.91** | 33.60 ± 22.81 | 45.47 ± 21.97 | 51.07 ± 24.92 | 53.60 ± 25.38 | **55.87 ± 26.73** | 1.27 |
| | PGD | 25.07 ± 4.65 | 36.27 ± 9.07 | 41.60 ± 11.09 | 43.60 ± 10.99 | 47.60 ± 10.20 | 20.53 ± 7.87 | 36.13 ± 10.29 | 38.93 ± 12.78 | 41.87 ± 13.34 | 45.60 ± 15.27 | 28.13 ± 16.94 | 37.33 ± 18.79 | 36.80 ± 17.66 | 40.00 ± 18.02 | 39.87 ± 20.54 | 4.93 |
| | PR-BCD (NA) | 27.47 ± 5.32 | 40.53 ± 7.73 | 48.27 ± 11.61 | 52.80 ± 11.51 | 54.13 ± 10.41 | 19.20 ± 7.08 | 37.07 ± 9.35 | 42.93 ± 6.92 | 49.07 ± 9.50 | 49.60 ± 13.18 | 25.73 ± 13.35 | 30.00 ± 13.75 | 29.87 ± 13.60 | 34.00 ± 16.14 | 36.13 ± 17.67 | 3.73 |
| | SGA | 22.40 ± 4.67 | 34.67 ± 6.62 | 38.93 ± 9.65 | 44.40 ± 11.81 | 51.20 ± 13.50 | 18.93 ± 8.71 | 34.53 ± 8.77 | 38.40 ± 10.40 | 46.40 ± 10.29 | 49.73 ± 9.98 | **35.87 ± 23.90** | 39.33 ± 22.62 | 44.13 ± 24.81 | 48.00 ± 23.82 | 50.27 ± 26.33 | 4.60 |
| | GOttack | 25.33 ± 4.76 | 37.73 ± 9.85 | 44.80 ± 8.06 | 48.67 ± 12.09 | 51.73 ± 11.90 | 20.13 ± 9.09 | 34.13 ± 9.58 | 43.47 ± 12.29 | 49.20 ± 12.89 | 51.33 ± 12.75 | 31.87 ± 23.00 | 45.73 ± 24.85 | 44.53 ± 24.91 | 48.00 ± 24.72 | 46.80 ± 24.84 | 3.27 |
| GSAGE | L¹D-RND | 15.87 ± 6.25 | 34.00 ± 10.95 | 38.93 ± 7.48 | 40.93 ± 5.70 | 41.87 ± 7.19 | 19.07 ± 7.28 | 38.27 ± 11.56 | 40.40 ± 10.32 | 42.67 ± 13.62 | 45.20 ± 10.98 | 10.00 ± 3.70 | 18.00 ± 5.07 | 16.40 ± 8.04 | 14.27 ± 5.65 | 13.33 ± 7.39 | 6.27 |
| | FGA | 22.93 ± 6.80 | 31.07 ± 6.88 | 39.20 ± 9.91 | 49.47 ± 8.57 | 55.33 ± 7.66 | 23.87 ± 4.37 | 37.60 ± 13.25 | 42.27 ± 11.56 | 51.47 ± 12.01 | 57.87 ± 7.27 | 23.73 ± 7.85 | 33.20 ± 7.08 | 36.13 ± 7.84 | 39.47 ± 11.77 | 42.40 ± 13.61 | 3.00 |
| | NETTACK | 22.00 ± 3.93 | **41.87 ± 6.74** | **55.60 ± 6.73** | **60.93 ± 6.18** | **63.73 ± 4.89** | 22.27 ± 4.27 | **49.07 ± 10.77** | **54.93 ± 9.16** | **62.27 ± 8.14** | **58.53 ± 11.80** | 24.40 ± 7.83 | 31.73 ± 6.84 | **39.87 ± 12.06** | **42.93 ± 11.39** | 42.53 ± 14.76 | 1.53 |
| | PGD | 24.00 ± 4.84 | 33.47 ± 7.31 | 40.27 ± 9.13 | 43.20 ± 10.44 | 46.13 ± 9.40 | 22.53 ± 3.96 | 37.07 ± 8.10 | 36.67 ± 11.28 | 42.40 ± 10.12 | 45.33 ± 10.05 | 23.47 ± 7.95 | 29.33 ± 5.89 | 29.47 ± 5.42 | 30.13 ± 6.48 | 32.00 ± 6.00 | 5.00 |
| | PR-BCD (NA) | 25.07 ± 4.65 | 35.47 ± 9.33 | 43.07 ± 9.47 | 45.73 ± 7.67 | 51.07 ± 9.62 | 22.13 ± 3.96 | 41.20 ± 8.81 | 47.60 ± 7.68 | 49.73 ± 12.44 | 53.60 ± 11.39 | 21.73 ± 6.76 | 25.73 ± 6.09 | 27.20 ± 6.75 | 28.13 ± 7.11 | 28.80 ± 6.36 | 3.93 |
| | SGA | 21.47 ± 2.97 | 33.47 ± 9.02 | 43.20 ± 12.04 | 46.67 ± 12.78 | 49.47 ± 13.82 | 19.60 ± 4.08 | 33.33 ± 9.85 | 38.13 ± 6.25 | 44.00 ± 8.42 | 45.20 ± 8.87 | 24.40 ± 8.42 | 31.00 ± 5.18 | 32.53 ± 6.91 | 36.13 ± 7.50 | 36.53 ± 7.07 | 4.60 |
| | GOttack | 23.20 ± 4.33 | 35.33 ± 10.71 | 40.40 ± 9.36 | 48.27 ± 10.92 | 51.73 ± 12.62 | 21.33 ± 3.27 | 35.33 ± 8.77 | 43.73 ± 10.22 | 45.07 ± 9.75 | 52.27 ± 11.97 | 24.80 ± 8.13 | 32.93 ± 7.98 | 31.20 ± 8.84 | 34.13 ± 7.50 | 33.60 ± 9.08 | 3.67 |
| PNA | L¹D-RND | 33.20 ± 8.81 | 41.60 ± 12.86 | 47.20 ± 9.44 | 45.20 ± 12.14 | 53.07 ± 15.36 | 39.33 ± 11.82 | 46.13 ± 9.84 | 46.27 ± 10.05 | 52.13 ± 10.78 | 55.47 ± 11.12 | **39.33 ± 26.16** | 18.27 ± 8.41 | 19.20 ± 8.34 | 20.27 ± 8.34 | 24.27 ± 12.80 | 3.73 |
| | FGA | 26.13 ± 7.84 | 40.40 ± 7.83 | 43.20 ± 9.99 | 55.33 ± 7.66 | 62.80 ± 8.74 | 26.93 ± 7.40 | 35.07 ± 8.24 | 43.20 ± 13.87 | 50.40 ± 12.61 | 56.13 ± 13.10 | 30.27 ± 9.65 | 39.87 ± 17.38 | 42.13 ± 18.24 | 48.67 ± 18.83 | 50.80 ± 18.04 | 3.87 |
| | NETTACK | 28.53 ± 6.99 | **49.87 ± 9.49** | **59.60 ± 7.97** | **66.00 ± 7.09** | **68.80 ± 9.13** | 32.80 ± 8.87 | **51.60 ± 9.95** | **66.80 ± 9.34** | **68.40 ± 7.94** | **69.60 ± 7.60** | 30.53 ± 10.38 | 39.33 ± 16.45 | **46.80 ± 19.21** | **50.80 ± 18.28** | **52.80 ± 17.77** | 1.53 |
| | PGD | 26.80 ± 5.12 | 40.27 ± 10.61 | 39.47 ± 8.12 | 45.47 ± 6.86 | 48.40 ± 11.09 | 28.53 ± 5.63 | 33.20 ± 9.00 | 39.47 ± 9.36 | 42.53 ± 7.59 | 42.53 ± 10.46 | 28.53 ± 6.74 | 28.27 ± 10.36 | 30.53 ± 10.86 | 30.13 ± 10.39 | 30.93 ± 10.36 | 5.93 |
| | PR-BCD (NA) | 26.80 ± 7.63 | 43.07 ± 5.44 | 47.20 ± 6.41 | 48.13 ± 9.24 | 52.13 ± 10.51 | 30.93 ± 7.78 | 42.53 ± 7.15 | 48.27 ± 10.58 | 47.60 ± 4.42 | 53.07 ± 8.51 | 20.27 ± 11.93 | 23.73 ± 10.31 | 24.93 ± 10.31 | 25.87 ± 12.03 | 28.53 ± 9.15 | 4.60 |
| | SGA | 31.73 ± 7.09 | 40.27 ± 10.05 | 46.67 ± 9.76 | 44.40 ± 6.81 | 50.53 ± 8.96 | 30.27 ± 7.17 | 36.27 ± 8.31 | 44.93 ± 11.00 | 48.00 ± 7.71 | 48.13 ± 7.84 | 30.93 ± 10.98 | 35.07 ± 14.10 | 36.80 ± 17.07 | 43.47 ± 17.31 | 44.93 ± 19.85 | 4.27 |
| | GOttack | 31.60 ± 7.57 | 41.47 ± 11.62 | 49.33 ± 13.87 | 52.13 ± 9.64 | 50.53 ± 12.25 | 27.87 ± 11.30 | 42.27 ± 15.45 | 43.73 ± 10.22 | 45.60 ± 9.77 | 45.07 ± 12.09 | 28.93 ± 13.11 | 40.53 ± 16.69 | 37.60 ± 13.72 | 39.20 ± 14.34 | 40.40 ± 14.33 | 4.07 |
| **Poison** GCN | L¹D-RND | 15.47 ± 4.03 | 23.33 ± 6.79 | 30.27 ± 6.18 | 33.47 ± 7.03 | 34.67 ± 6.87 | 16.27 ± 4.33 | 29.87 ± 12.77 | 37.73 ± 9.56 | 40.13 ± 9.12 | 42.93 ± 6.04 | 9.73 ± 3.10 | 15.47 ± 4.81 | 17.33 ± 5.64 | 19.33 ± 5.49 | 23.20 ± 9.91 | 7.00 |
| | FGA | 30.00 ± 5.61 | 50.00 ± 4.78 | 56.27 ± 4.13 | 60.53 ± 4.37 | **65.07 ± 4.77** | 31.87 ± 7.39 | 49.87 ± 3.81 | 56.67 ± 5.43 | 60.13 ± 5.83 | **63.73 ± 5.39** | 35.33 ± 3.09 | 47.47 ± 4.93 | 55.33 ± 4.58 | **58.53 ± 1.77** | 58.00 ± 2.62 | 2.93 |
| | NETTACK | **33.47 ± 3.58** | 51.87 ± 4.56 | **58.67 ± 2.69** | **62.13 ± 3.66** | 63.07 ± 3.10 | **36.40 ± 6.51** | 51.73 ± 7.52 | 55.87 ± 10.38 | 62.27 ± 5.65 | 63.33 ± 5.43 | 34.13 ± 4.69 | **54.80 ± 3.36** | **57.47 ± 1.92** | 58.40 ± 2.03 | 58.40 ± 1.88 | 2.07 |
| | PGD | 31.73 ± 3.69 | 48.80 ± 4.59 | 55.60 ± 2.95 | 59.33 ± 2.89 | 58.53 ± 4.87 | 30.80 ± 6.79 | 45.87 ± 5.83 | 51.73 ± 9.56 | 56.27 ± 7.33 | 57.73 ± 5.50 | 33.60 ± 3.48 | 40.67 ± 4.45 | 40.93 ± 5.12 | 45.07 ± 5.50 | 45.47 ± 6.44 | 5.47 |
| | PR-BCD (NA) | 32.80 ± 6.04 | **53.87 ± 3.66** | 57.60 ± 3.31 | 60.67 ± 3.52 | 63.73 ± 3.28 | 34.27 ± 6.32 | **52.40 ± 3.64** | **57.87 ± 5.42** | **62.40 ± 4.73** | 63.73 ± 3.84 | 29.60 ± 3.14 | 36.53 ± 3.07 | 45.47 ± 3.42 | 50.13 ± 4.50 | 54.40 ± 2.85 | 3.00 |
| | SGA | 29.33 ± 3.83 | 48.13 ± 6.48 | 55.87 ± 3.81 | 60.00 ± 2.73 | 63.73 ± 4.46 | 25.20 ± 3.61 | 44.13 ± 4.93 | 53.20 ± 4.52 | 54.53 ± 4.10 | 60.27 ± 4.77 | 34.13 ± 4.03 | 41.60 ± 5.41 | 50.93 ± 7.48 | 54.00 ± 6.05 | 54.40 ± 4.91 | 4.93 |
| | GOttack | 33.33 ± 7.20 | 51.47 ± 6.65 | 57.73 ± 5.18 | 61.07 ± 3.92 | 63.20 ± 3.84 | 34.27 ± 8.38 | 48.40 ± 6.77 | 57.87 ± 4.63 | 59.33 ± 4.82 | 62.00 ± 6.09 | **35.60 ± 3.40** | 47.60 ± 5.51 | 55.73 ± 4.53 | 58.13 ± 2.67 | **59.20 ± 1.97** | 2.60 |
| GIN | L¹D-RND | 23.07 ± 8.07 | 38.00 ± 13.50 | 42.40 ± 8.53 | 43.73 ± 11.54 | 45.33 ± 12.09 | 20.40 ± 9.39 | 36.13 ± 13.13 | 38.40 ± 11.86 | 40.13 ± 11.55 | 43.87 ± 10.54 | 14.67 ± 6.79 | 27.07 ± 14.10 | 29.07 ± 13.75 | 23.33 ± 12.11 | 24.27 ± 12.69 | 7.00 |
| | FGA | 31.87 ± 6.39 | 46.27 ± 12.44 | 58.40 ± 8.92 | 66.13 ± 7.69 | **72.27 ± 6.50** | 30.67 ± 10.24 | 44.80 ± 10.58 | 54.80 ± 8.61 | 63.87 ± 7.84 | 68.00 ± 11.36 | 32.93 ± 20.18 | 44.67 ± 23.17 | 45.47 ± 22.51 | **55.60 ± 24.18** | 55.73 ± 26.43 | 2.73 |
| | NETTACK | 34.13 ± 6.61 | **55.07 ± 7.67** | **64.93 ± 9.35** | **69.60 ± 6.47** | 71.87 ± 6.25 | 34.93 ± 10.17 | **54.13 ± 7.50** | **64.93 ± 11.18** | **70.93 ± 6.76** | **73.60 ± 9.51** | 32.00 ± 21.46 | 45.20 ± 21.20 | 50.80 ± 24.53 | 54.67 ± 25.23 | **56.53 ± 25.57** | 1.47 |
| | PGD | 29.07 ± 4.13 | 43.60 ± 9.36 | 48.93 ± 8.65 | 53.60 ± 8.15 | 54.80 ± 9.97 | 28.67 ± 8.16 | 46.07 ± 10.42 | 49.33 ± 9.40 | 51.47 ± 9.45 | 56.27 ± 10.28 | 28.40 ± 17.37 | 38.00 ± 18.53 | 36.80 ± 18.31 | 41.73 ± 19.56 | 40.80 ± 20.51 | 5.53 |
| | PR-BCD (NA) | 33.87 ± 6.57 | 50.67 ± 7.20 | 57.07 ± 7.17 | 61.07 ± 7.48 | 66.80 ± 5.99 | 28.27 ± 11.34 | 50.53 ± 10.73 | 58.00 ± 7.89 | 59.87 ± 9.64 | 65.47 ± 9.72 | 25.07 ± 12.49 | 30.40 ± 14.35 | 31.47 ± 16.79 | 35.47 ± 17.56 | 36.40 ± 18.93 | 4.33 |
| | SGA | 31.33 ± 8.06 | 45.33 ± 9.31 | 54.00 ± 8.42 | 62.40 ± 6.77 | 69.33 ± 7.77 | 30.40 ± 7.57 | 46.93 ± 6.67 | 53.33 ± 10.05 | 62.00 ± 8.32 | 64.27 ± 6.71 | **34.40 ± 22.13** | 41.60 ± 24.98 | 44.00 ± 23.20 | 47.73 ± 23.04 | 49.20 ± 25.37 | 3.87 |
| | GOttack | **34.93 ± 10.90** | 46.40 ± 16.16 | 54.13 ± 11.99 | 57.87 ± 15.77 | 66.67 ± 12.66 | 33.07 ± 12.33 | 47.87 ± 7.87 | 60.93 ± 7.70 | 62.93 ± 9.85 | 65.47 ± 10.54 | 33.87 ± 23.81 | **46.67 ± 24.56** | 45.20 ± 24.19 | 47.20 ± 24.68 | 47.20 ± 24.27 | 3.07 |
| GSAGE | L¹D-RND | 18.13 ± 6.57 | 38.13 ± 10.41 | 40.40 ± 7.45 | 42.67 ± 7.43 | 44.40 ± 7.06 | 21.33 ± 9.09 | 40.13 ± 10.43 | 42.27 ± 9.76 | 43.47 ± 11.84 | 45.33 ± 8.27 | 10.13 ± 4.44 | 17.73 ± 4.95 | 18.00 ± 8.45 | 14.27 ± 5.50 | 14.80 ± 4.65 | 6.93 |
| | FGA | 26.67 ± 7.55 | 38.13 ± 10.41 | 48.67 ± 14.49 | 59.07 ± 10.25 | 63.87 ± 8.19 | 25.60 ± 5.41 | 46.80 ± 13.60 | 52.00 ± 13.92 | 59.20 ± 10.22 | 65.20 ± 9.97 | 23.33 ± 8.51 | 31.87 ± 6.35 | 35.73 ± 7.52 | 39.20 ± 10.69 | **42.40 ± 13.51** | 2.67 |
| | NETTACK | 27.07 ± 8.71 | **48.27 ± 8.91** | **59.60 ± 9.51** | **65.33 ± 7.08** | **70.93 ± 7.52** | 24.93 ± 6.09 | **54.80 ± 7.88** | **58.27 ± 10.66** | **66.53 ± 8.05** | 63.20 ± 11.75 | 23.73 ± 7.67 | **32.53 ± 6.30** | **39.60 ± 11.34** | **42.53 ± 11.82** | 42.27 ± 14.94 | 1.53 |
| | PGD | **28.80 ± 8.38** | 38.93 ± 10.05 | 44.67 ± 9.58 | 48.80 ± 11.08 | 53.60 ± 11.57 | 23.07 ± 4.13 | 40.93 ± 8.55 | 47.07 ± 10.08 | 48.67 ± 10.38 | 51.73 ± 10.08 | 23.33 ± 7.95 | 28.80 ± 5.75 | 28.93 ± 6.54 | 29.87 ± 6.78 | 31.87 ± 7.50 | 5.00 |
| | PR-BCD (NA) | 28.80 ± 7.63 | 42.53 ± 11.84 | 50.13 ± 12.29 | 54.00 ± 9.86 | 61.47 ± 11.10 | 23.47 ± 4.93 | 45.87 ± 8.40 | 53.33 ± 8.06 | 55.73 ± 12.71 | 60.67 ± 9.58 | 23.47 ± 6.67 | 25.07 ± 5.85 | 25.47 ± 7.31 | 28.00 ± 7.79 | 28.40 ± 5.91 | 4.07 |
| | SGA | 22.67 ± 3.18 | 40.93 ± 11.95 | 51.07 ± 13.98 | 57.07 ± 14.56 | 60.53 ± 15.72 | 22.00 ± 4.96 | 37.60 ± 10.26 | 46.13 ± 6.99 | 49.47 ± 10.13 | 53.73 ± 10.22 | **25.20 ± 8.87** | 29.07 ± 5.26 | 32.13 ± 5.78 | 36.27 ± 6.96 | 35.47 ± 6.44 | 4.13 |
| | GOttack | 27.47 ± 9.55 | 43.87 ± 16.27 | 51.20 ± 12.18 | 53.47 ± 13.57 | 59.73 ± 13.98 | 23.73 ± 6.54 | 40.80 ± 9.47 | 51.87 ± 13.76 | 54.53 ± 14.23 | 60.13 ± 13.91 | 24.13 ± 8.90 | 29.87 ± 7.58 | 29.33 ± 9.06 | 33.20 ± 6.62 | 33.20 ± 8.55 | 3.67 |
| PNA | L¹D-RND | 37.47 ± 7.80 | 44.00 ± 13.65 | 48.13 ± 11.82 | 50.40 ± 13.27 | 57.20 ± 13.26 | 42.27 ± 12.62 | 51.07 ± 10.98 | 52.13 ± 9.96 | 52.53 ± 6.70 | 59.60 ± 9.42 | **39.33 ± 26.37** | 19.60 ± 10.83 | 20.67 ± 8.74 | 20.40 ± 8.69 | 23.47 ± 10.89 | 5.67 |
| | FGA | 36.67 ± 8.67 | 46.93 ± 7.67 | 58.80 ± 7.59 | 64.67 ± 9.22 | 71.20 ± 8.97 | 40.53 ± 9.55 | 52.00 ± 12.19 | 62.53 ± 10.18 | 66.27 ± 10.02 | 71.20 ± 12.94 | 41.73 ± 15.75 | 44.13 ± 15.75 | 46.13 ± 17.80 | 48.00 ± 18.93 | 50.80 ± 18.04 | 2.60 |
| | NETTACK | 35.60 ± 8.11 | **59.60 ± 9.77** | **64.67 ± 7.66** | **73.33 ± 5.84** | **74.93 ± 6.27** | **42.93 ± 9.00** | **62.40 ± 9.01** | **73.73 ± 5.06** | **77.73 ± 6.23** | **79.87 ± 5.48** | 30.53 ± 11.48 | 38.80 ± 18.23 | **45.87 ± 18.29** | **49.33 ± 19.22** | **53.33 ± 19.47** | 1.53 |
| | PGD | 31.47 ± 7.27 | 43.60 ± 8.29 | 49.07 ± 6.71 | 53.87 ± 7.11 | 57.47 ± 6.99 | 39.87 ± 9.30 | 43.73 ± 10.66 | 50.00 ± 7.60 | 56.40 ± 6.77 | 55.73 ± 7.55 | 29.07 ± 6.96 | 29.07 ± 11.54 | 31.33 ± 10.89 | 32.53 ± 11.38 | 30.93 ± 10.00 | 6.00 |
| | PR-BCD (NA) | 36.13 ± 7.19 | 50.80 ± 6.36 | 56.00 ± 6.55 | 59.47 ± 6.99 | 59.33 ± 6.40 | 38.40 ± 8.95 | 55.47 ± 7.87 | 59.73 ± 7.44 | 62.13 ± 8.67 | 65.20 ± 5.75 | 19.07 ± 10.36 | 24.13 ± 9.18 | 24.93 ± 10.31 | 26.13 ± 11.60 | 29.20 ± 9.00 | 4.93 |
| | SGA | **41.47 ± 9.61** | 50.27 ± 10.50 | 56.53 ± 12.97 | 60.53 ± 10.84 | 68.00 ± 9.47 | 39.20 ± 9.53 | 51.07 ± 8.17 | 60.80 ± 10.58 | 63.33 ± 8.74 | 65.07 ± 8.10 | 31.20 ± 10.66 | 37.60 ± 13.14 | 36.93 ± 16.09 | 43.87 ± 17.88 | 43.07 ± 17.02 | 3.67 |
| | GOttack | 38.67 ± 7.16 | 52.93 ± 15.53 | 59.87 ± 11.82 | 63.33 ± 10.87 | 64.53 ± 11.72 | 40.13 ± 10.01 | 55.07 ± 11.97 | 55.73 ± 7.96 | 61.33 ± 9.79 | 64.13 ± 7.11 | 29.47 ± 12.66 | 37.87 ± 14.51 | 37.73 ± 12.09 | 40.93 ± 15.56 | 39.73 ± 15.27 | 3.60 |

Table 19: **Vanilla Heterophily Results - 1/2.** Evaluating six adversarial attacks on four vanilla attack victim models on SQUIRREL- Miss-classification rate (%). Higher is better. Best performance in **bold**, second best underlined. GIN and PNA are excluded from this table due to OOR.

| Dataset | | Δ → | 1 | 2 | 3 | 4 | 5 | Avg. Rank |
|---|---|---|---|---|---|---|---|---|
| | | | | | SQUIRREL | | | |
| Evasion | GCN | L¹D-RND | 24.93 ± 33.09 | 64.53 ± 6.91 | **72.80 ± 7.36** | 71.60 ± 5.67 | 71.47 ± 4.56 | 2.80 |
| | | FGA | 62.40 ± 14.64 | 65.87 ± 15.45 | 71.07 ± 7.40 | 71.07 ± 5.80 | 71.60 ± 4.36 | 2.20 |
| | | NETTACK | 1.87 ± 3.34 | 1.47 ± 2.77 | 1.33 ± 1.63 | 1.60 ± 2.29 | 2.13 ± 2.20 | 7.00 |
| | | PGD | 47.73 ± 10.25 | 58.93 ± 12.28 | 65.87 ± 10.04 | 62.80 ± 11.73 | 64.80 ± 13.54 | 4.60 |
| | | PR-BCD (NA) | **69.87 ± 10.76** | **69.73 ± 6.41** | 69.60 ± 6.56 | **71.87 ± 4.63** | **73.07 ± 5.12** | 1.40 |
| | | SGA | 52.00 ± 6.19 | 59.20 ± 4.46 | 60.53 ± 8.83 | 65.07 ± 8.24 | 66.80 ± 7.04 | 4.00 |
| | | GOttack | 13.33 ± 4.64 | 10.93 ± 5.55 | 14.00 ± 6.05 | 11.20 ± 5.85 | 13.60 ± 5.46 | 6.00 |
| | GSAGE | L¹D-RND | 13.60 ± 16.23 | 33.87 ± 11.77 | 37.87 ± 9.36 | 44.67 ± 11.05 | 49.47 ± 12.43 | 4.80 |
| | | FGA | 30.80 ± 14.89 | **44.00 ± 19.20** | **49.33 ± 21.90** | **49.73 ± 19.36** | **53.47 ± 19.52** | 1.40 |
| | | NETTACK | 19.07 ± 7.40 | 20.40 ± 10.06 | 21.73 ± 9.07 | 24.80 ± 6.49 | 21.60 ± 7.41 | 6.80 |
| | | PGD | 30.27 ± 12.71 | 39.07 ± 14.54 | 43.73 ± 14.06 | 48.40 ± 15.95 | 49.47 ± 16.89 | 3.20 |
| | | PR-BCD (NA) | **36.27 ± 11.63** | 42.80 ± 9.34 | 41.33 ± 7.20 | 42.93 ± 10.17 | 45.87 ± 11.17 | 3.40 |
| | | SGA | 34.00 ± 9.47 | 40.80 ± 8.06 | 41.60 ± 7.75 | 48.27 ± 12.28 | 50.53 ± 8.19 | 2.60 |
| | | GOttack | 24.00 ± 9.74 | 27.33 ± 8.87 | 28.13 ± 7.46 | 28.27 ± 8.58 | 30.80 ± 9.41 | 5.80 |
| Poison | GCN | L¹D-RND | 34.67 ± 27.36 | 66.80 ± 7.00 | **73.87 ± 7.11** | 71.47 ± 5.68 | 71.47 ± 4.56 | 2.80 |
| | | FGA | 63.87 ± 12.25 | 67.20 ± 12.85 | 71.73 ± 6.23 | 71.47 ± 6.02 | 72.40 ± 4.01 | 2.20 |
| | | NETTACK | 2.80 ± 2.24 | 1.87 ± 2.88 | 2.00 ± 2.14 | 2.00 ± 2.51 | 2.27 ± 2.37 | 7.00 |
| | | PGD | 52.27 ± 8.21 | 62.00 ± 10.25 | 68.00 ± 7.82 | 64.80 ± 9.85 | 65.73 ± 12.21 | 4.20 |
| | | PR-BCD (NA) | **70.27 ± 11.16** | **70.67 ± 6.75** | 70.67 ± 5.89 | **72.67 ± 4.05** | **74.27 ± 4.77** | 1.40 |
| | | SGA | 53.47 ± 5.97 | 59.87 ± 4.98 | 61.87 ± 8.53 | 64.80 ± 8.55 | 66.40 ± 7.14 | 4.40 |
| | | GOttack | 13.60 ± 3.79 | 12.13 ± 5.48 | 13.47 ± 5.78 | 12.00 ± 5.50 | 13.60 ± 4.91 | 6.00 |
| | GSAGE | L¹D-RND | 24.53 ± 16.62 | 39.33 ± 11.56 | 44.53 ± 9.43 | 48.40 ± 8.92 | 53.07 ± 8.68 | 5.00 |
| | | FGA | 37.20 ± 10.39 | **48.93 ± 15.62** | **52.13 ± 18.05** | **54.00 ± 18.06** | **57.33 ± 18.09** | 1.40 |
| | | NETTACK | 20.27 ± 7.00 | 23.20 ± 10.16 | 23.60 ± 8.85 | 25.47 ± 5.37 | 27.20 ± 4.33 | 7.00 |
| | | PGD | 36.67 ± 9.52 | 43.60 ± 11.09 | 46.93 ± 12.14 | 52.80 ± 14.56 | 55.60 ± 12.99 | 3.20 |
| | | PR-BCD (NA) | **40.93 ± 10.25** | 47.60 ± 10.15 | 50.80 ± 7.63 | 50.93 ± 9.07 | 50.13 ± 7.23 | 2.80 |
| | | SGA | 39.20 ± 7.48 | 47.47 ± 6.70 | 48.40 ± 9.11 | 51.47 ± 11.20 | 55.33 ± 8.87 | 2.80 |
| | | GOttack | 27.20 ± 8.44 | 26.80 ± 8.27 | 30.53 ± 6.74 | 30.13 ± 9.05 | 34.80 ± 5.28 | 5.80 |

Table 20: **Vanilla Heterophily Results - 2/2.** Evaluating six adversarial attacks on four vanilla attack victim models on CHAMELEON- Miss-classification rate (%). Higher is better. Best performance in **bold**, second best underlined. PNA is excluded from this table due to OOR.

| Dataset | | Δ → | 1 | 2 | 3 | 4 | 5 | Avg. Rank |
|---|---|---|---|---|---|---|---|---|
| | | | | | CHAMELEON | | | |
| Evasion | GCN | L¹D-RND | 21.87 ± 28.89 | 50.93 ± 9.71 | 62.00 ± 6.05 | 66.67 ± 7.12 | 66.53 ± 7.07 | 3.00 |
| | | FGA | **62.40 ± 7.72** | 58.00 ± 12.72 | 59.87 ± 15.97 | 56.40 ± 12.54 | 61.07 ± 12.30 | 3.00 |
| | | NETTACK | 3.07 ± 2.60 | 5.47 ± 3.96 | 4.93 ± 4.33 | 6.27 ± 4.20 | 6.27 ± 4.83 | 7.00 |
| | | PGD | 44.00 ± 20.95 | 50.53 ± 18.10 | 61.07 ± 14.64 | 57.73 ± 15.47 | 59.47 ± 14.05 | 3.60 |
| | | PR-BCD (NA) | 58.00 ± 18.53 | **67.87 ± 20.46** | **70.27 ± 8.88** | **67.07 ± 10.66** | **73.47 ± 8.77** | 1.20 |
| | | SGA | 45.47 ± 10.38 | 49.07 ± 9.82 | 55.60 ± 7.64 | 57.33 ± 9.37 | 55.33 ± 6.70 | 4.40 |
| | | GOttack | 23.47 ± 8.16 | 19.20 ± 5.75 | 22.53 ± 7.27 | 17.33 ± 6.62 | 22.80 ± 9.13 | 5.80 |
| | GIN | L¹D-RND | 18.13 ± 22.63 | **56.00 ± 11.16** | 55.60 ± 16.16 | **64.13 ± 13.70** | 62.93 ± 13.81 | 2.00 |
| | | FGA | **40.93 ± 15.42** | 46.40 ± 10.75 | 53.07 ± 15.08 | 58.80 ± 15.91 | 54.93 ± 16.18 | 1.80 |
| | | NETTACK | 4.00 ± 4.66 | 7.33 ± 7.43 | 6.67 ± 7.00 | 9.87 ± 9.64 | 12.00 ± 11.59 | 7.00 |
| | | PGD | 35.07 ± 16.49 | 41.73 ± 12.98 | 47.47 ± 15.54 | 50.67 ± 14.86 | 49.33 ± 8.71 | 2.80 |
| | | PR-BCD (NA) | 27.73 ± 12.67 | 25.87 ± 12.34 | 38.67 ± 15.36 | 38.53 ± 13.53 | 38.80 ± 11.48 | 4.80 |
| | | SGA | 28.00 ± 8.62 | 35.07 ± 12.60 | 39.07 ± 16.66 | 39.47 ± 14.55 | 42.40 ± 15.50 | 3.80 |
| | | GOttack | 20.40 ± 7.97 | 15.20 ± 13.45 | 23.20 ± 11.18 | 14.53 ± 8.60 | 22.93 ± 9.35 | 5.80 |
| | GSAGE | L¹D-RND | 13.47 ± 18.13 | 34.00 ± 13.33 | **43.07 ± 10.55** | **44.93 ± 11.21** | **49.47 ± 12.11** | 2.20 |
| | | FGA | 28.93 ± 8.21 | **34.40 ± 9.54** | 35.20 ± 10.63 | 39.20 ± 9.37 | 40.80 ± 7.32 | 1.60 |
| | | NETTACK | 8.80 ± 6.54 | 10.13 ± 6.07 | 12.27 ± 6.04 | 11.47 ± 6.21 | 15.73 ± 4.83 | 7.00 |
| | | PGD | 24.80 ± 7.92 | 27.60 ± 6.77 | 30.27 ± 9.56 | 33.07 ± 9.35 | 37.20 ± 11.05 | 3.80 |
| | | PR-BCD (NA) | 20.40 ± 8.11 | 25.60 ± 12.29 | 32.00 ± 16.42 | 30.13 ± 13.21 | 31.07 ± 15.21 | 4.60 |
| | | SGA | 27.60 ± 11.14 | 30.53 ± 13.57 | 32.00 ± 13.05 | 34.13 ± 14.88 | 32.80 ± 10.39 | 3.20 |
| | | GOttack | 24.67 ± 10.24 | 21.20 ± 8.65 | 27.87 ± 8.63 | 21.33 ± 10.68 | 29.07 ± 8.41 | 5.60 |
| Poison | GCN | L¹D-RND | 35.60 ± 22.31 | 56.27 ± 9.85 | 65.20 ± 5.89 | **69.07 ± 7.44** | 68.13 ± 5.88 | 2.80 |
| | | FGA | **66.40 ± 8.25** | 61.47 ± 12.70 | 62.13 ± 13.70 | 60.53 ± 11.75 | 63.07 ± 10.69 | 3.00 |
| | | NETTACK | 7.87 ± 4.98 | 8.13 ± 2.56 | 7.33 ± 5.49 | 7.73 ± 3.77 | 8.40 ± 5.72 | 7.00 |
| | | PGD | 53.47 ± 17.98 | 56.80 ± 15.83 | 64.13 ± 12.64 | 61.87 ± 14.67 | 60.53 ± 14.23 | 3.20 |
| | | PR-BCD (NA) | **64.80 ± 14.69** | **70.80 ± 14.89** | **70.67 ± 8.54** | 67.60 ± 7.57 | **74.93 ± 6.63** | 1.40 |
| | | SGA | 51.47 ± 9.12 | 54.67 ± 8.57 | 60.13 ± 7.65 | 61.87 ± 9.49 | 59.07 ± 7.74 | 4.60 |
| | | GOttack | 27.07 ± 8.48 | 23.73 ± 5.65 | 23.33 ± 7.16 | 24.80 ± 7.04 | 24.67 ± 7.81 | 6.00 |
| | GIN | L¹D-RND | 37.33 ± 18.29 | 61.07 ± 12.16 | 65.20 ± 8.84 | 69.20 ± 10.71 | **68.67 ± 10.49** | 2.60 |
| | | FGA | **61.20 ± 12.32** | **67.33 ± 14.34** | 70.27 ± 12.23 | **71.07 ± 11.85** | 68.40 ± 16.62 | 1.20 |
| | | NETTACK | 20.40 ± 11.27 | 22.27 ± 11.66 | 23.60 ± 12.56 | 25.47 ± 12.97 | 27.87 ± 11.38 | 7.00 |
| | | PGD | 49.33 ± 15.28 | 59.07 ± 13.13 | 64.93 ± 17.05 | 70.27 ± 8.88 | 67.47 ± 8.67 | 2.80 |
| | | PR-BCD (NA) | 50.00 ± 12.58 | 48.40 ± 10.06 | 58.93 ± 9.50 | 59.07 ± 9.19 | 58.40 ± 10.32 | 4.00 |
| | | SGA | 46.27 ± 10.90 | 50.80 ± 11.83 | 57.73 ± 15.32 | 56.13 ± 13.87 | 61.60 ± 12.92 | 4.40 |
| | | GOttack | 35.73 ± 9.50 | 32.40 ± 15.33 | 37.33 ± 12.09 | 31.73 ± 12.46 | 37.87 ± 13.23 | 6.00 |
| | GSAGE | L¹D-RND | 22.93 ± 17.63 | 40.27 ± 9.50 | 44.40 ± 11.62 | 50.27 ± 11.66 | 52.53 ± 13.26 | 3.40 |
| | | FGA | **42.67 ± 14.20** | **45.47 ± 15.13** | **55.20 ± 13.75** | **56.40 ± 14.68** | **58.67 ± 16.64** | 1.00 |
| | | NETTACK | 12.40 ± 7.60 | 12.40 ± 6.01 | 16.13 ± 8.19 | 16.13 ± 8.33 | 20.00 ± 6.46 | 7.00 |
| | | PGD | 36.27 ± 13.24 | 41.60 ± 13.74 | 41.33 ± 14.53 | 47.60 ± 14.99 | 52.67 ± 15.74 | 2.80 |
| | | PR-BCD (NA) | 31.60 ± 10.29 | 38.53 ± 12.64 | 44.67 ± 15.39 | 43.87 ± 12.15 | 44.27 ± 15.51 | 3.80 |
| | | SGA | 35.60 ± 10.93 | 36.53 ± 12.18 | 42.00 ± 13.37 | 44.53 ± 15.97 | 40.67 ± 14.40 | 4.20 |
| | | GOttack | 31.07 ± 10.66 | 23.07 ± 9.74 | 30.80 ± 7.55 | 26.67 ± 10.65 | 33.87 ± 8.99 | 5.80 |

Table 21: **Defended Homophily Results - 1/3.** Evaluating six adversarial attacks on four defense victim models (GNNGuard, GCN-Jaccard, GCN-GARNET ElasticGNN) - Miss-classification rate (%). Higher is better. Best performance in **bold**, second best underlined.

| Dataset | | CORA | | | | | CITESEER | | | | | PUBMED | | | | | Avg. Rank |
|---|---|---|---|---|---|---|---|---|---|---|---|---|---|---|---|---|---|
| Δ → | | 1 | 2 | 3 | 4 | 5 | 1 | 2 | 3 | 4 | 5 | 1 | 2 | 3 | 4 | 5 | |
| **Evasion — ElasticGNN** | L¹D-RND | 12.67±2.47 | 14.80±4.13 | 19.33±3.83 | 20.93±5.55 | 24.40±5.36 | 14.00±5.40 | 19.07±5.50 | 24.53±8.80 | 27.33±9.22 | 32.27±10.14 | 10.67±3.75 | 10.67±4.32 | 12.27±4.71 | 13.87±3.96 | 13.73±4.33 | 7.00 |
| | FGA | 24.67±3.83 | 32.40±4.91 | 37.73±6.96 | 44.00±8.55 | 50.93±6.63 | 22.00±3.30 | 30.13±6.16 | 41.20±9.70 | 46.80±12.60 | 51.07±12.02 | 24.67±6.87 | 26.27±5.44 | 30.80±7.85 | 33.47±8.50 | 35.73±9.13 | 2.60 |
| | NETTACK | 23.47±3.16 | 30.27±5.99 | 43.20±5.60 | 49.87±6.07 | 54.80±5.17 | 22.80±4.26 | 33.87±9.46 | 42.80±11.85 | 46.67±12.71 | 49.87±11.82 | 23.87±5.63 | 26.00±6.28 | 29.87±7.84 | 33.73±9.94 | 34.67±10.13 | 2.87 |
| | PGD | 24.00±4.34 | 30.40±2.75 | 38.67±6.44 | 44.00±7.56 | 47.33±8.09 | 22.93±4.65 | 30.27±7.09 | 37.60±9.01 | 42.53±9.46 | 45.60±9.75 | 24.80±5.89 | 28.00±5.95 | 30.00±5.71 | 30.53±5.26 | 32.27±5.18 | 4.07 |
| | PR-BCD (NA) | 24.67±3.75 | 32.00±5.76 | 41.60±6.85 | 44.80±7.28 | 50.40±9.05 | 23.33±4.76 | 30.93±7.44 | 38.40±9.66 | 45.47±11.12 | 51.33±11.87 | 22.13±3.16 | 25.60±4.42 | 28.00±4.28 | 30.53±5.53 | 30.53±3.89 | 3.67 |
| | SGA | 24.27±4.27 | 30.93±6.32 | 37.87±7.84 | 45.47±9.66 | 49.47±8.43 | 21.60±5.14 | 25.47±4.98 | 32.67±6.40 | 39.47±8.02 | 42.53±8.43 | 24.67±5.98 | 27.60±7.30 | 30.13±7.03 | 33.33±8.37 | 33.20±5.99 | 4.33 |
| | GOttack | 23.60±2.75 | 32.13±4.24 | 37.73±6.18 | 46.27±7.32 | 50.00±5.61 | 22.67±6.58 | 32.53±8.33 | 36.80±12.44 | 43.07±12.94 | 46.40±13.96 | 24.40±5.25 | 28.27±4.77 | 30.53±5.93 | 32.53±6.35 | 33.60±5.72 | 3.47 |
| **Evasion — GCN-GARNET** | L¹D-RND | 16.53±11.75 | 37.60±13.86 | 43.73±13.67 | 46.40±16.29 | 51.07±19.20 | 21.33±13.95 | 46.40±15.35 | 49.73±12.02 | 56.67±14.46 | 57.33±15.87 | 18.93±17.84 | 34.27±19.02 | 33.33±21.93 | 37.20±20.07 | 36.00±20.06 | 2.47 |
| | FGA | 18.27±5.55 | 22.80±5.54 | 30.00±10.17 | 38.93±13.00 | 45.33±14.38 | 16.13±4.17 | 19.73±8.97 | 27.20±10.14 | 38.00±10.74 | 47.87±8.02 | 15.47±8.57 | 17.60±11.29 | 20.93±9.53 | 33.60±14.04 | 38.53±12.08 | 5.20 |
| | NETTACK | 20.67±5.59 | 34.13±11.77 | 46.80±9.62 | 56.27±10.87 | 62.93±9.79 | 21.87±6.35 | 40.93±11.36 | 54.93±9.94 | 58.13±13.21 | 61.73±11.34 | 17.60±7.75 | 33.20±13.50 | 51.33±12.53 | 54.93±11.00 | 61.60±12.12 | 1.47 |
| | PGD | 20.93±6.27 | 30.93±7.40 | 36.53±9.61 | 42.67±10.27 | 49.47±13.21 | 20.93±5.06 | 30.67±6.70 | 31.20±9.16 | 41.73±10.22 | 39.07±11.71 | 14.53±5.83 | 19.07±6.80 | 20.93±9.41 | 21.20±9.65 | 23.87±11.72 | 5.00 |
| | PR-BCD (NA) | 21.73±5.90 | 32.67±7.24 | 43.73±7.25 | 50.40±13.01 | 52.27±11.18 | 25.73±8.03 | 34.93±9.04 | 43.60±9.26 | 50.00±9.97 | 50.13±15.50 | 17.33±8.34 | 24.93±11.85 | 29.33±12.55 | 29.47±10.13 | 32.93±8.24 | 3.00 |
| | SGA | 14.27±4.46 | 20.40±5.08 | 23.73±6.58 | 27.07±7.96 | 29.73±8.48 | 12.93±6.27 | 20.67±8.80 | 25.87±8.86 | 32.00±12.87 | 35.33±13.91 | 13.33±5.69 | 13.87±6.02 | 16.00±5.13 | 18.13±9.43 | 21.33±8.09 | 6.93 |
| | GOttack | 20.00±6.93 | 31.47±9.87 | 42.93±13.22 | 52.40±15.12 | 55.87±11.65 | 21.20±8.91 | 26.80±7.08 | 34.67±13.06 | 34.67±15.30 | 44.27±13.24 | 15.07±9.76 | 23.87±10.38 | 26.93±12.53 | 34.53±13.32 | 36.80±15.76 | 3.93 |
| **Evasion — GCN-Jaccard** | L¹D-RND | 12.00±3.78 | 18.40±5.67 | 24.67±6.79 | 28.53±6.82 | 30.80±6.13 | 11.47±3.66 | 23.33±8.47 | 33.07±5.55 | 34.40±6.73 | 40.67±6.26 | 10.67±2.79 | 13.20±4.06 | 16.27±4.06 | 21.73±4.53 | 25.47±6.07 | 7.00 |
| | FGA | 20.00±4.28 | 35.20±6.75 | 43.47±6.25 | 48.93±4.53 | 56.27±6.18 | 21.33±5.00 | 33.07±4.20 | 39.20±7.55 | 46.93±6.04 | 52.40±4.91 | 31.07±2.49 | 42.13±3.96 | 50.40±4.73 | 54.13±2.88 | 55.60±2.95 | 3.73 |
| | NETTACK | 23.47±3.50 | 37.87±4.93 | 46.53±7.61 | 51.33±8.09 | 55.87±3.66 | 21.20±4.46 | 40.13±8.80 | 45.60±6.10 | 51.87±7.69 | 56.53±5.78 | 30.67±3.18 | 50.40±2.95 | 53.87±3.07 | 56.13±2.67 | 58.93±2.37 | 2.07 |
| | PGD | 24.80±4.59 | 35.20±6.04 | 41.07±5.75 | 44.53±8.16 | 47.60±5.14 | 21.47±5.10 | 32.93±5.90 | 42.67±4.88 | 48.27±4.83 | 49.47±5.88 | 31.07±3.53 | 36.40±2.29 | 42.53±5.42 | 44.13±6.02 | 45.20±3.61 | 4.80 |
| | PR-BCD (NA) | 20.93±5.01 | 33.07±6.76 | 41.47±5.04 | 46.67±4.76 | 49.33±5.74 | 22.53±3.89 | 37.73±4.95 | 47.60±5.96 | 50.93±3.37 | 54.80±3.36 | 32.80±3.69 | 36.93±3.84 | 41.47±3.34 | 45.20±3.84 | 49.73±3.45 | 3.67 |
| | SGA | 20.93±6.09 | 35.60±5.41 | 43.60±4.36 | 49.60±4.22 | 54.93±6.45 | 18.13±5.26 | 28.13±4.50 | 39.87±7.73 | 45.07±6.36 | 51.20±6.04 | 31.33±4.70 | 40.00±4.07 | 47.60±4.79 | 51.07±4.71 | 52.53±7.19 | 4.27 |
| | GOttack | 23.60±4.91 | 39.73±6.04 | 46.93±6.36 | 52.27±7.25 | 54.67±6.40 | 20.93±5.00 | 33.73±8.00 | 43.60±10.06 | 47.20±7.99 | 53.20±8.87 | 31.87±4.50 | 40.80±4.83 | 50.93±4.27 | 56.53±2.67 | 56.00±2.62 | 2.47 |
| **Evasion — GNNGuard** | L¹D-RND | 6.27±4.13 | 9.73±5.60 | 11.73±4.83 | 17.20±5.44 | 18.67±6.13 | 4.67±3.68 | 7.60±5.46 | 9.60±5.30 | 10.53±5.10 | 11.07±5.18 | 6.53±4.63 | 8.27±4.20 | 9.73±4.53 | 10.13±5.68 | 12.40±5.46 | 4.80 |
| | FGA | 6.67±3.44 | 11.87±4.37 | 14.93±4.71 | 17.20±4.71 | 22.13±6.21 | 3.33±3.18 | 6.93±3.69 | 9.07±4.06 | 13.33±5.16 | 13.07±6.04 | 3.60±1.88 | 4.67±2.35 | 7.47±2.97 | 10.27±2.81 | 12.40±3.56 | 5.00 |
| | NETTACK | 6.80±4.77 | 13.07±5.01 | 16.93±6.54 | 20.40±6.68 | 25.07±6.45 | 4.67±2.35 | 9.73±4.27 | 13.47±5.93 | 17.07±5.12 | 20.53±5.78 | 2.93±1.67 | 7.33±3.75 | 10.13±3.50 | 13.87±4.89 | 15.87±3.89 | 2.27 |
| | PGD | 6.80±2.60 | 11.33±4.82 | 16.80±5.28 | 20.40±5.08 | 21.07±4.95 | 3.07±2.49 | 5.33±3.27 | 8.80±4.77 | 11.60±3.12 | 13.07±4.46 | 2.53±1.60 | 5.47±2.07 | 6.80±3.28 | 10.40±2.95 | 11.33±4.05 | 5.73 |
| | PR-BCD (NA) | 8.13±3.34 | 13.33±3.98 | 15.60±4.67 | 17.47±3.81 | 22.80±6.13 | 3.07±2.12 | 6.53±2.33 | 8.93±3.20 | 11.07±4.40 | 15.07±5.23 | 4.27±2.25 | 8.53±3.74 | 12.40±4.01 | 15.33±4.19 | 17.33±3.52 | 3.40 |
| | SGA | 8.40±4.29 | 12.80±3.28 | 18.53±5.04 | 21.07±4.53 | 22.53±5.53 | 4.00±2.73 | 8.00±5.81 | 10.13±4.56 | 13.20±5.06 | 14.40±4.08 | 3.47±1.92 | 6.13±3.07 | 7.33±2.35 | 9.87±4.03 | 11.73±4.27 | 3.60 |
| | GOttack | 8.40±4.97 | 13.20±4.33 | 17.20±4.59 | 19.20±4.89 | 23.20±5.89 | 4.67±2.89 | 6.80±4.77 | 10.13±4.56 | 13.20±5.17 | 15.87±3.81 | 3.07±2.25 | 6.80±3.10 | 9.20±3.00 | 11.20±3.40 | 13.87±3.42 | 3.20 |
| **Poison — ElasticGNN** | L¹D-RND | 14.00±3.70 | 16.40±3.79 | 21.07±4.20 | 23.33±5.49 | 28.00±5.90 | 15.33±4.58 | 21.47±6.48 | 27.20±8.51 | 31.87±9.66 | 36.27±9.38 | 10.53±3.42 | 11.20±4.46 | 12.67±5.05 | 14.13±3.58 | 13.60±4.48 | 7.00 |
| | FGA | 28.67±6.58 | 40.93±11.76 | 50.93±11.68 | 56.53±7.11 | 61.73±4.53 | 28.80±8.51 | 43.60±9.26 | 51.87±7.27 | 57.47±9.90 | 59.60±6.85 | 23.60±7.45 | 26.53±6.30 | 30.13±8.19 | 34.27±9.50 | 36.67±9.88 | 2.33 |
| | NETTACK | 29.20±7.16 | 39.47±6.30 | 51.47±5.73 | 57.47±5.48 | 62.93±6.67 | 30.27±8.51 | 45.87±11.10 | 54.40±7.68 | 54.67±8.20 | 59.60±6.24 | 23.33±5.84 | 26.67±6.91 | 29.60±8.18 | 34.27±11.31 | 36.13±10.24 | 2.27 |
| | PGD | 26.53±3.58 | 38.53±6.70 | 44.27±6.13 | 51.07±5.65 | 54.40±7.17 | 29.73±6.80 | 41.20±8.03 | 46.13±6.35 | 50.80±6.71 | 54.27±6.13 | 23.73±5.95 | 27.07±5.99 | 30.27±6.36 | 31.33±5.79 | 32.27±5.75 | 4.47 |
| | PR-BCD (NA) | 25.60±4.15 | 36.53±5.93 | 47.47±6.25 | 55.33±3.83 | 59.47±5.04 | 29.47±9.21 | 38.93±11.29 | 50.93±11.71 | 53.73±10.22 | 57.73±10.00 | 20.53±3.07 | 25.60±3.94 | 27.07±4.83 | 30.80±6.13 | 30.80±4.06 | 4.93 |
| | SGA | 27.07±4.13 | 40.40±7.18 | 50.40±5.77 | 57.47±6.57 | 60.27±8.00 | 26.40±4.29 | 36.40±5.14 | 47.47±6.16 | 50.80±6.09 | 57.20±5.39 | 23.47±6.52 | 28.40±7.18 | 30.80±8.06 | 32.67±8.44 | 34.27±8.68 | 3.47 |
| | GOttack | 28.67±5.94 | 39.87±8.63 | 47.73±9.13 | 54.40±5.67 | 58.27±5.55 | 30.93±7.21 | 44.27±8.97 | 49.60±7.64 | 55.33±6.66 | 56.27±5.80 | 23.07±6.67 | 26.80±4.83 | 30.27±6.84 | 32.13±6.82 | 33.60±5.41 | 3.53 |
| **Poison — GCN-GARNET** | L¹D-RND | 18.67±9.96 | 38.27±13.89 | 44.80±14.34 | 46.13±16.56 | 58.13±18.55 | 20.27±13.89 | 44.47±14.05 | 51.33±11.88 | 58.13±13.82 | 58.27±15.21 | 18.67±18.46 | 34.53±18.63 | 34.67±21.09 | 36.67±19.42 | 38.67±18.36 | 2.47 |
| | FGA | 20.13±4.56 | 23.60±7.10 | 31.20±10.60 | 38.40±12.81 | 45.73±15.42 | 18.80±5.33 | 21.60±8.29 | 30.67±11.13 | 41.20±13.54 | 49.60±8.59 | 16.80±10.79 | 16.80±9.97 | 19.33±8.09 | 32.40±14.35 | 36.40±12.65 | 5.33 |
| | NETTACK | 22.80±6.41 | 35.60±11.06 | 46.00±10.47 | 56.67±10.73 | 64.40±8.63 | 23.33±6.58 | 43.07±10.22 | 55.20±9.34 | 59.47±11.96 | 64.67±8.51 | 17.47±6.70 | 32.80±13.91 | 50.13±12.15 | 54.27±13.16 | 61.20±11.85 | 1.47 |
| | PGD | 21.87±5.73 | 30.53±7.87 | 37.07±10.19 | 42.93±10.69 | 49.47±13.74 | 24.80±6.27 | 30.27±9.79 | 35.20±9.79 | 41.33±9.79 | 43.87±12.73 | 14.80±5.39 | 18.93±9.41 | 22.40±10.06 | 22.13±9.32 | 24.67±11.48 | 4.80 |
| | PR-BCD (NA) | 22.53±6.02 | 33.60±9.26 | 42.53±9.81 | 50.27±13.52 | 52.67±12.41 | 27.07±8.55 | 37.60±7.90 | 46.67±11.00 | 53.20±10.58 | 52.27±13.62 | 18.00±9.62 | 22.40±10.09 | 29.73±14.48 | 29.73±10.25 | 35.33±9.76 | 3.13 |
| | SGA | 15.87±4.24 | 21.73±5.50 | 25.47±7.11 | 28.67±8.06 | 30.67±10.92 | 15.33±4.32 | 24.93±8.71 | 29.07±10.71 | 36.53±12.82 | 39.20±14.85 | 12.93±3.99 | 13.87±4.98 | 15.73±7.09 | 15.87±7.76 | 22.00±9.53 | 6.93 |
| | GOttack | 21.73±7.17 | 32.13±9.40 | 42.93±13.22 | 53.07±14.56 | 56.27±12.58 | 20.40±8.15 | 31.33±9.15 | 39.07±12.60 | 38.13±15.20 | 49.60±13.94 | 16.53±9.84 | 23.73±10.25 | 26.13±11.22 | 35.07±14.81 | 36.00±15.62 | 3.87 |
| **Poison — GCN-Jaccard** | L¹D-RND | 14.40±5.77 | 19.20±5.99 | 26.40±7.86 | 30.13±7.80 | 30.00±5.35 | 13.20±4.89 | 25.87±10.68 | 34.00±4.60 | 36.27±6.67 | 41.33±6.87 | 11.07±3.61 | 14.13±4.24 | 16.00±4.14 | 22.13±5.15 | 25.20±6.18 | 7.00 |
| | FGA | 23.73±5.80 | 38.13±6.70 | 45.47±4.98 | 53.47±6.16 | 56.13±4.37 | 28.67±7.47 | 41.07±6.13 | 46.13±6.65 | 54.80±7.32 | 56.67±4.64 | 12.00±2.60 | 42.00±4.00 | 50.27±4.13 | 53.20±3.10 | 55.47±2.56 | 3.47 |
| | NETTACK | 26.67±5.79 | 39.47±3.42 | 48.27±6.54 | 53.20±6.22 | 58.40±4.79 | 31.73±8.03 | 43.20±6.54 | 53.07±7.05 | 56.67±6.44 | 59.87±5.15 | 31.60±2.85 | 49.47±3.58 | 54.00±2.83 | 56.13±2.67 | 59.07±2.12 | 1.87 |
| | PGD | 26.13±4.17 | 38.53±5.42 | 43.47±4.93 | 46.27±6.63 | 49.73±6.36 | 28.00±7.60 | 39.33±7.04 | 47.60±5.41 | 51.33±5.54 | 53.20±5.28 | 32.13±3.74 | 38.00±2.00 | 42.93±4.95 | 44.80±5.99 | 45.33±4.51 | 5.00 |
| | PR-BCD (NA) | 22.80±5.12 | 36.53±4.10 | 42.53±5.26 | 48.67±4.32 | 50.67±5.11 | 29.60±5.25 | 44.00±5.66 | 52.13±4.81 | 54.13±4.31 | 58.93±4.20 | 31.60±3.14 | 37.20±3.53 | 41.73±3.45 | 46.13±3.34 | 50.67±3.52 | 4.47 |
| | SGA | 26.80±7.63 | 38.93±4.40 | 47.07±4.95 | 53.47±5.10 | 56.80±6.45 | 23.87±4.40 | 38.80±5.89 | 46.93±6.32 | 51.87±4.44 | 57.60±4.67 | 32.00±5.71 | 40.13±4.24 | 47.73±3.99 | 51.07±5.29 | 53.07±6.54 | 3.87 |
| | GOttack | 27.73±6.96 | 42.00±5.18 | 48.40±4.55 | 54.53±5.58 | 55.07±5.28 | 32.00±7.56 | 40.40±6.15 | 50.67±10.76 | 52.93±6.50 | 56.53±8.63 | 32.27±4.83 | 41.33±3.90 | 51.07±4.20 | 55.60±2.53 | 56.67±2.23 | 2.33 |
| **Poison — GNNGuard** | L¹D-RND | 6.93±4.40 | 10.40±5.14 | 12.80±3.76 | 18.00±4.90 | 19.47±5.88 | 4.80±3.84 | 7.87±5.53 | 9.87±5.21 | 10.53±4.93 | 11.47±4.69 | 6.53±4.93 | 8.00±4.21 | 9.33±3.90 | 10.40±5.96 | 12.53±5.48 | 5.33 |
| | FGA | 7.47±3.96 | 12.53±4.50 | 16.80±4.89 | 19.20±4.65 | 24.80±5.60 | 4.40±3.31 | 9.07±3.77 | 10.93±3.61 | 14.40±4.65 | 14.27±6.13 | 4.80±3.19 | 5.73±3.10 | 8.93±3.01 | 11.60±2.41 | 13.20±4.26 | 4.53 |
| | NETTACK | 7.47±5.37 | 14.93±5.55 | 19.20±6.67 | 22.13±6.30 | 28.40±5.87 | 6.00±2.93 | 10.53±4.24 | 14.67±6.03 | 19.33±6.62 | 22.53±5.83 | 4.40±2.53 | 8.67±3.75 | 10.53±4.17 | 13.87±4.69 | 15.73±3.69 | 2.20 |
| | PGD | 6.93±3.01 | 12.40±5.08 | 17.47±5.10 | 21.07±5.39 | 22.00±5.81 | 3.20±2.11 | 6.80±3.28 | 9.87±4.98 | 13.20±3.19 | 14.67±4.51 | 4.27±2.12 | 6.53±3.25 | 7.87±3.16 | 12.00±3.02 | 11.87±4.17 | 5.80 |
| | PR-BCD (NA) | 8.93±3.99 | 14.00±4.07 | 17.07±5.75 | 19.07±3.99 | 23.87±7.27 | 3.20±1.97 | 7.47±2.67 | 10.27±3.53 | 12.53±5.48 | 17.87±5.97 | 5.60±4.08 | 9.60±4.67 | 13.33±4.19 | 16.67±3.75 | 18.40±3.31 | 3.67 |
| | SGA | 9.60±3.64 | 15.33±3.75 | 20.80±5.89 | 25.07±5.95 | 28.13±7.11 | 4.80±3.00 | 8.93±6.04 | 10.67±4.88 | 15.33±5.64 | 16.93±4.83 | 4.93±3.20 | 8.00±3.85 | 8.27±2.49 | 11.33±3.75 | 12.53±3.81 | 3.20 |
| | GOttack | 10.27±6.54 | 15.07±5.01 | 19.07±4.27 | 22.27±6.18 | 26.13±5.78 | 4.80±2.70 | 7.87±4.44 | 10.80±4.71 | 14.27±5.01 | 17.20±4.52 | 4.93±3.10 | 7.87±3.66 | 9.87±3.66 | 12.00±3.21 | 14.40±3.64 | 3.27 |

Table 22: **Defended Homophily Results - 2/3.** Evaluating six adversarial attacks on four defense victim models (GRAND, RobustGCN, GCORN, RUNG) - Miss-classification rate (%). Higher is better. Best performance in **bold**, second best underlined. OOR entries are omitted in average rank calculation.

| | | Dataset | CORA | | | | | CITESEER | | | | | PUBMED | | | | | Avg. Rank |
|---|---|---|---|---|---|---|---|---|---|---|---|---|---|---|---|---|---|---|
| | | Δ → | 1 | 2 | 3 | 4 | 5 | 1 | 2 | 3 | 4 | 5 | 1 | 2 | 3 | 4 | 5 | |
| **Evasion** | GRAND | L¹D-RND | 34.13±14.99 | 63.47±7.54 | 71.07±6.54 | 71.07±7.70 | 72.00±8.72 | 45.47±11.80 | 60.67±8.64 | 63.33±9.43 | 66.00±8.49 | 66.40±8.72 | 33.20±15.62 | 39.07±17.07 | 44.53±16.41 | 44.93±16.78 | 44.67±17.22 | 1.13 |
| | | FGA | 11.87±4.87 | 14.40±4.85 | 16.67±5.79 | 30.40±8.92 | 36.80±5.17 | 13.87±4.93 | 18.80±4.39 | 24.27±5.95 | 34.93±10.44 | 45.73±8.03 | 17.20±5.94 | 20.13±3.25 | 20.40±3.56 | 30.13±8.40 | 39.33±8.67 | 5.13 |
| | | NETTACK | 12.67±5.64 | 30.00±6.14 | 39.87±5.10 | 48.13±5.15 | 51.73±4.53 | 16.00±4.21 | 34.13±7.46 | 47.33±9.70 | 54.53±9.81 | 57.73±8.84 | 19.20±5.65 | 24.00±8.59 | 44.00±5.76 | 47.87±7.69 | 49.87±9.15 | 2.40 |
| | | PGD | 13.73±4.83 | 17.73±5.34 | 20.40±5.96 | 24.80±7.32 | 29.33±5.89 | 16.40±5.51 | 24.40±5.62 | 28.13±5.37 | 29.73±5.28 | 33.33±5.94 | 17.20±3.28 | 20.00±3.70 | 22.13±3.16 | 24.67±3.60 | 26.00±3.30 | 5.20 |
| | | PR-BCD (NA) | 14.40±4.36 | 21.73±5.12 | 27.20±6.09 | 30.00±6.97 | 33.87±7.42 | 17.20±5.75 | 28.80±5.06 | 34.40±6.33 | 39.60±6.64 | 41.33±5.49 | 19.87±3.66 | 23.87±2.20 | 26.00±4.00 | 27.87±4.17 | 30.40±4.73 | 3.27 |
| | | SGA | 14.13±4.98 | 17.47±5.58 | 19.60±5.51 | 21.20±6.96 | 24.27±6.63 | 18.67±5.69 | 27.47±9.90 | 29.47±7.46 | 34.80±7.08 | 42.67±8.34 | 18.27±5.34 | 21.87±3.58 | 24.27±4.59 | 26.40±5.30 | 29.87±7.54 | 4.40 |
| | | GOttack | 12.93±5.70 | 14.93±5.60 | 17.60±4.73 | 19.20±4.46 | 21.07±5.70 | 15.73±4.20 | 18.80±5.65 | 25.20±3.91 | 26.27±4.71 | 28.53±5.97 | 17.87±6.02 | 20.00±4.07 | 19.60±3.87 | 22.27±3.53 | 22.13±4.63 | 6.47 |
| | RobustGCN | L¹D-RND | 15.60±4.15 | 29.60±9.48 | 32.80±3.91 | 33.87±5.48 | 35.07±4.77 | 10.67±3.98 | 29.73±11.05 | 32.53±7.87 | 34.40±9.33 | 42.27±5.90 | 9.20±4.89 | 20.67±3.68 | 23.60±7.45 | 26.67±6.08 | 27.33±5.84 | 7.00 |
| | | FGA | 27.07±3.77 | 45.87±7.15 | 54.13±5.63 | 60.40±4.73 | 63.20±4.77 | 23.87±3.07 | 41.33±7.77 | 51.47±3.96 | 56.80±3.69 | 57.73±3.20 | 36.80±4.71 | 47.73±6.18 | 54.80±5.00 | 58.00±2.00 | 58.67±1.45 | 3.00 |
| | | NETTACK | 28.40±5.25 | 52.13±4.44 | 57.73±2.91 | 60.13±3.50 | 61.33±3.60 | 25.73±6.96 | 45.60±8.18 | 54.27±2.81 | 56.27±4.13 | 58.13±3.81 | 36.00±2.27 | 53.20±3.84 | 57.60±2.03 | 58.80±1.47 | 57.73±1.67 | 2.07 |
| | | PGD | 29.47±3.16 | 46.80±3.84 | 54.93±4.06 | 57.73±4.83 | 60.40±3.72 | 24.40±4.61 | 40.27±7.63 | 49.33±5.79 | 52.40±7.83 | 55.87±4.44 | 35.33±4.19 | 41.60±4.73 | 44.67±5.64 | 45.87±6.12 | 47.60±5.41 | 4.73 |
| | | PR-BCD (NA) | 32.40±4.79 | 58.00±3.85 | 62.00±3.93 | 64.67±3.44 | 64.93±2.49 | 23.87±3.74 | 42.67±5.98 | 53.20±4.89 | 55.87±3.25 | 58.27±3.53 | 31.47±2.07 | 38.40±4.61 | 48.53±3.42 | 54.00±3.12 | 57.07±2.25 | 3.20 |
| | | SGA | 26.93±3.84 | 43.07±10.58 | 55.20±6.09 | 59.47±5.21 | 60.00±5.90 | 20.13±6.63 | 32.00±6.63 | 39.47±9.66 | 48.27±6.41 | 52.80±5.39 | 34.93±3.45 | 42.67±5.27 | 50.53±6.91 | 54.80±3.61 | 57.20±3.10 | 5.13 |
| | | GOttack | 27.47±4.75 | 50.00±3.78 | 56.40±4.48 | 57.73±3.37 | 61.47±3.34 | 25.07±4.13 | 43.47±12.48 | 49.60±8.69 | 56.67±4.45 | 58.40±2.64 | 36.13±5.04 | 46.00±7.71 | 56.13±2.88 | 57.33±1.80 | 57.73±1.83 | 2.87 |
| | GCORN | L¹D-RND | 28.53±11.22 | 32.67±11.43 | 34.27±8.61 | 31.33±7.47 | 30.80±7.36 | 26.67±6.53 | 28.00±8.88 | 26.80±7.00 | 29.33±5.16 | 33.47±6.70 | OOR | OOR | OOR | OOR | OOR | 2.27 |
| | | FGA | 19.47±4.50 | 23.20±4.65 | 26.53±3.42 | 29.60±6.20 | 31.60±6.51 | 15.87±4.31 | 21.47±4.69 | 23.20±5.85 | 28.67±7.20 | 31.33±7.32 | OOR | OOR | OOR | OOR | OOR | 4.53 |
| | | NETTACK | 22.53±5.32 | 31.60±6.33 | 35.20±7.96 | 38.40±8.36 | 47.20±11.48 | 18.53±5.42 | 29.07±8.81 | 31.33±9.96 | 33.20±10.52 | 38.27±13.11 | OOR | OOR | OOR | OOR | OOR | 2.00 |
| | | PGD | 21.33±5.69 | 24.93±5.95 | 24.67±4.45 | 26.67±5.54 | 33.47±7.42 | 16.80±3.36 | 23.73±5.23 | 23.87±6.44 | 26.13±5.97 | 26.67±6.31 | OOR | OOR | OOR | OOR | OOR | 4.93 |
| | | PR-BCD (NA) | 21.33±5.00 | 29.60±4.42 | 28.27±6.18 | 32.00±7.17 | 40.27±8.34 | 18.40±5.87 | 26.67±5.49 | 27.60±5.96 | 32.67±6.44 | 35.33±5.59 | OOR | OOR | OOR | OOR | OOR | 4.00 |
| | | SGA | 20.13±5.97 | 22.80±4.65 | 23.07±5.18 | 23.87±5.21 | 26.67±5.33 | 21.07±5.23 | 21.33±4.88 | 24.13±5.32 | 24.40±6.38 | 23.73±4.83 | OOR | OOR | OOR | OOR | OOR | 6.13 |
| | | GOttack | 22.53±5.26 | 28.67±6.70 | 32.93±10.53 | 34.80±8.20 | 45.47±10.54 | 19.07±6.32 | 27.87±6.25 | 27.87±6.65 | 32.80±6.20 | 35.87±9.12 | OOR | OOR | OOR | OOR | OOR | 4.13 |
| | RUNG | L¹D-RND | 22.80±8.31 | 25.87±5.97 | 32.53±8.57 | 31.33±7.47 | 30.40±7.34 | 23.33±4.39 | 23.07±4.33 | 25.87±4.50 | 29.33±5.16 | 33.47±6.70 | OOR | OOR | OOR | OOR | OOR | 2.13 |
| | | FGA | 18.40±4.73 | 21.73±5.18 | 24.93±4.06 | 29.20±6.62 | 33.20±7.81 | 16.27±3.77 | 18.67±5.11 | 24.53±7.80 | 29.07±5.95 | 32.53±5.93 | OOR | OOR | OOR | OOR | OOR | 3.80 |
| | | NETTACK | 19.20±5.23 | 28.40±5.14 | 34.13±8.05 | 40.13±6.07 | 42.67±8.90 | 21.20±6.54 | 22.80±7.99 | 30.67±7.00 | 32.40±9.75 | 38.93±8.78 | OOR | OOR | OOR | OOR | OOR | 2.00 |
| | | PGD | 17.33±4.88 | 19.33±3.75 | 23.87±4.24 | 25.87±5.53 | 29.73±6.67 | 17.20±4.65 | 20.53±4.37 | 22.13±5.04 | 24.40±4.67 | 25.47±6.44 | OOR | OOR | OOR | OOR | OOR | 5.00 |
| | | PR-BCD (NA) | 18.27±3.37 | 24.53±5.10 | 28.80±7.55 | 31.87±7.69 | 33.20±7.59 | 18.93±3.61 | 24.80±3.36 | 26.67±5.94 | 29.73±6.50 | 33.60±4.55 | OOR | OOR | OOR | OOR | OOR | 3.67 |
| | | SGA | 14.93±4.20 | 17.60±3.14 | 21.33±5.22 | 22.13±6.21 | 24.40±4.73 | 16.27±4.53 | 19.20±4.39 | 20.40±5.51 | 21.87±5.42 | 22.27±5.12 | OOR | OOR | OOR | OOR | OOR | 6.47 |
| | | GOttack | 19.60±6.64 | 26.80±5.33 | 30.00±8.07 | 30.00±8.00 | 39.33±8.33 | 17.07±6.54 | 22.67±7.39 | 24.40±7.97 | 31.20±8.24 | 36.13±7.03 | OOR | OOR | OOR | OOR | OOR | 4.93 |
| **Poison** | GRAND | L¹D-RND | 36.80±10.84 | 64.00±9.97 | 69.87±7.27 | 67.33±7.04 | 66.13±8.47 | 46.13±11.45 | 61.47±7.46 | 64.27±8.31 | 64.13±7.91 | 64.67±8.16 | 34.27±13.92 | 40.40±16.79 | 46.27±13.35 | 45.73±14.12 | 45.20±12.30 | 1.13 |
| | | FGA | 16.93±4.71 | 19.33±6.26 | 23.07±7.44 | 36.40±8.29 | 42.00±5.61 | 17.87±3.34 | 25.07±4.13 | 30.53±5.58 | 43.73±10.44 | 52.53±8.70 | 17.33±5.79 | 19.73±5.12 | 20.53±4.69 | 28.40±9.39 | 38.80±9.25 | 5.07 |
| | | NETTACK | 18.93±5.39 | 33.60±6.77 | 44.00±8.00 | 49.47±6.25 | 54.27±7.21 | 20.67±4.82 | 36.93±6.58 | 51.47±10.51 | 60.53±8.90 | 63.60±8.66 | 18.27±5.55 | 22.53±9.18 | 41.20±10.98 | 48.27±8.03 | 49.60±8.04 | 2.20 |
| | | PGD | 18.67±5.27 | 22.00±6.65 | 26.40±8.01 | 32.13±8.16 | 35.73±8.58 | 19.20±3.99 | 27.07±5.23 | 32.27±4.89 | 35.07±6.50 | 39.20±6.36 | 15.87±4.03 | 18.93±3.99 | 20.93±6.58 | 21.87±5.88 | 23.73±5.60 | 5.53 |
| | | PR-BCD (NA) | 19.07±4.89 | 25.20±5.89 | 33.47±5.88 | 37.33±7.43 | 41.60±9.23 | 20.67±4.94 | 32.67±5.59 | 39.47±5.53 | 44.80±7.66 | 50.00±6.32 | 21.60±7.68 | 23.60±5.14 | 25.73±5.55 | 27.33±7.24 | 30.00±6.14 | 3.07 |
| | | SGA | 18.80±5.12 | 21.07±9.59 | 23.47±5.15 | 28.13±5.83 | 31.20±8.20 | 20.53±5.48 | 27.73±7.81 | 34.53±8.16 | 39.47±8.40 | 48.13±9.33 | 18.67±7.47 | 23.07±6.36 | 24.93±6.32 | 26.93±6.54 | 29.33±8.84 | 4.53 |
| | | GOttack | 16.27±4.53 | 20.40±4.48 | 23.07±6.23 | 25.33±6.70 | 28.40±5.82 | 19.73±4.89 | 23.33±5.33 | 26.93±5.12 | 31.47±4.87 | 34.53±6.21 | 17.87±6.91 | 18.80±5.28 | 21.73±6.63 | 23.20±6.49 | 22.93±5.85 | 6.47 |
| | RobustGCN | L¹D-RND | 16.40±3.48 | 30.80±9.91 | 32.93±4.20 | 33.33±5.59 | 36.00±4.54 | 11.33±3.90 | 29.73±11.78 | 33.20±7.99 | 35.47±8.83 | 42.93±5.60 | 9.20±4.89 | 20.67±3.83 | 24.13±6.82 | 26.80±6.27 | 27.20±5.80 | 7.00 |
| | | FGA | 30.93±4.77 | 47.07±7.28 | 55.47±4.87 | 60.93±3.37 | 63.20±3.10 | 31.73±6.13 | 51.47±5.21 | 58.67±2.35 | 59.73±2.71 | 61.07±4.33 | 36.80±4.59 | 47.87±5.53 | 55.20±4.46 | 57.87±2.07 | 58.53±1.60 | 2.53 |
| | | NETTACK | 32.93±5.70 | 54.40±4.15 | 58.93±3.28 | 60.80±3.84 | 62.00±4.00 | 32.53±7.50 | 52.67±4.32 | 56.67±2.79 | 59.07±3.28 | 61.07±2.91 | 35.47±3.16 | 53.47±3.58 | 57.33±2.09 | 58.67±1.45 | 57.60±1.88 | 2.27 |
| | | PGD | 32.93±4.40 | 48.53±5.32 | 55.33±2.99 | 58.40±3.87 | 60.27±3.77 | 31.33±6.44 | 50.27±6.09 | 53.47±6.02 | 56.13±4.93 | 59.20±3.99 | 35.33±4.19 | 42.27±5.12 | 45.33±5.11 | 46.40±6.64 | 47.33±5.22 | 5.07 |
| | | PR-BCD (NA) | 34.40±4.08 | 57.20±3.19 | 61.73±3.37 | 64.13±2.97 | 65.33±2.99 | 25.87±5.49 | 49.60±6.56 | 56.00±4.47 | 57.47±4.47 | 60.00±4.07 | 31.13±2.27 | 38.93±4.89 | 49.47±3.50 | 54.27±2.12 | 56.93±2.12 | 3.53 |
| | | SGA | 30.27±5.50 | 47.73±7.70 | 57.07±4.59 | 60.67±4.49 | 62.13±5.63 | 23.33±6.22 | 42.67±7.88 | 49.33±7.51 | 54.00±5.40 | 58.53±5.32 | 34.53±3.74 | 43.20±4.46 | 51.60±6.94 | 55.20±3.76 | 57.33±2.23 | 4.93 |
| | | GOttack | 33.20±5.89 | 50.27±4.53 | 58.00±3.38 | 58.80±2.91 | 62.27±3.92 | 31.47±9.02 | 50.93±8.58 | 54.67±5.84 | 60.00±3.63 | 61.33±2.35 | 35.73±4.65 | 46.27±7.09 | 56.00±3.02 | 57.47±1.60 | 57.87±1.92 | 2.67 |
| | GCORN | L¹D-RND | 28.93±11.11 | 32.80±11.31 | 34.27±8.61 | 31.47±7.39 | 30.67±7.28 | 26.80±6.75 | 28.53±8.96 | 27.07±6.92 | 29.47±4.98 | 33.47±6.70 | OOR | OOR | OOR | OOR | OOR | 2.47 |
| | | FGA | 19.87±3.96 | 23.87±4.56 | 26.93±3.53 | 30.27±5.90 | 32.27±6.58 | 17.20±4.20 | 22.80±4.20 | 25.33±6.58 | 30.40±5.30 | 34.80±6.88 | OOR | OOR | OOR | OOR | OOR | 4.33 |
| | | NETTACK | 22.80±5.28 | 32.80±5.60 | 35.87±7.35 | 39.20±8.06 | 48.27±11.31 | 18.53±5.78 | 30.00±8.72 | 32.40±9.95 | 35.20±10.82 | 40.00±12.60 | OOR | OOR | OOR | OOR | OOR | 2.07 |
| | | PGD | 21.60±5.08 | 25.47±4.98 | 25.07±4.71 | 27.33±5.27 | 33.60±7.49 | 17.73±3.84 | 24.27±5.70 | 24.93±5.95 | 27.73±5.44 | 28.80±5.99 | OOR | OOR | OOR | OOR | OOR | 5.00 |
| | | PR-BCD (NA) | 21.73±5.39 | 30.00±4.84 | 28.80±6.13 | 32.53±6.86 | 40.53±8.37 | 18.93±5.50 | 27.33±5.49 | 29.73±5.95 | 33.33±5.94 | 36.40±4.97 | OOR | OOR | OOR | OOR | OOR | 3.80 |
| | | SGA | 21.07±6.41 | 24.00±4.14 | 24.00±5.18 | 24.67±5.64 | 28.00±4.72 | 21.73±5.23 | 21.87±4.98 | 24.80±5.12 | 25.47±6.82 | 26.00±5.45 | OOR | OOR | OOR | OOR | OOR | 6.20 |
| | | GOttack | 22.67±5.38 | 30.27±6.32 | 33.73±10.77 | 35.33±8.16 | 46.40±10.12 | 18.93±6.09 | 29.33±7.51 | 28.00±6.99 | 33.33±8.37 | 37.33±9.40 | OOR | OOR | OOR | OOR | OOR | 4.13 |
| | RUNG | L¹D-RND | 22.80±8.38 | 26.27±5.95 | 32.53±8.16 | 31.47±7.39 | 30.40±7.22 | 23.33±4.39 | 23.33±4.19 | 26.00±4.54 | 29.47±4.98 | 33.47±6.70 | OOR | OOR | OOR | OOR | OOR | 2.47 |
| | | FGA | 19.33±4.39 | 22.00±5.50 | 24.93±4.40 | 29.73±6.63 | 34.13±7.69 | 16.80±4.06 | 20.80±4.33 | 25.07±7.03 | 30.40±5.30 | 34.93±6.04 | OOR | OOR | OOR | OOR | OOR | 3.67 |
| | | NETTACK | 19.33±4.51 | 28.80±4.71 | 35.20±7.70 | 41.33±5.64 | 43.73±9.13 | 21.07±6.76 | 24.13±8.05 | 32.00±7.60 | 33.73±9.56 | 40.80±9.13 | OOR | OOR | OOR | OOR | OOR | 1.93 |
| | | PGD | 17.33±4.64 | 19.73±3.84 | 24.27±4.06 | 26.27±5.75 | 30.80±6.84 | 17.73±5.12 | 21.33±4.25 | 23.73±4.06 | 25.87±4.24 | 27.60±5.96 | OOR | OOR | OOR | OOR | OOR | 4.93 |
| | | PR-BCD (NA) | 18.67±3.60 | 25.07±5.44 | 28.80±7.48 | 32.13±7.42 | 34.40±7.18 | 19.33±3.68 | 25.33±3.18 | 28.00±6.09 | 31.33±5.89 | 35.33±4.76 | OOR | OOR | OOR | OOR | OOR | 3.60 |
| | | SGA | 14.93±4.59 | 18.27±3.20 | 21.73±5.39 | 23.20±6.96 | 26.13±4.44 | 16.93±4.46 | 20.53±4.03 | 21.60±4.91 | 23.20±5.80 | 24.53±5.21 | OOR | OOR | OOR | OOR | OOR | 6.47 |
| | | GOttack | 19.33±6.79 | 27.07±5.34 | 30.40±8.29 | 35.87±7.50 | 41.07±8.38 | 17.47±6.70 | 23.33±7.39 | 27.60±7.26 | 32.93±8.31 | 37.47±7.03 | OOR | OOR | OOR | OOR | OOR | 4.93 |

Table 23: **Defended Homophily Results - 3/3.** Evaluating six adversarial attacks on four defense NoisyGNN - Miss-classification rate (%). Higher is better. Best performance in **bold**, second best underlined.

| Dataset | CORA | | | | | CITESEER | | | | | PUBMED | | | | | Avg. Rank |
|---|---|---|---|---|---|---|---|---|---|---|---|---|---|---|---|---|
| $\Delta \rightarrow$ | 1 | 2 | 3 | 4 | 5 | 1 | 2 | 3 | 4 | 5 | 1 | 2 | 3 | 4 | 5 | |
| **Evasion NoisyGNN** | | | | | | | | | | | | | | | | |
| L¹D-RND | **28.40 ± 10.56** | 31.60 ± 12.05 | 38.80 ± 7.88 | 40.80 ± 8.61 | 41.20 ± 9.06 | **35.60 ± 11.96** | 36.00 ± 11.81 | 40.27 ± 8.58 | 42.93 ± 6.36 | 45.20 ± 5.33 | **31.07 ± 15.43** | 29.20 ± 11.13 | 28.80 ± 8.13 | 25.73 ± 6.36 | 26.40 ± 6.38 | 3.73 |
| FGA | 20.93 ± 4.06 | 25.07 ± 6.84 | 32.00 ± 5.01 | 38.67 ± 6.03 | 42.00 ± 8.42 | 22.13 ± 3.42 | 25.87 ± 5.78 | 34.00 ± 9.23 | 42.80 ± 11.75 | 46.93 ± 11.63 | 0.00 ± 0.00 | 0.00 ± 0.00 | 0.00 ± 0.00 | 0.00 ± 0.00 | 0.00 ± 0.00 | 5.93 |
| NETTACK | 23.33 ± 5.00 | 35.87 ± 8.40 | **50.27 ± 8.55** | **60.27 ± 9.59** | **69.07 ± 9.88** | 26.93 ± 6.96 | **40.67 ± 11.78** | **49.87 ± 10.13** | 53.20 ± 13.79 | **60.67 ± 11.82** | 21.20 ± 5.12 | **34.53 ± 4.75** | **46.27 ± 7.28** | **52.53 ± 7.42** | **60.53 ± 3.07** | 1.60 |
| PGD | 24.67 ± 4.88 | 31.47 ± 4.75 | 40.67 ± 8.57 | 44.93 ± 8.94 | 49.47 ± 8.23 | 24.93 ± 4.06 | 34.00 ± 7.48 | 40.67 ± 8.09 | 44.80 ± 9.19 | 48.93 ± 8.58 | 0.00 ± 0.00 | 0.00 ± 0.00 | 0.00 ± 0.00 | 0.00 ± 0.00 | 0.00 ± 0.00 | 5.13 |
| PR-BCD | 25.07 ± 3.77 | **36.13 ± 6.57** | 47.73 ± 6.67 | 52.53 ± 6.70 | 59.73 ± 9.13 | 26.27 ± 3.69 | 38.00 ± 4.72 | 48.27 ± 8.03 | **54.13 ± 6.16** | 57.73 ± 6.09 | 18.40 ± 3.64 | 25.73 ± 3.28 | 29.87 ± 3.58 | 33.07 ± 3.53 | 36.53 ± 4.03 | 2.73 |
| SGA | 20.93 ± 5.60 | 22.27 ± 5.01 | 25.47 ± 4.44 | 30.27 ± 4.33 | 33.33 ± 6.22 | 28.93 ± 7.09 | 22.67 ± 2.89 | 25.47 ± 4.03 | 27.87 ± 4.31 | 30.40 ± 5.46 | 18.67 ± 3.44 | 21.47 ± 3.96 | 23.33 ± 3.35 | 24.67 ± 5.54 | 25.87 ± 4.10 | 5.93 |
| GOttack | 22.80 ± 6.22 | 31.60 ± 7.30 | 42.13 ± 12.61 | 52.93 ± 12.28 | 61.47 ± 10.84 | 24.00 ± 5.90 | 35.73 ± 9.04 | 48.53 ± 10.68 | 52.40 ± 10.78 | 51.60 ± 9.51 | 23.07 ± 3.69 | 30.00 ± 5.18 | 36.80 ± 7.55 | 41.87 ± 7.84 | 48.13 ± 9.49 | 2.93 |
| **Poison NoisyGNN** | | | | | | | | | | | | | | | | |
| L¹D-RND | **29.07 ± 10.69** | 32.27 ± 11.58 | 41.33 ± 8.61 | 43.07 ± 8.94 | 42.93 ± 9.00 | **35.33 ± 11.46** | 37.60 ± 12.22 | 41.47 ± 8.40 | 44.13 ± 6.39 | 46.00 ± 5.71 | **31.07 ± 15.10** | 29.07 ± 11.11 | 29.07 ± 8.21 | 25.73 ± 6.36 | 26.00 ± 6.46 | 4.20 |
| FGA | 23.20 ± 5.23 | 30.67 ± 9.15 | 38.13 ± 7.91 | 46.67 ± 10.02 | 51.60 ± 11.81 | 24.00 ± 4.21 | 32.40 ± 8.39 | 42.93 ± 8.61 | 51.47 ± 10.07 | 56.40 ± 9.80 | 0.00 ± 0.00 | 0.00 ± 0.00 | 0.00 ± 0.00 | 0.00 ± 0.00 | 0.00 ± 0.00 | 5.73 |
| NETTACK | **27.60 ± 4.67** | **44.27 ± 7.05** | **57.47 ± 7.39** | **64.67 ± 9.25** | **73.60 ± 7.34** | 30.27 ± 6.54 | **50.40 ± 10.70** | **57.33 ± 7.35** | **60.67 ± 10.38** | **65.47 ± 8.37** | 21.20 ± 5.00 | **34.27 ± 4.83** | **46.80 ± 7.20** | **52.93 ± 6.80** | **60.53 ± 2.88** | 1.33 |
| PGD | 27.07 ± 4.71 | 35.47 ± 5.48 | 46.27 ± 9.32 | 50.40 ± 9.48 | 56.67 ± 8.51 | 28.80 ± 6.75 | 40.13 ± 7.54 | 48.53 ± 9.12 | 52.00 ± 7.82 | 55.47 ± 7.46 | 0.00 ± 0.00 | 0.00 ± 0.00 | 0.00 ± 0.00 | 0.00 ± 0.00 | 0.00 ± 0.00 | 5.00 |
| PR-BCD | 26.27 ± 3.20 | 40.27 ± 8.03 | 52.13 ± 6.65 | 60.67 ± 7.24 | 64.80 ± 7.66 | 29.60 ± 4.91 | 44.67 ± 7.16 | 54.67 ± 6.40 | 59.87 ± 7.35 | 61.60 ± 6.98 | 18.27 ± 3.77 | 25.87 ± 3.07 | 30.40 ± 3.56 | 33.60 ± 3.79 | 37.07 ± 5.12 | 2.80 |
| SGA | 23.33 ± 4.82 | 27.73 ± 6.04 | 30.40 ± 7.72 | 38.53 ± 7.73 | 41.73 ± 7.05 | 30.80 ± 6.45 | 28.93 ± 4.77 | 34.80 ± 9.19 | 37.07 ± 6.32 | 40.53 ± 8.63 | 18.27 ± 3.37 | 21.47 ± 4.17 | 22.93 ± 3.69 | 24.67 ± 5.27 | 25.60 ± 4.61 | 5.93 |
| GOttack | 26.00 ± 6.76 | 36.53 ± 9.36 | 46.27 ± 11.36 | 60.00 ± 10.80 | 66.93 ± 9.28 | 27.47 ± 4.63 | 41.47 ± 9.69 | 54.53 ± 10.07 | 58.67 ± 7.28 | 59.33 ± 9.31 | 23.07 ± 3.84 | 30.27 ± 5.12 | 37.20 ± 7.66 | 42.27 ± 8.31 | 49.20 ± 9.06 | 3.00 |

Table 24: **Defended Heterophily Results - 1/3.** Evaluating six adversarial attacks on two defense victim models (GCN-GARNET, ElasticGNN, GNNGuard) - Miss-classification rate (%). Higher is better. Best performance in **bold**, second best underlined. GCN-Jaccard is excluded from this table due to an error caused by GCN-Jaccard's implementation.

| | | Dataset | SQUIRREL | | | | | CHAMELEON | | | | | Avg. Rank |
|---|---|---|---|---|---|---|---|---|---|---|---|---|---|
| | | Δ → | 1 | 2 | 3 | 4 | 5 | 1 | 2 | 3 | 4 | 5 | |
| Evasion | ElasticGNN | L¹D-RND | 7.87 ± 4.56 | 9.33 ± 4.32 | 15.33 ± 4.39 | 18.40 ± 4.01 | 19.87 ± 4.37 | 10.53 ± 9.78 | 16.67 ± 8.97 | 22.00 ± 11.11 | 25.60 ± 12.52 | 31.73 ± 10.77 | 5.40 |
| | | FGA | 23.47 ± 11.45 | **32.93 ± 8.75** | **35.73 ± 9.47** | **40.53 ± 8.53** | **42.67 ± 10.41** | 15.07 ± 10.02 | 21.20 ± 9.70 | 25.33 ± 9.31 | 28.13 ± 7.61 | 30.13 ± 7.58 | 1.70 |
| | | NETTACK | 7.87 ± 3.96 | 12.00 ± 4.72 | 16.53 ± 5.37 | 20.40 ± 4.73 | 22.80 ± 5.06 | 4.00 ± 3.55 | 5.33 ± 3.44 | 7.33 ± 3.35 | 8.13 ± 3.58 | 8.67 ± 4.25 | 6.40 |
| | | PGD | 18.93 ± 9.88 | 25.07 ± 10.74 | 29.60 ± 9.48 | 28.93 ± 10.71 | 31.60 ± 11.72 | 14.93 ± 8.03 | 19.07 ± 7.48 | 22.13 ± 7.27 | 27.20 ± 5.12 | 27.87 ± 6.70 | 3.10 |
| | | PR-BCD (NA) | 12.13 ± 3.81 | 15.73 ± 5.70 | 18.13 ± 5.83 | 19.33 ± 6.13 | 20.13 ± 6.21 | 12.27 ± 7.59 | 14.40 ± 8.85 | 16.67 ± 8.80 | 17.20 ± 7.55 | 18.40 ± 8.32 | 5.40 |
| | | SGA | **23.73 ± 4.53** | 29.33 ± 8.13 | 35.20 ± 6.36 | 39.20 ± 7.66 | 41.07 ± 8.28 | **19.33 ± 6.91** | **25.60 ± 9.26** | **28.40 ± 10.03** | **32.67 ± 10.05** | **34.13 ± 9.33** | 1.40 |
| | | GOttack | 14.93 ± 5.12 | 17.60 ± 7.79 | 24.40 ± 6.81 | 24.93 ± 7.63 | 30.67 ± 8.64 | 14.67 ± 9.15 | 13.60 ± 8.72 | 18.40 ± 8.92 | 16.13 ± 8.33 | 20.80 ± 7.40 | 4.60 |
| | GCN-GARNET | L¹D-RND | 20.00 ± 12.47 | 30.40 ± 13.57 | 35.47 ± 16.71 | 39.87 ± 14.21 | 36.67 ± 13.73 | 19.87 ± 9.61 | 23.20 ± 10.33 | 27.07 ± 10.87 | 25.07 ± 11.76 | 35.60 ± 10.15 | 1.30 |
| | | FGA | 15.87 ± 12.73 | 17.87 ± 11.27 | 22.13 ± 11.20 | 22.27 ± 10.87 | 24.53 ± 10.49 | 17.07 ± 10.58 | 16.80 ± 11.88 | 22.00 ± 10.20 | **26.40 ± 12.05** | 21.87 ± 9.72 | 3.20 |
| | | NETTACK | 2.13 ± 1.41 | 1.73 ± 1.49 | 2.67 ± 2.09 | 2.27 ± 2.60 | 1.73 ± 1.49 | 3.07 ± 2.71 | 3.60 ± 2.75 | 3.33 ± 2.47 | 4.13 ± 2.33 | 4.40 ± 2.64 | 7.00 |
| | | PGD | 10.40 ± 5.25 | 16.00 ± 7.86 | 19.47 ± 7.80 | 20.00 ± 8.38 | 21.47 ± 9.98 | 12.27 ± 7.28 | 15.33 ± 8.84 | 22.53 ± 8.96 | 22.93 ± 6.63 | 22.00 ± 8.45 | 4.50 |
| | | PR-BCD (NA) | **23.33 ± 10.81** | 27.07 ± 9.62 | 27.47 ± 13.26 | 30.80 ± 13.39 | 29.47 ± 11.72 | 15.73 ± 9.32 | **24.40 ± 8.49** | 25.20 ± 5.49 | 24.00 ± 8.65 | | 2.20 |
| | | SGA | 13.60 ± 7.75 | 17.20 ± 7.44 | 19.73 ± 10.00 | 21.47 ± 9.02 | 23.20 ± 7.92 | 19.87 ± 9.75 | 21.20 ± 9.53 | 19.73 ± 8.61 | 21.33 ± 10.68 | 22.80 ± 10.60 | 3.80 |
| | | GOttack | 5.33 ± 4.64 | 3.73 ± 2.71 | 4.27 ± 3.01 | 5.33 ± 2.79 | 5.60 ± 3.04 | 8.27 ± 4.27 | 6.67 ± 3.35 | 10.27 ± 4.83 | 12.00 ± 8.18 | 9.33 ± 4.39 | 6.00 |
| | GNNGuard | L¹D-RND | **35.73 ± 28.31** | **35.73 ± 28.31** | **35.73 ± 28.31** | **35.73 ± 28.31** | **35.73 ± 28.31** | **0.00 ± 0.00** | **0.00 ± 0.00** | **0.00 ± 0.00** | **0.00 ± 0.00** | **0.00 ± 0.00** | 1.00 |
| | | FGA | 35.73 ± 28.31 | 35.73 ± 28.31 | 35.73 ± 28.31 | 35.47 ± 28.04 | 34.80 ± 27.21 | 0.00 ± 0.00 | 0.00 ± 0.00 | 0.00 ± 0.00 | 0.00 ± 0.00 | 0.00 ± 0.00 | 2.60 |
| | | NETTACK | 18.80 ± 15.47 | 18.80 ± 15.47 | 18.80 ± 15.47 | 18.80 ± 15.47 | 18.80 ± 15.47 | 0.00 ± 0.00 | 0.00 ± 0.00 | 0.00 ± 0.00 | 0.00 ± 0.00 | 0.00 ± 0.00 | 5.00 |
| | | PGD | 35.73 ± 28.31 | 35.73 ± 28.31 | 35.73 ± 28.31 | 35.73 ± 28.31 | 35.73 ± 28.31 | 0.00 ± 0.00 | 0.00 ± 0.00 | 0.00 ± 0.00 | 0.00 ± 0.00 | 0.00 ± 0.00 | 3.30 |
| | | PR-BCD (NA) | 35.73 ± 28.31 | 35.73 ± 28.31 | 35.73 ± 28.31 | 35.73 ± 28.31 | 35.73 ± 28.31 | 0.00 ± 0.00 | 0.00 ± 0.00 | 0.00 ± 0.00 | 0.00 ± 0.00 | 0.00 ± 0.00 | 4.30 |
| | | SGA | 35.73 ± 28.31 | 35.73 ± 28.31 | 35.73 ± 28.31 | 35.73 ± 28.31 | 35.73 ± 28.31 | 0.00 ± 0.00 | 0.00 ± 0.00 | 0.00 ± 0.00 | 0.00 ± 0.00 | 0.00 ± 0.00 | 5.30 |
| | | GOttack | 22.13 ± 17.54 | 22.13 ± 17.54 | 22.13 ± 17.54 | 22.13 ± 17.54 | 22.13 ± 17.54 | 0.00 ± 0.00 | 0.00 ± 0.00 | 0.00 ± 0.00 | 0.00 ± 0.00 | 0.00 ± 0.00 | 6.50 |
| Poison | ElasticGNN | L¹D-RND | 16.67 ± 7.92 | 21.07 ± 5.23 | 24.67 ± 7.12 | 26.67 ± 6.62 | 27.33 ± 8.57 | 19.07 ± 9.07 | 25.60 ± 7.75 | 31.73 ± 9.85 | 34.80 ± 10.63 | 42.67 ± 9.52 | 5.50 |
| | | FGA | **38.80 ± 13.15** | **47.73 ± 11.41** | 47.73 ± 11.18 | **53.33 ± 11.70** | **54.53 ± 13.80** | 29.73 ± 11.66 | **40.27 ± 8.97** | **44.13 ± 11.43** | **49.60 ± 12.36** | **49.60 ± 13.46** | 1.40 |
| | | NETTACK | 12.80 ± 4.95 | 16.80 ± 6.45 | 19.33 ± 4.76 | 22.13 ± 4.44 | 24.00 ± 4.90 | 7.60 ± 4.48 | 8.40 ± 3.04 | 10.13 ± 5.10 | 12.27 ± 5.18 | 11.60 ± 5.46 | 7.00 |
| | | PGD | 28.40 ± 10.93 | 35.33 ± 14.40 | 39.87 ± 8.96 | 42.67 ± 13.17 | 46.80 ± 13.64 | 28.40 ± 12.31 | 35.87 ± 10.73 | 40.13 ± 14.43 | 44.00 ± 10.11 | 45.73 ± 12.49 | 3.00 |
| | | PR-BCD (NA) | 24.00 ± 8.42 | 24.27 ± 7.92 | 27.47 ± 9.46 | 29.87 ± 8.77 | 32.40 ± 9.69 | 24.40 ± 7.57 | 30.67 ± 9.55 | 32.00 ± 11.34 | 37.33 ± 12.11 | 37.60 ± 9.45 | 4.40 |
| | | SGA | 35.20 ± 9.97 | 45.60 ± 12.74 | **49.60 ± 11.42** | 54.80 ± 10.74 | 56.67 ± 11.73 | **31.33 ± 6.40** | 38.53 ± 10.07 | 42.40 ± 8.53 | 48.67 ± 10.73 | 48.67 ± 10.38 | 1.60 |
| | | GOttack | 20.13 ± 7.03 | 23.20 ± 7.04 | 28.93 ± 6.41 | 30.00 ± 9.13 | 33.60 ± 8.39 | 19.73 ± 9.22 | 17.47 ± 7.07 | 23.73 ± 8.55 | 21.07 ± 8.65 | 27.73 ± 6.54 | 5.10 |
| | GCN-GARNET | L¹D-RND | 31.07 ± 8.78 | 37.60 ± 11.39 | 39.20 ± 13.33 | 44.13 ± 12.64 | 42.00 ± 11.54 | 35.00 ± 10.17 | 30.80 ± 9.99 | 38.67 ± 10.08 | 39.73 ± 15.65 | 44.40 ± 12.47 | 1.30 |
| | | FGA | 28.13 ± 8.16 | 28.67 ± 8.57 | 31.47 ± 8.37 | 30.00 ± 8.25 | 33.87 ± 11.53 | 30.53 ± 7.46 | 30.13 ± 11.55 | 34.13 ± 10.10 | 36.53 ± 9.64 | 31.07 ± 9.74 | 3.60 |
| | | NETTACK | 6.13 ± 3.81 | 3.73 ± 2.49 | 5.07 ± 2.60 | 6.00 ± 3.46 | 4.53 ± 2.45 | 7.20 ± 5.94 | 9.20 ± 5.12 | 7.20 ± 4.71 | 11.47 ± 7.39 | 9.33 ± 6.13 | 7.00 |
| | | PGD | 21.87 ± 6.61 | 26.93 ± 5.70 | 30.53 ± 7.95 | 30.00 ± 9.23 | 30.13 ± 7.80 | 28.67 ± 9.22 | 30.40 ± 7.18 | 36.53 ± 9.52 | 36.13 ± 7.61 | 36.00 ± 9.32 | 4.10 |
| | | PR-BCD (NA) | **32.27 ± 9.85** | 34.93 ± 10.00 | 37.60 ± 13.44 | 36.67 ± 12.09 | 36.27 ± 9.85 | 29.87 ± 10.18 | **37.07 ± 8.61** | 37.20 ± 7.55 | 39.20 ± 10.14 | 35.20 ± 8.61 | 2.10 |
| | | SGA | 24.40 ± 5.82 | 26.13 ± 5.37 | 29.20 ± 8.03 | 30.67 ± 8.54 | 32.80 ± 4.95 | 33.47 ± 10.68 | 33.87 ± 6.48 | 32.40 ± 9.63 | 34.67 ± 13.87 | 34.93 ± 8.28 | 3.90 |
| | | GOttack | 9.60 ± 4.01 | 7.73 ± 4.89 | 8.27 ± 2.81 | 8.93 ± 3.53 | 10.27 ± 3.10 | 18.00 ± 6.59 | 16.40 ± 6.77 | 22.00 ± 8.55 | 22.00 ± 11.46 | 20.93 ± 7.96 | 6.00 |
| | GNNGuard | L¹D-RND | **35.73 ± 28.31** | **35.73 ± 28.31** | **35.73 ± 28.31** | **35.73 ± 28.31** | **35.73 ± 28.31** | **0.00 ± 0.00** | **0.00 ± 0.00** | **0.00 ± 0.00** | **0.00 ± 0.00** | **0.00 ± 0.00** | 1.00 |
| | | FGA | 35.73 ± 28.31 | 35.73 ± 28.31 | 35.73 ± 28.31 | 35.47 ± 28.04 | 34.80 ± 27.21 | 0.00 ± 0.00 | 0.00 ± 0.00 | 0.00 ± 0.00 | 0.00 ± 0.00 | 0.00 ± 0.00 | 2.60 |
| | | NETTACK | 18.80 ± 15.47 | 18.80 ± 15.47 | 18.80 ± 15.47 | 18.80 ± 15.47 | 18.80 ± 15.47 | 0.00 ± 0.00 | 0.00 ± 0.00 | 0.00 ± 0.00 | 0.00 ± 0.00 | 0.00 ± 0.00 | 5.00 |
| | | PGD | 35.73 ± 28.31 | 35.73 ± 28.31 | 35.73 ± 28.31 | 35.73 ± 28.31 | 35.73 ± 28.31 | 0.00 ± 0.00 | 0.00 ± 0.00 | 0.00 ± 0.00 | 0.00 ± 0.00 | 0.00 ± 0.00 | 3.30 |
| | | PR-BCD (NA) | 35.73 ± 28.31 | 35.73 ± 28.31 | 35.73 ± 28.31 | 35.73 ± 28.31 | 35.73 ± 28.31 | 0.00 ± 0.00 | 0.00 ± 0.00 | 0.00 ± 0.00 | 0.00 ± 0.00 | 0.00 ± 0.00 | 4.30 |
| | | SGA | 35.73 ± 28.31 | 35.73 ± 28.31 | 35.73 ± 28.31 | 35.73 ± 28.31 | 35.73 ± 28.31 | 0.00 ± 0.00 | 0.00 ± 0.00 | 0.00 ± 0.00 | 0.00 ± 0.00 | 0.00 ± 0.00 | 5.30 |
| | | GOttack | 22.13 ± 17.54 | 22.13 ± 17.54 | 22.13 ± 17.54 | 22.13 ± 17.54 | 22.13 ± 17.54 | 0.00 ± 0.00 | 0.00 ± 0.00 | 0.00 ± 0.00 | 0.00 ± 0.00 | 0.00 ± 0.00 | 6.50 |

Table 25: **Defended Heterophily Results - 2/3.** Evaluating six adversarial attacks on four vanilla attack victim models (GRAND, RobustGCN, GCORN RUNG) - Miss-classification rate (%). Higher is better. Best performance in **bold**, second best underlined.

| | Dataset | | SQUIRREL | | | | | CHAMELEON | | | | | Avg. Rank |
|---|---|---|---|---|---|---|---|---|---|---|---|---|---|
| | Δ → | 1 | 2 | 3 | 4 | 5 | 1 | 2 | 3 | 4 | 5 | |
| **Evasion — GRAND** | L¹D-RND | **10.93 ± 10.74** | 8.13 ± 9.78 | 8.80 ± 10.58 | 9.33 ± 12.34 | 8.93 ± 11.26 | 2.67 ± 5.49 | 6.00 ± 10.58 | 6.93 ± 12.28 | 7.73 ± 14.36 | 7.60 ± 14.21 | 1.30 |
| | FGA | 2.00 ± 3.46 | 2.00 ± 2.83 | 2.53 ± 3.74 | 5.20 ± 7.51 | 3.60 ± 4.55 | 0.13 ± 0.52 | 1.20 ± 2.24 | 1.73 ± 4.13 | 2.53 ± 5.58 | 2.93 ± 6.71 | 6.10 |
| | NETTACK | 2.00 ± 2.73 | 2.53 ± 3.16 | 2.53 ± 3.16 | 3.07 ± 3.69 | 3.07 ± 3.69 | 2.13 ± 4.17 | 2.67 ± 5.59 | 2.27 ± 4.53 | 2.27 ± 4.53 | 2.27 ± 4.53 | 6.00 |
| | PGD | 2.40 ± 3.79 | 2.53 ± 3.16 | 3.73 ± 4.06 | 3.73 ± 4.27 | 3.60 ± 3.79 | 3.20 ± 7.55 | 2.27 ± 4.13 | 5.47 ± 9.98 | 7.20 ± 13.91 | 6.27 ± 11.21 | 4.20 |
| | PR-BCD (NA) | 6.27 ± 7.36 | 6.80 ± 8.03 | 6.40 ± 7.18 | 6.27 ± 7.13 | 6.53 ± 7.54 | 3.47 ± 6.16 | 6.53 ± 13.10 | 6.80 ± 13.56 | 6.80 ± 13.02 | 7.07 ± 13.56 | 1.90 |
| | SGA | 3.47 ± 3.34 | 4.53 ± 4.17 | 4.67 ± 4.32 | 4.80 ± 4.77 | 5.33 ± 5.16 | 1.73 ± 3.53 | 2.40 ± 4.61 | 4.80 ± 8.17 | 2.80 ± 5.23 | 4.27 ± 8.94 | 4.10 |
| | GOttack | 3.60 ± 4.79 | 2.67 ± 2.99 | 4.13 ± 4.81 | 2.93 ± 3.28 | 4.13 ± 4.44 | 2.67 ± 5.27 | 2.67 ± 6.31 | 4.13 ± 9.36 | 3.47 ± 9.27 | 4.27 ± 9.82 | 4.40 |
| **Evasion — RobustGCN** | L¹D-RND | 9.73 ± 14.38 | 25.33 ± 9.09 | 40.40 ± 5.03 | 41.73 ± 6.58 | 46.40 ± 7.02 | 11.07 ± 13.43 | 31.47 ± 7.39 | 41.60 ± 9.36 | 47.60 ± 9.36 | 46.67 ± 8.16 | 4.40 |
| | FGA | 35.47 ± 12.52 | 46.27 ± 12.33 | 50.27 ± 9.85 | 49.33 ± 12.06 | 52.00 ± 11.44 | 38.13 ± 9.52 | 45.87 ± 10.18 | 45.33 ± 11.65 | 51.73 ± 11.66 | 49.73 ± 11.63 | 1.10 |
| | NETTACK | 9.60 ± 4.48 | 13.33 ± 4.70 | 15.60 ± 6.47 | 17.47 ± 6.78 | 20.27 ± 6.80 | 6.27 ± 3.53 | 7.20 ± 3.10 | 7.07 ± 4.40 | 8.40 ± 5.57 | 10.13 ± 6.30 | 7.00 |
| | PGD | 35.47 ± 11.67 | 41.20 ± 13.02 | 41.07 ± 13.37 | 44.80 ± 16.90 | 42.93 ± 12.46 | 32.13 ± 7.07 | 40.67 ± 10.52 | 41.20 ± 10.63 | 48.27 ± 11.08 | 48.40 ± 12.40 | 3.00 |
| | PR-BCD (NA) | 35.47 ± 7.50 | 41.47 ± 7.95 | 45.20 ± 6.36 | 44.80 ± 5.54 | 49.07 ± 7.28 | 28.93 ± 7.09 | 33.33 ± 9.70 | 40.13 ± 10.60 | 39.33 ± 10.27 | 38.67 ± 10.89 | 3.70 |
| | SGA | **40.00 ± 5.86** | 44.80 ± 8.65 | 44.67 ± 8.09 | 48.13 ± 7.73 | 48.67 ± 7.43 | 32.13 ± 9.18 | 36.53 ± 9.98 | 37.60 ± 10.40 | 43.60 ± 12.19 | 43.73 ± 11.23 | 3.00 |
| | GOttack | 18.40 ± 6.98 | 17.20 ± 8.10 | 21.47 ± 7.54 | 22.13 ± 9.84 | 27.60 ± 8.08 | 26.40 ± 7.90 | 26.00 ± 8.94 | 30.93 ± 10.02 | 24.53 ± 9.36 | 30.27 ± 6.71 | 5.80 |
| **Evasion — GCORN** | L¹D-RND | 18.67 ± 15.72 | 26.13 ± 17.00 | 29.20 ± 18.15 | 32.27 ± 16.97 | 34.53 ± 17.39 | 16.53 ± 15.94 | 32.53 ± 12.34 | 38.13 ± 14.65 | 41.47 ± 18.60 | 45.73 ± 20.60 | 3.10 |
| | FGA | 31.07 ± 21.02 | 33.47 ± 16.54 | 36.27 ± 18.79 | 36.67 ± 19.00 | 37.60 ± 19.14 | 34.13 ± 19.23 | 36.53 ± 15.99 | 48.67 ± 17.03 | 48.67 ± 18.40 | 48.67 ± 19.03 | 1.10 |
| | NETTACK | 6.67 ± 5.38 | 8.53 ± 5.73 | 9.60 ± 5.14 | 10.27 ± 5.85 | 11.07 ± 6.50 | 3.47 ± 3.58 | 5.33 ± 3.75 | 6.67 ± 4.39 | 8.13 ± 3.74 | 9.20 ± 4.06 | 6.90 |
| | PGD | 23.47 ± 10.57 | 31.20 ± 10.50 | 31.60 ± 11.19 | 34.00 ± 13.33 | 35.60 ± 13.12 | 21.07 ± 13.69 | 32.13 ± 16.89 | 31.73 ± 17.19 | 36.00 ± 17.44 | 37.20 ± 18.25 | 2.90 |
| | PR-BCD (NA) | 13.20 ± 13.18 | 17.07 ± 11.18 | 17.33 ± 9.93 | 19.20 ± 10.55 | 22.67 ± 10.02 | 10.13 ± 7.87 | 15.20 ± 9.47 | 17.07 ± 11.05 | 20.27 ± 12.46 | 22.93 ± 14.91 | 5.00 |
| | SGA | 24.27 ± 12.14 | 31.87 ± 13.10 | 35.73 ± 13.31 | 34.67 ± 10.49 | **37.73 ± 11.36** | 15.60 ± 7.86 | 21.33 ± 8.57 | 22.93 ± 6.84 | 27.87 ± 10.38 | 29.33 ± 9.61 | 2.90 |
| | GOttack | 9.47 ± 6.74 | 10.40 ± 7.75 | 12.93 ± 5.99 | 12.93 ± 6.13 | 14.27 ± 6.80 | 6.00 ± 4.14 | 5.07 ± 3.92 | 9.73 ± 5.70 | 10.13 ± 5.88 | 10.27 ± 5.18 | 6.10 |
| **Evasion — RUNG** | L¹D-RND | 2.13 ± 1.77 | 2.13 ± 2.77 | 2.13 ± 2.20 | 2.67 ± 2.58 | 2.80 ± 2.48 | 2.27 ± 3.28 | 2.27 ± 5.18 | 3.47 ± 6.61 | 4.53 ± 6.65 | 5.07 ± 7.17 | 2.70 |
| | FGA | 1.87 ± 2.88 | 1.87 ± 2.20 | 3.20 ± 3.36 | 2.93 ± 2.25 | 3.60 ± 2.53 | 0.93 ± 1.83 | 2.27 ± 3.69 | 2.53 ± 3.58 | 3.33 ± 5.22 | 4.13 ± 5.37 | 3.40 |
| | NETTACK | 0.27 ± 1.03 | 0.80 ± 1.47 | 1.33 ± 1.63 | 1.47 ± 1.77 | 1.60 ± 1.72 | 0.27 ± 0.70 | 0.67 ± 1.23 | 0.53 ± 1.19 | 0.93 ± 1.83 | 1.33 ± 2.23 | 6.90 |
| | PGD | 2.00 ± 1.85 | 2.67 ± 2.89 | 4.40 ± 2.85 | 4.67 ± 2.99 | 4.93 ± 3.10 | 1.87 ± 4.69 | 2.00 ± 3.02 | 1.60 ± 2.03 | 3.20 ± 4.71 | 3.47 ± 5.93 | 3.10 |
| | PR-BCD (NA) | 1.73 ± 2.49 | 2.67 ± 2.47 | 3.20 ± 2.60 | 2.27 ± 2.12 | 3.33 ± 3.09 | 0.80 ± 2.24 | 1.87 ± 3.58 | 1.87 ± 3.96 | 2.40 ± 3.87 | 3.20 ± 5.65 | 4.60 |
| | SGA | **2.67 ± 3.68** | **3.47 ± 4.17** | **5.07 ± 4.77** | **5.07 ± 4.59** | **5.60 ± 4.15** | 2.13 ± 4.31 | 2.93 ± 4.95 | 3.20 ± 3.61 | 5.60 ± 7.38 | 4.40 ± 6.47 | 1.30 |
| | GOttack | 0.93 ± 2.25 | 0.93 ± 1.83 | 1.73 ± 2.71 | 1.47 ± 2.07 | 2.00 ± 2.83 | 0.67 ± 1.23 | 1.33 ± 2.09 | 1.60 ± 2.16 | 1.87 ± 3.25 | 3.47 ± 5.04 | 6.00 |
| **Poison — GRAND** | L¹D-RND | 16.00 ± 13.56 | 14.00 ± 13.16 | 16.80 ± 15.89 | 19.07 ± 17.81 | 19.07 ± 17.63 | 4.00 ± 5.95 | 9.33 ± 13.75 | 12.67 ± 19.50 | 12.40 ± 20.38 | 13.07 ± 21.93 | 2.10 |
| | FGA | 12.13 ± 21.71 | 16.53 ± 26.57 | 19.33 ± 26.08 | 21.33 ± 26.15 | 17.07 ± 17.56 | 2.00 ± 3.12 | 5.87 ± 9.66 | 5.07 ± 8.00 | 11.20 ± 24.69 | 8.93 ± 15.87 | 3.30 |
| | NETTACK | 4.40 ± 7.53 | 5.47 ± 6.91 | 6.27 ± 8.34 | 8.13 ± 10.32 | 7.47 ± 10.27 | 4.00 ± 7.09 | 5.20 ± 7.92 | 4.93 ± 7.89 | 5.20 ± 7.55 | 4.00 ± 5.55 | 6.60 |
| | PGD | 11.07 ± 11.66 | 12.93 ± 11.08 | 18.53 ± 16.15 | 21.20 ± 18.03 | 21.07 ± 17.55 | 5.33 ± 8.42 | 8.13 ± 11.80 | 12.40 ± 18.64 | 11.33 ± 17.13 | 12.13 ± 16.59 | 2.30 |
| | PR-BCD (NA) | 9.07 ± 9.47 | 12.13 ± 12.39 | 11.07 ± 10.69 | 12.13 ± 12.39 | 11.33 ± 12.87 | 7.07 ± 9.07 | 9.47 ± 14.17 | 11.07 ± 16.18 | 11.07 ± 15.47 | 11.60 ± 16.87 | 3.40 |
| | SGA | 7.07 ± 6.92 | 11.87 ± 11.75 | 10.40 ± 8.22 | 14.93 ± 14.83 | 14.40 ± 13.14 | 6.53 ± 11.04 | 6.00 ± 9.13 | 7.87 ± 10.89 | 6.80 ± 9.73 | 8.27 ± 11.61 | 4.30 |
| | GOttack | 6.93 ± 9.44 | 6.13 ± 10.99 | 6.80 ± 6.96 | 5.20 ± 5.28 | 6.13 ± 6.21 | 4.80 ± 7.16 | 4.00 ± 6.85 | 5.47 ± 9.61 | 6.40 ± 12.29 | 6.67 ± 10.81 | 6.00 |
| **Poison — RobustGCN** | L¹D-RND | 26.00 ± 14.89 | 33.33 ± 10.13 | 45.60 ± 6.98 | 45.33 ± 8.13 | 48.93 ± 8.10 | 23.07 ± 11.83 | 36.13 ± 9.69 | 44.27 ± 10.17 | 49.60 ± 9.98 | 48.80 ± 8.48 | 4.80 |
| | FGA | 43.47 ± 13.64 | 51.73 ± 12.33 | 56.67 ± 11.05 | 54.93 ± 12.09 | 57.33 ± 12.69 | 43.07 ± 11.08 | 53.07 ± 11.46 | 52.67 ± 13.58 | 56.80 ± 12.78 | 53.33 ± 11.99 | 1.30 |
| | NETTACK | 14.67 ± 7.58 | 18.67 ± 8.51 | 21.60 ± 9.30 | 21.60 ± 7.72 | 24.53 ± 8.33 | 9.73 ± 5.28 | 9.73 ± 5.30 | 10.27 ± 5.50 | 12.93 ± 6.92 | 12.27 ± 8.00 | 7.00 |
| | PGD | 44.00 ± 14.44 | 46.93 ± 14.66 | 46.53 ± 16.20 | 50.13 ± 18.51 | 50.27 ± 14.28 | 37.20 ± 8.44 | 46.40 ± 8.76 | 48.13 ± 11.10 | 54.13 ± 9.66 | 54.27 ± 12.33 | 2.80 |
| | PR-BCD (NA) | 41.07 ± 7.89 | 48.00 ± 11.56 | 52.27 ± 7.17 | 51.33 ± 8.13 | 49.73 ± 9.28 | 37.73 ± 8.10 | 41.33 ± 11.82 | 44.53 ± 10.21 | 48.40 ± 9.51 | 44.40 ± 10.53 | 3.50 |
| | SGA | **46.00 ± 8.94** | 48.80 ± 8.81 | 50.40 ± 10.48 | 50.67 ± 8.80 | 53.07 ± 9.28 | 40.67 ± 10.05 | 46.27 ± 10.47 | 44.13 ± 9.81 | 51.07 ± 12.65 | 50.53 ± 12.01 | 2.70 |
| | GOttack | 20.80 ± 6.92 | 24.13 ± 11.48 | 25.07 ± 8.94 | 27.33 ± 11.58 | 31.87 ± 10.78 | 29.60 ± 5.96 | 28.80 ± 9.50 | 36.80 ± 7.24 | 31.87 ± 10.32 | 35.33 ± 6.26 | 5.90 |
| **Poison — GCORN** | L¹D-RND | 20.00 ± 16.30 | 26.67 ± 17.35 | 30.93 ± 19.71 | 33.60 ± 17.88 | 36.13 ± 17.38 | 17.33 ± 16.74 | 33.47 ± 13.06 | 38.80 ± 15.08 | 42.67 ± 18.07 | 46.27 ± 20.41 | 3.30 |
| | FGA | **34.27 ± 22.68** | 34.40 ± 16.67 | 37.33 ± 16.26 | 40.80 ± 18.80 | 37.73 ± 17.43 | 34.53 ± 17.93 | 36.80 ± 15.25 | 48.13 ± 17.06 | 49.87 ± 18.29 | 48.40 ± 18.66 | 1.20 |
| | NETTACK | 8.13 ± 6.61 | 9.20 ± 7.48 | 10.40 ± 5.96 | 12.13 ± 6.21 | 13.07 ± 6.96 | 4.27 ± 3.45 | 5.87 ± 4.44 | 7.60 ± 4.22 | 9.47 ± 3.96 | 11.47 ± 5.68 | 7.00 |
| | PGD | 25.60 ± 12.03 | 33.47 ± 11.07 | 34.53 ± 11.92 | 37.33 ± 13.04 | 37.60 ± 11.69 | 22.67 ± 13.81 | 34.00 ± 16.82 | 33.33 ± 16.08 | 37.87 ± 16.54 | 37.73 ± 17.32 | 2.80 |
| | PR-BCD (NA) | 14.53 ± 13.68 | 18.13 ± 10.81 | 19.07 ± 9.94 | 21.33 ± 7.84 | 24.40 ± 9.80 | 11.33 ± 7.24 | 16.13 ± 8.67 | 18.13 ± 10.70 | 21.60 ± 11.96 | 25.20 ± 14.48 | 5.00 |
| | SGA | 26.13 ± 13.89 | 34.40 ± 13.38 | 37.60 ± 13.80 | 37.60 ± 10.96 | **39.33 ± 9.67** | 18.80 ± 8.20 | 24.80 ± 8.87 | 25.73 ± 7.25 | 29.07 ± 10.55 | 32.53 ± 10.32 | 2.70 |
| | GOttack | 10.67 ± 8.61 | 11.47 ± 9.52 | 14.67 ± 7.24 | 14.53 ± 7.58 | 15.73 ± 8.48 | 6.53 ± 3.66 | 6.40 ± 4.42 | 11.47 ± 5.24 | 12.00 ± 5.55 | 12.00 ± 5.50 | 6.00 |
| **Poison — RUNG** | L¹D-RND | 11.33 ± 8.64 | 16.13 ± 8.26 | 17.07 ± 8.94 | 18.67 ± 8.02 | 19.33 ± 8.09 | 12.00 ± 8.88 | 21.33 ± 12.30 | 24.00 ± 10.61 | 28.00 ± 13.96 | 32.40 ± 14.35 | 3.30 |
| | FGA | **20.53 ± 9.69** | 24.67 ± 10.27 | 29.47 ± 7.23 | 28.53 ± 9.21 | 28.67 ± 10.63 | 16.40 ± 7.53 | 17.47 ± 12.06 | 17.20 ± 7.70 | 18.80 ± 10.77 | 19.20 ± 11.18 | 2.00 |
| | NETTACK | 6.53 ± 4.44 | 6.93 ± 5.18 | 6.53 ± 4.81 | 6.93 ± 4.83 | 6.80 ± 4.26 | 7.33 ± 2.89 | 6.67 ± 2.89 | 7.07 ± 3.61 | 7.47 ± 4.06 | 7.47 ± 3.34 | 6.80 |
| | PGD | 17.33 ± 7.81 | 23.20 ± 9.91 | 26.40 ± 9.05 | 27.07 ± 6.23 | 27.87 ± 8.63 | 13.73 ± 6.41 | 18.40 ± 7.64 | 19.20 ± 8.94 | 20.00 ± 9.80 | 19.60 ± 11.76 | 2.30 |
| | PR-BCD (NA) | 15.07 ± 6.18 | 19.47 ± 7.31 | 19.20 ± 6.84 | 20.67 ± 7.73 | 20.53 ± 6.99 | 11.73 ± 5.90 | 14.40 ± 6.15 | 13.87 ± 7.87 | 14.53 ± 5.42 | 15.73 ± 6.45 | 4.50 |
| | SGA | 18.40 ± 9.33 | 22.27 ± 8.78 | 26.27 ± 8.61 | 26.93 ± 10.63 | 27.73 ± 10.11 | 14.27 ± 6.76 | 15.07 ± 5.23 | 20.40 ± 7.41 | 19.60 ± 7.97 | 18.00 ± 6.46 | 2.90 |
| | GOttack | 6.27 ± 5.18 | 6.93 ± 6.23 | 8.00 ± 5.66 | 7.60 ± 5.62 | 9.20 ± 6.58 | 10.80 ± 4.89 | 8.27 ± 5.80 | 9.20 ± 3.69 | 10.53 ± 3.42 | 12.53 ± 6.35 | 6.20 |

Table 26: **Defended Heterophily Results - 3/3.** Evaluating six adversarial attacks on four vanilla attack victim models (NoisyGNN) - Miss-classification rate (%). Higher is better. Best performance in **bold**, second best underlined.

| Dataset | | $\Delta \rightarrow$ | squirrel | | | | | chameleon | | | | | Avg. Rank |
|---|---|---|---|---|---|---|---|---|---|---|---|---|---|
| | | | 1 | 2 | 3 | 4 | 5 | 1 | 2 | 3 | 4 | 5 | |
| Evasion | NoisyGNN | L$^1$D-RND | 68.93 ± 7.05 | 67.47 ± 7.87 | 67.47 ± 7.87 | 67.47 ± 7.87 | 67.60 ± 7.83 | **63.73 ± 5.99** | 64.67 ± 7.95 | 64.93 ± 6.67 | **63.73 ± 6.13** | 65.87 ± 5.32 | 2.70 |
| | | FGA | 66.80 ± 11.13 | 66.93 ± 10.82 | 66.53 ± 12.70 | 67.07 ± 13.33 | 64.93 ± 17.09 | 62.27 ± 11.49 | **64.93 ± 14.24** | **70.00 ± 13.31** | 62.67 ± 15.50 | 67.20 ± 14.54 | 3.40 |
| | | NETTACK | 26.13 ± 4.93 | 26.00 ± 5.01 | 26.00 ± 5.01 | 26.00 ± 5.01 | 26.00 ± 5.01 | 4.80 ± 8.51 | 4.80 ± 8.55 | 5.47 ± 8.26 | 6.13 ± 8.02 | 5.73 ± 8.17 | 7.00 |
| | | PGD | 66.93 ± 8.58 | 68.40 ± 7.90 | 67.33 ± 9.00 | 67.73 ± 9.47 | 69.47 ± 8.83 | 57.60 ± 8.08 | 62.53 ± 8.70 | 63.47 ± 6.95 | 63.60 ± 7.68 | **68.53 ± 6.25** | 3.00 |
| | | PR-BCD | **69.73 ± 6.54** | **69.87 ± 6.44** | 69.47 ± 7.07 | 69.20 ± 7.78 | 70.40 ± 6.38 | 53.33 ± 7.70 | 54.80 ± 7.59 | 53.87 ± 7.73 | 53.20 ± 8.20 | 53.47 ± 8.90 | 3.30 |
| | | SGA | 69.07 ± 6.88 | 69.73 ± 7.48 | **69.73 ± 7.74** | **70.53 ± 7.35** | **71.47 ± 6.95** | 60.13 ± 7.69 | 60.13 ± 6.52 | 60.80 ± 7.55 | 60.00 ± 7.13 | 61.60 ± 8.72 | 2.60 |
| | | GOttack | 34.13 ± 4.87 | 34.27 ± 4.89 | 34.40 ± 5.03 | 34.00 ± 4.72 | 34.40 ± 4.67 | 19.20 ± 8.68 | 21.47 ± 9.87 | 20.27 ± 8.55 | 21.20 ± 9.25 | 20.80 ± 9.47 | 6.00 |
| Poison | NoisyGNN | L$^1$D-RND | 71.87 ± 6.25 | 71.87 ± 6.25 | 72.27 ± 6.50 | 71.60 ± 6.38 | 72.27 ± 6.04 | 64.93 ± 5.80 | 66.40 ± 7.14 | 65.87 ± 8.50 | 65.73 ± 4.71 | 67.73 ± 6.09 | 4.10 |
| | | FGA | 72.53 ± 9.81 | 74.13 ± 7.31 | 70.40 ± 9.05 | 71.87 ± 8.86 | 72.67 ± 10.79 | **68.00 ± 9.41** | **74.53 ± 13.78** | **76.67 ± 13.06** | **74.00 ± 19.61** | **80.40 ± 16.65** | 2.50 |
| | | NETTACK | 29.87 ± 4.37 | 29.60 ± 5.14 | 30.53 ± 5.10 | 31.33 ± 4.82 | 31.47 ± 3.89 | 8.40 ± 9.83 | 9.47 ± 9.15 | 10.13 ± 10.32 | 9.47 ± 9.36 | 10.13 ± 8.60 | 7.00 |
| | | PGD | 72.67 ± 7.39 | 74.40 ± 7.14 | 74.27 ± 8.14 | 74.40 ± 8.18 | **76.40 ± 8.82** | 61.73 ± 7.70 | 68.27 ± 9.38 | 71.87 ± 5.48 | 73.47 ± 7.58 | 77.73 ± 6.76 | 2.30 |
| | | PR-BCD | 72.93 ± 5.28 | 73.47 ± 5.04 | 74.27 ± 5.01 | 74.53 ± 4.87 | 73.87 ± 6.12 | 59.33 ± 7.12 | 60.53 ± 6.70 | 60.93 ± 6.50 | 62.00 ± 7.25 | 60.53 ± 6.99 | 3.90 |
| | | SGA | **74.67 ± 6.62** | **74.80 ± 5.85** | **74.93 ± 6.45** | **75.47 ± 5.83** | 75.07 ± 5.60 | 62.67 ± 7.88 | 65.47 ± 6.99 | 69.47 ± 7.07 | 68.40 ± 8.32 | 69.60 ± 9.39 | 2.20 |
| | | GOttack | 37.87 ± 4.75 | 38.13 ± 5.58 | 39.87 ± 6.07 | 38.27 ± 6.32 | 40.00 ± 5.76 | 23.60 ± 8.29 | 26.13 ± 9.15 | 24.80 ± 10.63 | 27.20 ± 8.48 | 26.27 ± 9.68 | 6.00 |

Table 27: **Defended PRBCD on Homophily Datasets.** Evaluating PRBCD and adaptive version of PRBCD (PR-BCD (NA)) adversarial attacks on two defense victim models (ElasticGNN, RobustGCN) - Miss-classification rate (%). Higher is better. Best performance in **bold**, second best underlined

| Dataset | CORA | | | | | CITESEER | | | | | PUBMED | | | | | |
|---|---|---|---|---|---|---|---|---|---|---|---|---|---|---|---|---|
| Δ→ | 1 | 2 | 3 | 4 | 5 | 1 | 2 | 3 | 4 | 5 | 1 | 2 | 3 | 4 | 5 | Avg. Rank |
| Evasion ElasticGNN PR-BCD | **24.80 ± 2.81** | 30.67 ± 4.88 | 37.87 ± 7.27 | **45.47 ± 7.76** | 50.13 ± 8.16 | **23.60 ± 7.53** | **31.87 ± 8.73** | **39.20 ± 10.71** | **45.47 ± 10.76** | 49.73 ± 13.52 | **22.93 ± 3.84** | **25.60 ± 4.15** | 27.33 ± 4.94 | 29.60 ± 5.25 | **31.73 ± 5.01** | 1.40 |
| Evasion ElasticGNN PR-BCD (NA) | 24.67 ± 3.75 | **32.00 ± 5.76** | **41.60 ± 6.85** | 44.80 ± 7.28 | **50.40 ± 9.05** | 23.33 ± 4.76 | 30.93 ± 7.44 | 38.40 ± 9.66 | 45.47 ± 11.12 | **51.33 ± 11.87** | 22.13 ± 3.16 | 25.60 ± 4.42 | **28.00 ± 4.28** | **30.53 ± 5.53** | 30.53 ± 3.89 | 1.60 |
| Evasion RobustGCN PR-BCD | 27.73 ± 5.44 | 49.33 ± 5.94 | 56.13 ± 4.31 | 60.53 ± 3.58 | 62.93 ± 4.20 | **24.27 ± 4.13** | 40.80 ± 5.00 | 53.07 ± 3.20 | **56.53 ± 2.77** | **58.67 ± 2.79** | 29.60 ± 3.79 | **40.40 ± 5.36** | **48.53 ± 3.58** | **55.20 ± 2.37** | 56.27 ± 1.83 | 1.60 |
| Evasion RobustGCN PR-BCD (NA) | **32.40 ± 4.79** | **58.00 ± 3.85** | **62.00 ± 3.93** | **64.67 ± 3.44** | **64.93 ± 2.49** | 23.87 ± 3.74 | **42.67 ± 5.98** | **53.20 ± 4.89** | 55.87 ± 3.25 | 58.27 ± 3.53 | **31.47 ± 2.07** | 38.40 ± 4.61 | 48.53 ± 3.42 | 54.00 ± 3.12 | **57.07 ± 2.25** | 1.40 |
| Poison ElasticGNN PR-BCD | **27.33 ± 3.18** | **38.00 ± 7.29** | **47.47 ± 5.32** | 53.20 ± 4.83 | **60.53 ± 5.15** | 27.20 ± 9.00 | **41.20 ± 10.08** | 48.27 ± 12.40 | **54.67 ± 11.97** | 56.93 ± 12.16 | **21.87 ± 4.87** | 24.53 ± 3.96 | 26.80 ± 4.52 | 29.47 ± 5.04 | **31.47 ± 4.63** | 1.47 |
| Poison ElasticGNN PR-BCD (NA) | 25.60 ± 4.15 | 36.53 ± 5.93 | 47.47 ± 6.25 | **55.33 ± 3.83** | 59.47 ± 5.04 | **29.47 ± 9.21** | 38.93 ± 11.29 | **50.93 ± 11.71** | 53.73 ± 10.22 | **57.73 ± 10.00** | 20.53 ± 3.07 | **25.60 ± 3.94** | **27.07 ± 4.83** | **30.80 ± 6.13** | 30.80 ± 4.06 | 1.53 |
| Poison RobustGCN PR-BCD | 30.93 ± 4.40 | 50.00 ± 4.47 | 58.00 ± 3.78 | 60.13 ± 2.97 | 62.40 ± 4.61 | **29.47 ± 4.10** | 48.13 ± 4.98 | 55.60 ± 2.53 | **58.27 ± 3.37** | 59.60 ± 3.94 | 29.47 ± 3.34 | **40.53 ± 4.56** | 50.27 ± 3.45 | **55.60 ± 2.95** | 56.27 ± 1.98 | 1.67 |
| Poison RobustGCN PR-BCD (NA) | **34.40 ± 4.08** | **57.20 ± 3.19** | **61.73 ± 3.37** | **64.13 ± 2.97** | **65.33 ± 2.99** | 28.67 ± 5.49 | **49.60 ± 6.56** | **56.00 ± 4.47** | 57.47 ± 4.37 | **60.00 ± 4.07** | **31.33 ± 2.58** | 38.93 ± 4.89 | 49.47 ± 3.50 | 54.27 ± 2.12 | **56.93 ± 2.12** | 1.33 |

Table 28: **Defended PRBCD on Heterophily Datasets.** Evaluating non adaptive PR-BCD (PR-BCD (NA)), and adaptive version of PR-BCD adversarial attacks on two defense victim models (ElasticGNN, RobustGCN) - Miss-classification rate (%). Higher is better. Best performance in **bold**, second best underlined

| Dataset | SQUIRREL | | | | | CHAMELEON | | | | | |
|---|---|---|---|---|---|---|---|---|---|---|---|
| Δ→ | 1 | 2 | 3 | 4 | 5 | 1 | 2 | 3 | 4 | 5 | Avg. Rank |
| Evasion ElasticGNN PR-BCD | **12.93 ± 4.59** | **16.13 ± 5.26** | 17.33 ± 4.51 | **19.47 ± 4.56** | 19.07 ± 6.76 | 11.73 ± 9.13 | 13.33 ± 5.11 | **16.93 ± 7.52** | **19.47 ± 10.18** | **19.60 ± 10.18** | 1.40 |
| Evasion ElasticGNN PR-BCD (NA) | 12.13 ± 3.81 | 15.73 ± 5.70 | **18.13 ± 5.83** | 19.33 ± 6.13 | **20.13 ± 6.21** | **12.27 ± 7.59** | **14.40 ± 8.85** | 16.67 ± 8.80 | 17.20 ± 7.55 | 18.40 ± 8.32 | 1.60 |
| Evasion RobustGCN PR-BCD | **37.73 ± 7.00** | **46.40 ± 6.10** | **46.27 ± 8.17** | 44.13 ± 6.25 | 47.73 ± 8.34 | **31.33 ± 8.47** | 33.07 ± 8.48 | 36.67 ± 8.77 | 38.53 ± 11.07 | 37.47 ± 7.76 | 1.60 |
| Evasion RobustGCN PR-BCD (NA) | 35.47 ± 7.50 | 41.47 ± 7.95 | 45.20 ± 6.36 | **44.80 ± 5.54** | **49.07 ± 7.28** | 28.93 ± 7.09 | **33.33 ± 9.70** | **40.13 ± 10.60** | **39.33 ± 10.27** | **38.67 ± 10.89** | 1.40 |
| Poison ElasticGNN PR-BCD | **28.00 ± 6.37** | **28.00 ± 8.04** | **29.20 ± 6.92** | **32.53 ± 8.57** | 29.07 ± 8.48 | **26.40 ± 9.86** | 28.80 ± 5.99 | **34.13 ± 11.48** | 36.27 ± 13.54 | 36.53 ± 12.86 | 1.40 |
| Poison ElasticGNN PR-BCD (NA) | 24.00 ± 8.42 | 24.27 ± 7.92 | 27.47 ± 9.46 | 29.87 ± 8.77 | **32.40 ± 9.69** | 24.40 ± 7.57 | **30.67 ± 9.55** | 32.00 ± 11.34 | **37.33 ± 12.11** | **37.60 ± 9.45** | 1.60 |
| Poison RobustGCN PR-BCD | **44.27 ± 10.36** | **51.20 ± 8.48** | 48.27 ± 8.51 | 48.27 ± 9.28 | **52.00 ± 10.39** | 36.53 ± 8.63 | 38.93 ± 7.96 | 42.80 ± 7.00 | 44.27 ± 12.16 | 42.93 ± 9.44 | 1.70 |
| Poison RobustGCN PR-BCD (NA) | 41.07 ± 7.89 | 48.00 ± 11.56 | **52.27 ± 7.17** | **51.33 ± 8.13** | 49.73 ± 9.28 | **37.73 ± 8.10** | **41.33 ± 11.82** | **44.53 ± 10.21** | **48.40 ± 9.51** | **44.40 ± 10.53** | 1.30 |

Table 29: Effect of including node degree as criteria to select the target node on adversarial evaluation results. When evaluating adversarial attacks on target nodes, including node degree as selection criteria, is lower than evaluating those on target nodes without node degree as selection criteria, the results are shown in red; otherwise, the results are shown in blue.

| | Dataset | CORA | | | | | CITESEER | | | | | PUBMED | | | | | |
| --- | --- | 1 | 2 | 3 | 4 | 5 | 1 | 2 | 3 | 4 | 5 | 1 | 2 | 3 | 4 | 5 | Avg. Rank Difference |
| | $\Delta \rightarrow$ | 1 | 2 | 3 | 4 | 5 | 1 | 2 | 3 | 4 | 5 | 1 | 2 | 3 | 4 | 5 | |
| **Evasion** GCN | $L^1$D-RND | -3.30 | -3.63 | -1.63 | 0.10 | -5.87 | -5.13 | -1.50 | 5.37 | -2.60 | 0.13 | -5.07 | -9.20 | -11.13 | -10.03 | -10.23 | 0.00 |
| | FGA | -11.88 | -7.37 | -7.27 | -6.07 | -5.83 | -16.78 | -19.25 | -10.23 | -6.47 | -9.98 | -21.57 | -19.57 | -13.83 | -13.43 | -13.83 | 1.00 |
| | NETTACK | -7.15 | 0.60 | -2.45 | -2.62 | -3.02 | -12.12 | -5.67 | -10.13 | -6.32 | -8.52 | -23.93 | -10.43 | -10.70 | -10.50 | -13.83 | -1.20 |
| | PGD | -11.67 | -5.50 | -5.07 | -4.65 | -6.80 | -18.53 | -13.23 | -11.87 | -10.98 | -10.53 | -22.70 | -22.50 | -26.90 | -24.20 | -25.90 | 1.00 |
| | PR-BCD (NA) | -10.87 | -5.97 | -6.80 | -7.30 | -7.27 | -12.47 | -2.33 | -2.10 | -3.08 | -3.88 | -22.90 | -26.30 | -20.10 | -16.20 | -13.53 | -0.20 |
| | SGA | -9.98 | -2.83 | -0.42 | -1.87 | -4.08 | -10.28 | -10.38 | -5.28 | -8.18 | -8.52 | -22.90 | -20.87 | -17.50 | -14.73 | -16.03 | -0.27 |
| | GOttack | -10.97 | -7.50 | -8.70 | -6.80 | -7.70 | -11.90 | -12.00 | -6.90 | -5.83 | -10.87 | -22.47 | -16.33 | -12.70 | -11.03 | -12.07 | -0.33 |
| GIN | $L^1$D-RND | -2.03 | -1.93 | -3.67 | -4.47 | -5.10 | 0.63 | 1.30 | -3.77 | -5.13 | -6.67 | -3.67 | -8.77 | -2.50 | -8.93 | -9.57 | -0.33 |
| | FGA | -12.03 | -10.18 | -6.85 | -3.33 | -4.53 | -4.18 | -4.98 | -1.12 | -8.22 | -2.67 | -16.47 | -11.13 | -15.00 | -6.17 | -12.53 | 0.07 |
| | NETTACK | -5.97 | -2.80 | -3.63 | -6.80 | -4.80 | -1.43 | -2.37 | -5.65 | -7.98 | -11.72 | -11.40 | -13.37 | -11.27 | -11.73 | -9.97 | -0.40 |
| | PGD | -6.68 | -5.98 | -6.90 | -9.40 | -9.15 | -2.22 | -2.87 | -6.82 | -6.88 | -5.65 | -13.37 | -15.67 | -18.70 | -15.67 | -18.30 | 0.00 |
| | PR-BCD (NA) | -8.28 | -5.72 | -6.73 | -11.45 | -10.37 | -6.05 | -1.43 | -7.07 | -7.07 | -10.65 | -10.27 | -18.67 | -20.97 | -17.83 | -17.70 | 0.00 |
| | SGA | -8.85 | -5.83 | -12.57 | -9.85 | -10.80 | -0.32 | 1.53 | -7.60 | -3.85 | -0.65 | -9.47 | -16.33 | -13.70 | -14.50 | -11.57 | 0.13 |
| | GOttack | -8.00 | -4.43 | -6.70 | -10.67 | -10.93 | -5.70 | -5.53 | -7.37 | -11.63 | -11.17 | -16.97 | -9.43 | -14.47 | -14.17 | -16.70 | 0.53 |
| GSAGE | $L^1$D-RND | -3.63 | -1.00 | -2.07 | -1.73 | -4.30 | -3.77 | -5.23 | -4.93 | -4.83 | -6.30 | -4.00 | -8.83 | -8.93 | -8.73 | -10.00 | -0.67 |
| | FGA | -8.07 | -13.68 | -9.05 | -4.03 | -6.17 | -6.88 | -6.40 | -10.73 | -10.03 | -6.38 | -13.43 | -17.63 | -17.70 | -16.03 | -14.27 | -0.60 |
| | NETTACK | -10.75 | -6.38 | -4.40 | -6.57 | -9.02 | -9.48 | -3.18 | -3.57 | -7.98 | -6.22 | -13.10 | -20.43 | -12.13 | -14.07 | -16.47 | 0.20 |
| | PGD | -11.50 | -8.78 | -12.48 | -12.80 | -14.87 | -7.22 | -8.93 | -14.08 | -8.60 | -10.92 | -12.03 | -18.17 | -19.87 | -22.53 | -20.67 | 0.67 |
| | PR-BCD (NA) | -8.18 | -9.53 | -9.18 | -15.02 | -13.18 | -8.37 | -6.55 | -5.90 | -12.52 | -13.65 | -10.27 | -17.43 | -14.47 | -17.03 | -18.87 | 0.27 |
| | SGA | -10.28 | -9.78 | -5.80 | -13.08 | -15.28 | -9.65 | -5.17 | -7.37 | -12.75 | -13.05 | -12.27 | -20.27 | -18.80 | -21.87 | -19.13 | 0.47 |
| | GOttack | -8.80 | -11.00 | -11.93 | -11.07 | -8.77 | -10.00 | -10.83 | -2.17 | -12.80 | -10.23 | -10.53 | -16.13 | -15.80 | -18.53 | -22.40 | -0.33 |
| PNA | $L^1$D-RND | 2.53 | 0.10 | -4.30 | -9.47 | -5.27 | -0.33 | -8.37 | -9.23 | -10.37 | -7.87 | 0.67 | -5.73 | -7.30 | -11.07 | -10.07 | -1.33 |
| | FGA | -6.62 | -5.10 | -8.05 | -4.67 | -0.95 | -11.07 | -17.93 | -6.55 | -6.55 | -8.37 | -11.47 | -15.30 | -18.20 | -16.00 | -16.70 | 0.67 |
| | NETTACK | -4.97 | 3.37 | -1.15 | -2.00 | -3.70 | -11.45 | -9.90 | -7.45 | -7.85 | -12.15 | -11.47 | -12.50 | -15.70 | -11.87 | -10.20 | -0.33 |
| | PGD | -9.20 | -3.48 | -12.78 | -7.03 | -4.35 | -8.22 | -13.55 | -9.78 | -11.80 | -12.22 | -9.63 | -19.40 | -19.80 | -19.23 | -22.90 | 0.13 |
| | PR-BCD (NA) | -12.70 | -6.68 | -10.55 | -11.62 | -8.37 | -7.32 | -7.22 | -3.23 | -11.65 | -6.43 | -9.90 | -14.10 | -19.73 | -16.80 | -17.30 | 0.40 |
| | SGA | -6.02 | -6.23 | -7.08 | -15.85 | -7.72 | -3.73 | -13.23 | -6.32 | -8.00 | -8.12 | -13.23 | -16.77 | -21.37 | -17.37 | -17.90 | 0.13 |
| | GOttack | -7.07 | -6.53 | -7.33 | -7.53 | -8.13 | -17.47 | -2.50 | -9.60 | -6.57 | -15.43 | -11.23 | -13.13 | -19.57 | -16.13 | -20.60 | 0.33 |
| **Poison** GCN | $L^1$D-RND | -2.20 | -3.17 | -0.57 | 0.13 | -3.33 | -4.90 | -0.63 | 6.07 | 0.97 | -1.40 | -6.43 | -9.20 | -11.17 | -10.17 | -10.80 | 0.00 |
| | FGA | -10.00 | -7.00 | -8.23 | -7.22 | -8.43 | -15.88 | -10.63 | -9.58 | -8.37 | -7.02 | -21.17 | -19.70 | -13.83 | -13.13 | -13.83 | 0.67 |
| | NETTACK | -6.78 | 0.37 | -1.08 | -4.87 | -4.18 | -10.60 | -4.52 | -7.38 | -5.98 | -8.17 | -22.03 | -11.53 | -10.70 | -10.77 | -14.10 | -1.67 |
| | PGD | -10.52 | -6.70 | -3.90 | -5.17 | -7.22 | -17.20 | -11.63 | -12.27 | -10.48 | -11.27 | -22.90 | -22.33 | -26.57 | -24.10 | -25.53 | 1.47 |
| | PR-BCD (NA) | -10.20 | -5.13 | -7.23 | -9.33 | -8.10 | -11.98 | -7.10 | -8.88 | -10.35 | -9.77 | -22.57 | -25.97 | -20.03 | -17.20 | -13.10 | 0.00 |
| | SGA | -8.67 | -2.87 | -3.63 | -4.25 | -4.77 | -14.55 | -6.37 | -5.80 | -7.22 | -4.98 | -23.20 | -21.07 | -17.90 | -14.83 | -16.43 | -0.20 |
| | GOttack | -7.83 | -5.37 | -6.93 | -7.10 | -7.47 | -5.57 | -8.60 | -4.63 | -11.17 | -10.00 | -22.73 | -16.57 | -12.77 | -11.20 | -11.97 | -0.27 |
| GIN | $L^1$D-RND | 0.23 | -2.50 | -4.93 | -5.10 | -7.67 | 2.23 | -0.87 | -2.93 | -6.20 | -7.13 | -3.83 | -6.60 | -3.10 | -10.17 | -12.23 | 0.00 |
| | FGA | -7.38 | -9.48 | -9.10 | -7.12 | -5.73 | -9.08 | -9.45 | -8.20 | -6.13 | -5.50 | -15.23 | -11.33 | -16.53 | -6.57 | -11.43 | 0.40 |
| | NETTACK | -6.87 | -1.93 | -7.57 | -8.90 | -11.13 | -0.82 | -2.12 | -2.57 | -3.57 | -9.40 | -11.67 | -12.63 | -10.87 | -11.00 | -9.97 | 0.00 |
| | PGD | -9.93 | -1.90 | -9.57 | -7.65 | -12.20 | -3.83 | -2.73 | -7.42 | -12.28 | -11.98 | -13.60 | -15.50 | -17.70 | -14.93 | -20.70 | 0.07 |
| | PR-BCD (NA) | -3.88 | -2.58 | -3.18 | -7.68 | -7.70 | -9.23 | 1.28 | -3.50 | -8.13 | -7.63 | -11.10 | -19.43 | -20.53 | -18.70 | -17.43 | -0.60 |
| | SGA | -7.92 | -8.17 | -14.75 | -14.85 | -12.17 | -4.85 | -7.07 | -9.42 | -6.75 | -6.98 | -10.60 | -11.90 | -14.17 | -13.60 | -12.30 | 0.20 |
| | GOttack | -4.73 | -1.60 | -9.37 | -8.80 | -6.67 | -2.27 | -7.47 | -4.23 | -6.57 | -8.87 | -14.97 | -9.50 | -13.13 | -13.13 | -17.47 | -0.07 |
| GSAGE | $L^1$D-RND | -1.53 | -1.93 | -1.60 | -1.67 | -2.43 | -4.33 | -3.70 | -2.07 | -4.87 | -7.00 | -3.53 | -9.10 | -7.17 | -9.73 | -10.53 | -0.07 |
| | FGA | -10.83 | -8.87 | -12.33 | -6.93 | -12.13 | -6.15 | -5.45 | -6.50 | -10.80 | -7.30 | -11.50 | -19.13 | -16.60 | -16.13 | -15.60 | 0.13 |
| | NETTACK | -10.68 | -7.23 | -12.15 | -7.92 | -6.82 | -7.57 | -0.70 | -7.73 | -9.22 | -9.05 | -11.77 | -18.47 | -11.23 | -15.30 | -16.40 | 0.13 |
| | PGD | -8.70 | -7.32 | -12.08 | -12.70 | -16.90 | -9.18 | -8.32 | -10.93 | -9.58 | -11.02 | -11.33 | -18.20 | -19.23 | -21.63 | -19.97 | 0.33 |
| | PR-BCD (NA) | -6.70 | -7.22 | -9.87 | -14.50 | -13.53 | -8.78 | -6.63 | -4.17 | -11.02 | -10.58 | -9.63 | -16.93 | -14.70 | -17.00 | -18.77 | -0.13 |
| | SGA | -9.33 | -8.07 | -7.68 | -15.18 | -16.47 | -9.25 | -10.65 | -5.87 | -14.03 | -10.52 | -8.80 | -20.47 | -19.37 | -22.07 | -20.20 | 0.07 |
| | GOttack | -7.53 | -7.97 | -8.80 | -12.53 | -12.10 | -9.27 | -9.37 | -4.97 | -11.80 | -9.53 | -9.37 | -15.47 | -18.17 | -17.97 | -22.97 | -0.47 |
| PNA | $L^1$D-RND | 3.63 | -1.17 | -9.53 | -5.93 | -4.47 | 1.60 | -5.10 | -7.70 | -11.47 | -7.73 | 0.50 | -3.23 | -5.33 | -9.27 | -9.03 | -1.07 |
| | FGA | -2.08 | -5.32 | -3.70 | -4.33 | -3.30 | -5.22 | -12.00 | -6.72 | -5.23 | -5.55 | -13.67 | -16.63 | -20.10 | -18.53 | -19.87 | -0.20 |
| | NETTACK | -3.65 | 1.60 | -1.83 | -0.42 | -5.32 | -12.57 | -6.85 | -12.02 | -5.02 | -7.13 | -13.63 | -11.53 | -15.97 | -13.00 | -10.83 | -0.40 |
| | PGD | -8.03 | -13.15 | -8.18 | -5.38 | -7.78 | -4.13 | -16.52 | -6.75 | -9.10 | -9.52 | -10.10 | -18.43 | -18.83 | -19.13 | -24.57 | 0.60 |
| | PR-BCD (NA) | -10.62 | -2.45 | -11.50 | -7.78 | -11.67 | -8.85 | -3.28 | -4.52 | -8.37 | -8.55 | -10.93 | -14.53 | -19.73 | -18.70 | -17.47 | 0.33 |
| | SGA | -3.53 | -5.73 | -5.72 | -14.22 | -7.00 | -6.80 | -11.18 | -4.45 | -10.42 | -13.68 | -12.47 | -14.90 | -22.40 | -16.13 | -19.93 | 0.87 |
| | GOttack | -6.83 | -1.73 | -7.13 | -4.67 | -5.63 | -12.87 | -5.93 | -9.27 | -5.00 | -8.37 | -10.20 | -13.97 | -20.10 | -14.23 | -19.43 | -0.13 |

Table 30: Effect of model selection on adversarial evaluation results. When evaluating adversarial attacks on victim models with a configuration obtained from model selection, which is higher than evaluating on victim models with a fixed configuration, the results are shown in blue; the results are shown in red otherwise.

| | Dataset | CORA | | | | | CITESEER | | | | | PUBMED | | | | | Avg. Rank Difference |
|---|---|---|---|---|---|---|---|---|---|---|---|---|---|---|---|---|---|
| | Δ → | 1 | 2 | 3 | 4 | 5 | 1 | 2 | 3 | 4 | 5 | 1 | 2 | 3 | 4 | 5 | |
| **Evasion** GCN | L¹D-RND | -2.67 | 2.80 | 0.00 | -0.27 | -2.40 | 4.00 | 6.93 | 2.27 | 1.87 | -1.07 | 0.00 | 0.13 | -4.27 | -9.20 | -5.73 | 0.00 |
| | FGA | 0.67 | 0.80 | -0.93 | 0.27 | 2.13 | -1.33 | -7.20 | -3.47 | -1.07 | -0.67 | -1.07 | -0.27 | -0.53 | 0.40 | -0.67 | 0.00 |
| | NETTACK | 0.27 | -3.07 | -2.27 | -0.27 | 0.53 | -1.87 | -8.80 | -8.13 | -0.53 | -1.20 | -2.80 | 2.93 | -0.40 | 0.27 | -0.13 | 0.33 |
| | PGD | 1.60 | 11.47 | 13.60 | 15.33 | 13.87 | -2.00 | -0.27 | -0.80 | 0.80 | 2.13 | -2.00 | -2.13 | -4.80 | -3.73 | -4.40 | -0.13 |
| | PR-BCD (NA) | 0.40 | -3.33 | -3.33 | -2.00 | -1.07 | 3.07 | 0.00 | 2.27 | 4.27 | 4.00 | -1.60 | -0.53 | 1.87 | 0.67 | -0.67 | 0.00 |
| | SGA | 1.87 | 9.20 | 8.13 | 2.13 | 3.33 | 2.40 | 4.80 | 8.67 | 6.40 | 8.00 | -1.20 | 2.13 | 5.33 | 2.67 | 2.13 | -0.20 |
| | GOttack | -0.93 | -3.20 | -2.53 | -1.07 | -0.67 | -3.47 | -10.80 | -4.53 | -0.67 | -2.93 | -0.80 | 0.80 | 0.53 | 0.67 | 1.33 | 0.00 |
| GIN | L¹D-RND | -8.53 | -7.87 | -9.87 | -13.33 | -16.27 | -8.67 | -9.60 | -9.33 | -9.73 | -9.87 | -4.40 | -10.00 | -6.13 | -10.93 | -10.67 | 2.40 |
| | FGA | -0.53 | -1.47 | 0.67 | 1.07 | 1.60 | -6.93 | 3.07 | 1.20 | -4.27 | -0.67 | -0.53 | 8.40 | 3.60 | 3.73 | -1.60 | -0.27 |
| | NETTACK | -3.07 | -3.07 | -2.40 | -1.47 | 1.60 | -7.07 | -7.20 | -5.60 | 0.13 | 0.00 | 0.40 | 5.73 | -4.67 | -6.27 | -5.20 | 0.20 |
| | PGD | 0.00 | 3.47 | 5.87 | 4.13 | 4.67 | -5.20 | 0.80 | -2.27 | -2.53 | 0.53 | -2.40 | 2.93 | 1.07 | 4.53 | 4.40 | -0.47 |
| | PR-BCD (NA) | 2.13 | 3.60 | 5.20 | 4.93 | 2.67 | -8.00 | 0.27 | -2.93 | -0.93 | -1.47 | -1.87 | -2.00 | -4.80 | -2.00 | -0.53 | -0.67 |
| | SGA | 3.47 | 8.00 | 6.93 | 6.53 | 7.07 | -0.27 | 8.00 | 5.07 | 9.33 | 8.40 | 4.80 | 4.53 | 7.33 | 7.07 | 7.07 | -1.20 |
| | GOttack | -4.67 | -0.67 | -1.07 | -1.20 | -2.27 | -6.93 | 0.67 | 1.47 | 0.00 | -1.47 | -0.53 | 8.80 | 6.67 | 9.73 | 8.67 | 0.00 |
| GSAGE | L¹D-RND | -11.07 | -5.60 | -9.47 | -12.93 | -14.53 | -9.33 | -3.60 | -3.73 | -4.67 | -3.87 | -6.00 | -9.47 | -11.60 | -9.87 | -16.93 | 2.67 |
| | FGA | 0.80 | 1.33 | 1.87 | 3.07 | 4.53 | 2.13 | 8.93 | 7.20 | 9.87 | 8.13 | -7.33 | -3.73 | -5.60 | -12.40 | -15.33 | -0.67 |
| | NETTACK | -2.00 | -0.80 | 2.80 | 4.40 | 3.87 | -2.93 | 4.40 | 1.60 | 5.47 | -1.47 | -6.13 | -5.73 | -14.00 | -15.73 | -18.13 | 0.20 |
| | PGD | 1.33 | 2.13 | 5.87 | 5.07 | 5.60 | 0.00 | 7.87 | 3.60 | 7.47 | 8.53 | -7.60 | -5.60 | -8.13 | -7.73 | -6.13 | 0.00 |
| | PR-BCD (NA) | 2.80 | 2.53 | 2.00 | 1.20 | 1.20 | -2.40 | 6.67 | 6.53 | 3.73 | 5.07 | -4.67 | -6.27 | -8.67 | -8.27 | -8.27 | -0.67 |
| | SGA | 0.93 | 9.87 | 11.87 | 10.27 | 9.20 | 3.07 | 11.20 | 6.93 | 11.33 | 7.87 | -4.80 | -3.73 | -5.60 | -6.80 | -6.13 | -1.33 |
| | GOttack | -1.47 | 0.80 | -1.87 | 2.00 | 0.40 | -1.87 | 6.80 | 12.67 | 10.67 | 13.20 | -5.20 | -5.87 | -7.07 | -5.07 | -7.20 | -0.20 |
| PNA | L¹D-RND | -9.47 | -7.07 | -6.67 | -15.60 | -9.33 | -2.13 | -1.47 | -1.60 | 1.87 | 2.80 | 9.60 | -6.27 | -3.07 | -3.33 | -1.47 | 0.20 |
| | FGA | 1.20 | 7.20 | 3.87 | 6.13 | 4.00 | 2.93 | 4.67 | 5.33 | 0.40 | -0.93 | -4.53 | -0.53 | -3.33 | -6.00 | -6.67 | 0.33 |
| | NETTACK | 1.87 | -3.87 | -5.33 | -4.67 | -2.93 | 3.60 | -4.80 | 1.47 | 0.53 | 0.53 | -1.60 | -5.07 | -9.33 | -7.33 | -6.00 | 0.07 |
| | PGD | 3.87 | 10.53 | 6.40 | 12.67 | 14.27 | 3.60 | 0.13 | 5.33 | 5.73 | 3.87 | -1.47 | -4.53 | -2.93 | -3.07 | -4.00 | 0.07 |
| | PR-BCD (NA) | 0.53 | 6.93 | 6.40 | 4.27 | 8.13 | 3.47 | 1.33 | 1.20 | 0.40 | 3.07 | -8.00 | -4.53 | -5.20 | -4.53 | -2.13 | -0.27 |
| | SGA | 2.40 | 7.73 | 7.33 | 2.27 | 3.47 | 11.07 | 9.33 | 11.87 | 13.73 | 11.20 | -1.60 | -1.73 | -2.93 | -0.40 | -1.07 | -0.80 |
| | GOttack | 4.13 | 2.13 | 3.73 | 5.07 | -1.20 | -1.20 | 7.33 | 5.73 | 3.47 | 3.73 | -4.40 | 0.80 | -1.73 | -0.67 | 0.93 | 0.40 |
| **Poison** GCN | L¹D-RND | -0.93 | 2.53 | 0.40 | -0.27 | 0.13 | 4.53 | 4.13 | 3.20 | 3.87 | -0.53 | -1.47 | 0.13 | -4.13 | -8.80 | -6.00 | 0.00 |
| | FGA | -1.20 | 2.27 | -0.93 | -0.80 | 0.93 | -4.27 | -3.20 | -0.80 | 0.00 | 1.87 | -1.33 | -0.27 | -0.67 | 0.67 | -0.67 | 0.00 |
| | NETTACK | 1.47 | -3.33 | -0.93 | -0.40 | -0.40 | 0.80 | -3.47 | -3.47 | 1.73 | 0.67 | -2.27 | 2.00 | -0.40 | 0.00 | -0.40 | 0.07 |
| | PGD | 3.20 | 11.33 | 14.00 | 14.40 | 13.20 | -1.20 | -0.53 | 0.27 | 3.47 | 4.00 | -2.53 | -2.13 | -5.07 | -3.87 | -4.00 | 0.07 |
| | PR-BCD (NA) | -2.27 | -3.47 | -4.80 | -4.13 | -2.13 | -3.87 | -4.40 | -2.40 | 0.13 | -0.27 | -1.60 | -0.40 | 1.07 | -0.80 | -0.93 | 0.53 |
| | SGA | -0.27 | 3.60 | 3.20 | 1.87 | 4.00 | -3.47 | 5.07 | 6.80 | 2.53 | 5.33 | -0.93 | 1.60 | 4.40 | 2.27 | 1.87 | -0.20 |
| | GOttack | 1.87 | -2.00 | -0.93 | -0.80 | 0.13 | -2.13 | -5.87 | -0.67 | -0.53 | 1.07 | -1.07 | 0.27 | 0.13 | 0.67 | 0.93 | -0.47 |
| GIN | L¹D-RND | -8.27 | -9.07 | -12.40 | -13.20 | -14.67 | -6.13 | -6.40 | -8.67 | -8.93 | -12.13 | -3.47 | -3.60 | -3.07 | -12.00 | -11.87 | 2.67 |
| | FGA | 4.13 | 10.27 | 11.73 | 10.67 | 13.87 | 3.20 | 8.40 | 8.67 | 6.53 | 7.33 | 1.07 | 8.67 | 3.47 | 4.27 | -1.47 | -0.80 |
| | NETTACK | 0.93 | 3.20 | 5.87 | 7.60 | 5.73 | 3.07 | 0.27 | 4.67 | 7.60 | 6.53 | -1.33 | 4.93 | -6.00 | -5.07 | -4.80 | 0.47 |
| | PGD | 1.33 | 7.73 | 10.93 | 11.33 | 9.87 | 1.07 | 8.67 | 5.47 | 4.80 | 8.40 | -1.20 | 3.73 | 1.33 | 6.13 | 4.53 | -0.27 |
| | PR-BCD (NA) | 5.20 | 9.33 | 8.27 | 10.40 | 10.67 | -3.60 | 8.00 | 8.13 | 6.40 | 10.53 | -2.67 | -1.20 | -3.47 | -0.93 | -0.13 | 0.13 |
| | SGA | 9.20 | 15.07 | 14.27 | 18.40 | 20.80 | 6.67 | 14.53 | 15.60 | 19.07 | 19.07 | 3.73 | 7.07 | 7.73 | 6.93 | 6.27 | -1.80 |
| | GOttack | 2.80 | 4.13 | 5.73 | 5.60 | 8.67 | 3.73 | 6.93 | 14.27 | 11.47 | 10.53 | 1.87 | 10.27 | 7.73 | 8.80 | 8.13 | -0.40 |
| GSAGE | L¹D-RND | -10.67 | -8.40 | -9.87 | -12.53 | -13.87 | -8.13 | -3.07 | -3.20 | -3.33 | -3.47 | -6.13 | -10.80 | -11.07 | -14.80 | -17.87 | 3.20 |
| | FGA | 0.67 | 3.87 | 6.67 | 9.07 | 9.20 | 0.80 | 14.00 | 13.73 | 13.73 | 13.47 | -7.60 | -5.60 | -6.13 | -12.40 | -15.33 | -0.73 |
| | NETTACK | 0.40 | 2.53 | 4.40 | 6.40 | 9.33 | -2.27 | 7.73 | 3.47 | 7.20 | 0.67 | -7.07 | -5.73 | -14.53 | -16.40 | -18.53 | 0.13 |
| | PGD | 4.40 | 6.53 | 7.47 | 10.27 | 11.47 | -3.20 | 9.60 | 11.07 | 10.93 | 14.00 | -7.60 | -5.87 | -8.40 | -7.87 | -6.27 | -0.53 |
| | PR-BCD (NA) | 3.20 | 3.20 | 4.13 | 3.60 | 6.13 | -4.27 | 6.80 | 8.53 | 6.40 | 9.33 | -6.13 | -7.20 | -10.27 | -8.67 | -9.07 | 0.00 |
| | SGA | -1.47 | 13.20 | 14.27 | 16.00 | 16.00 | 1.07 | 10.93 | 10.27 | 12.53 | 11.60 | -4.27 | -5.07 | -6.40 | -6.40 | -7.60 | -1.53 |
| | GOttack | 0.27 | 4.40 | 5.87 | 3.47 | 6.13 | -2.00 | 9.07 | 13.60 | 12.27 | 16.13 | -5.87 | -6.67 | -9.07 | -6.53 | -7.73 | -0.53 |
| PNA | L¹D-RND | -5.87 | -5.07 | -7.60 | -10.53 | -5.60 | -1.60 | 0.27 | 2.27 | 2.53 | 4.67 | 10.00 | -5.60 | -1.47 | -3.60 | -1.73 | 1.33 |
| | FGA | 7.73 | 7.33 | 8.27 | 2.00 | 2.00 | 9.07 | 8.93 | 10.93 | 5.60 | 1.60 | -3.87 | -1.07 | -1.87 | -6.93 | -7.60 | -0.67 |
| | NETTACK | 2.27 | -2.13 | -4.93 | -2.40 | -1.33 | 8.40 | 2.80 | 4.40 | 5.33 | 4.80 | -1.47 | -4.80 | -11.20 | -10.67 | -6.13 | 0.13 |
| | PGD | 5.60 | 7.73 | 11.33 | 13.60 | 12.53 | 10.40 | 5.33 | 8.40 | 10.00 | 8.67 | -0.67 | -3.47 | -1.87 | -0.80 | -3.07 | -0.33 |
| | PR-BCD (NA) | 6.13 | 6.93 | 6.13 | 8.27 | 4.53 | 4.00 | 6.80 | 2.93 | 2.27 | 3.20 | -8.80 | -4.13 | -5.60 | -4.27 | -2.00 | 0.13 |
| | SGA | 7.73 | 7.20 | 4.27 | 1.60 | 10.27 | 7.20 | 8.40 | 12.67 | 9.73 | 14.40 | -0.93 | 1.47 | -2.67 | 0.27 | -2.13 | -0.53 |
| | GOttack | 5.73 | 3.07 | 4.80 | 3.47 | -0.27 | 6.13 | 9.60 | 8.00 | 8.67 | 7.07 | -3.47 | -1.20 | -1.33 | 1.47 | 0.80 | -0.07 |

Table 31: Performance of RND attack and our proposed random-based baseline, $L^1$D-RND, on defense and non-defense victim models used in evaluating adversarial attacks on three datasets.

| | | Dataset | CORA | | | | | CITESEER | | | | | PUBMED | | | | |
|---|---|---|---|---|---|---|---|---|---|---|---|---|---|---|---|---|---|
| | | Δ → | 1 | 2 | 3 | 4 | 5 | 1 | 2 | 3 | 4 | 5 | 1 | 2 | 3 | 4 | 5 |
| Evasion | GCN | $L^1$D-RND | 13.20 ± 4.33 | 22.53 ± 6.95 | 29.20 ± 5.60 | 32.93 ± 6.67 | 32.13 ± 5.04 | 15.20 ± 4.20 | 28.67 ± 12.39 | 36.53 ± 9.09 | 37.73 ± 9.44 | 42.13 ± 7.11 | 10.93 ± 3.01 | 15.47 ± 4.87 | 17.20 ± 6.58 | 18.80 ± 5.28 | 23.60 ± 9.63 |
| | | Random | 13.20 ± 4.89 | 20.13 ± 5.04 | 24.27 ± 4.95 | 29.47 ± 5.78 | 30.40 ± 6.90 | 14.00 ± 4.34 | 28.27 ± 8.24 | 35.33 ± 7.84 | 37.47 ± 7.03 | 36.00 ± 8.25 | 10.27 ± 3.69 | 14.40 ± 5.77 | 16.27 ± 3.61 | 17.73 ± 5.55 | 15.07 ± 4.77 |
| | GIN | $L^1$D-RND | 20.13 ± 7.35 | 37.07 ± 15.27 | 42.67 ± 8.80 | 43.20 ± 10.05 | 43.73 ± 12.02 | 16.80 ± 10.87 | 33.47 ± 15.76 | 35.73 ± 14.94 | 38.53 ± 13.17 | 43.33 ± 12.57 | 13.33 ± 5.64 | 23.07 ± 12.89 | 26.67 ± 12.75 | 22.40 ± 10.26 | 22.27 ± 11.54 |
| | | Random | 17.87 ± 5.97 | 36.27 ± 7.40 | 35.33 ± 5.33 | 34.53 ± 7.46 | 37.73 ± 7.55 | 16.40 ± 7.94 | 32.80 ± 11.16 | 34.93 ± 12.40 | 37.07 ± 11.97 | 37.47 ± 12.70 | 11.47 ± 7.03 | 19.87 ± 12.52 | 18.13 ± 9.09 | 17.07 ± 9.47 | 19.33 ± 9.37 |
| | GSAGE | $L^1$D-RND | 15.87 ± 6.25 | 34.00 ± 10.95 | 38.93 ± 7.48 | 40.93 ± 5.70 | 41.87 ± 7.19 | 19.07 ± 7.28 | 38.27 ± 11.56 | 40.40 ± 10.32 | 42.67 ± 13.62 | 45.20 ± 10.98 | 10.00 ± 3.70 | 18.00 ± 5.07 | 16.40 ± 8.04 | 14.27 ± 5.65 | 13.33 ± 7.39 |
| | | Random | 16.00 ± 6.50 | 34.40 ± 4.48 | 33.60 ± 9.14 | 34.93 ± 6.23 | 37.60 ± 6.60 | 18.80 ± 7.55 | 36.53 ± 7.03 | 37.47 ± 9.66 | 40.13 ± 9.61 | 35.07 ± 11.16 | 8.53 ± 3.34 | 14.13 ± 4.10 | 15.73 ± 4.13 | 13.60 ± 5.57 | 13.07 ± 4.27 |
| | PNA | $L^1$D-RND | 33.20 ± 8.81 | 41.60 ± 12.86 | 47.20 ± 9.44 | 45.20 ± 12.14 | 53.07 ± 15.36 | 39.33 ± 11.82 | 46.13 ± 9.84 | 46.27 ± 10.05 | 52.13 ± 10.78 | 55.47 ± 11.12 | 39.33 ± 26.16 | 18.27 ± 8.41 | 19.20 ± 8.34 | 20.27 ± 8.34 | 24.27 ± 12.80 |
| | | Random | 29.47 ± 11.72 | 35.33 ± 9.90 | 31.87 ± 11.17 | 37.20 ± 12.87 | 39.07 ± 10.47 | 32.27 ± 15.56 | 38.00 ± 8.32 | 43.20 ± 6.62 | 44.27 ± 9.16 | 46.40 ± 9.80 | 33.73 ± 25.69 | 13.47 ± 7.27 | 14.27 ± 7.17 | 12.93 ± 6.96 | 12.93 ± 6.67 |
| | GCN-GARNET | $L^1$D-RND | 16.53 ± 11.75 | 37.60 ± 13.86 | 43.73 ± 13.67 | 46.40 ± 16.29 | 51.07 ± 19.20 | 21.33 ± 13.95 | 46.40 ± 15.35 | 49.73 ± 12.02 | 56.67 ± 14.46 | 57.33 ± 15.87 | 18.93 ± 17.84 | 34.27 ± 19.02 | 33.33 ± 21.93 | 37.20 ± 20.07 | 36.00 ± 20.06 |
| | | Random | 13.73 ± 4.33 | 27.87 ± 9.52 | 34.40 ± 7.86 | 39.33 ± 7.20 | 40.13 ± 9.02 | 15.33 ± 5.98 | 37.87 ± 6.25 | 44.93 ± 9.59 | 45.73 ± 9.25 | 45.33 ± 8.20 | 11.47 ± 3.66 | 22.13 ± 11.58 | 28.13 ± 10.84 | 26.93 ± 10.90 | 26.13 ± 10.84 |
| | GNNGuard | $L^1$D-RND | 6.27 ± 4.13 | 9.73 ± 5.60 | 11.73 ± 4.83 | 17.20 ± 5.44 | 18.67 ± 6.13 | 4.67 ± 3.68 | 7.60 ± 5.46 | 9.60 ± 5.30 | 10.53 ± 5.10 | 11.07 ± 5.18 | 6.53 ± 4.63 | 8.27 ± 4.20 | 9.73 ± 4.53 | 10.13 ± 5.68 | 12.40 ± 5.46 |
| | | Random | 5.07 ± 3.84 | 10.27 ± 4.46 | 12.40 ± 3.72 | 16.00 ± 5.29 | 17.33 ± 4.12 | 5.60 ± 4.15 | 10.00 ± 2.83 | 11.07 ± 2.49 | 11.33 ± 2.69 | 12.67 ± 2.99 | 7.20 ± 4.33 | 10.40 ± 3.64 | 10.00 ± 3.63 | 10.93 ± 2.49 | 11.47 ± 3.34 |
| | GRAND | $L^1$D-RND | 34.13 ± 14.99 | 63.47 ± 7.54 | 71.07 ± 6.54 | 71.07 ± 7.70 | 72.00 ± 8.72 | 45.47 ± 11.80 | 60.67 ± 8.64 | 63.33 ± 9.43 | 66.00 ± 8.49 | 66.40 ± 8.72 | 33.20 ± 15.62 | 39.07 ± 17.07 | 44.53 ± 16.41 | 44.93 ± 16.78 | 44.67 ± 17.22 |
| | | Random | 27.33 ± 7.99 | 41.87 ± 7.42 | 40.67 ± 11.48 | 42.93 ± 9.47 | 45.47 ± 9.66 | 34.67 ± 7.20 | 45.47 ± 6.02 | 51.73 ± 4.95 | 53.20 ± 7.00 | 56.13 ± 6.21 | 28.80 ± 13.04 | 34.67 ± 9.99 | 31.33 ± 13.73 | 34.13 ± 13.30 | 34.67 ± 15.81 |
| Poison | GCN | $L^1$D-RND | 15.47 ± 4.03 | 23.33 ± 6.79 | 30.27 ± 6.18 | 33.47 ± 7.03 | 34.67 ± 6.87 | 16.27 ± 4.33 | 29.87 ± 12.77 | 37.73 ± 9.56 | 40.13 ± 9.12 | 42.93 ± 6.04 | 9.73 ± 3.10 | 15.47 ± 4.81 | 17.33 ± 5.64 | 19.33 ± 5.49 | 23.20 ± 9.91 |
| | | Random | 15.33 ± 3.44 | 25.33 ± 7.20 | 28.27 ± 5.28 | 31.47 ± 6.35 | 30.80 ± 5.80 | 16.00 ± 4.07 | 30.80 ± 6.79 | 38.00 ± 6.59 | 38.80 ± 5.06 | 37.33 ± 9.19 | 10.00 ± 4.14 | 14.13 ± 5.63 | 16.53 ± 3.74 | 19.07 ± 4.95 | 15.60 ± 5.46 |
| | GIN | $L^1$D-RND | 23.07 ± 8.07 | 38.00 ± 13.50 | 42.40 ± 8.53 | 43.73 ± 11.54 | 45.33 ± 12.39 | 20.40 ± 9.39 | 36.13 ± 13.13 | 38.40 ± 11.86 | 40.13 ± 11.55 | 43.87 ± 10.54 | 14.67 ± 6.79 | 27.07 ± 14.10 | 29.07 ± 13.75 | 23.33 ± 12.11 | 24.27 ± 12.69 |
| | | Random | 20.00 ± 6.76 | 37.73 ± 7.63 | 37.87 ± 7.15 | 39.73 ± 10.39 | 42.67 ± 9.34 | 21.07 ± 10.02 | 36.67 ± 8.54 | 38.13 ± 10.65 | 42.93 ± 10.08 | 44.80 ± 13.41 | 12.13 ± 6.12 | 20.13 ± 13.21 | 18.00 ± 9.47 | 18.00 ± 10.77 | 20.27 ± 9.50 |
| | GSAGE | $L^1$D-RND | 18.13 ± 6.57 | 33.73 ± 9.71 | 40.40 ± 7.45 | 42.67 ± 7.43 | 44.40 ± 7.06 | 21.33 ± 9.09 | 40.13 ± 10.43 | 42.27 ± 9.76 | 43.47 ± 11.84 | 45.33 ± 8.27 | 10.13 ± 4.44 | 17.73 ± 4.95 | 18.00 ± 8.45 | 14.27 ± 5.50 | 14.80 ± 6.96 |
| | | Random | 17.07 ± 5.06 | 36.13 ± 5.97 | 37.20 ± 7.70 | 38.67 ± 8.84 | 39.87 ± 6.02 | 19.33 ± 7.43 | 37.73 ± 7.17 | 39.33 ± 9.15 | 42.00 ± 9.50 | 38.67 ± 12.82 | 8.93 ± 3.69 | 14.53 ± 4.10 | 14.00 ± 4.84 | 12.80 ± 4.20 | 12.40 ± 5.14 |
| | PNA | $L^1$D-RND | 37.47 ± 7.80 | 44.00 ± 13.65 | 48.13 ± 11.82 | 50.40 ± 13.27 | 57.20 ± 13.26 | 42.27 ± 12.62 | 51.07 ± 10.98 | 52.13 ± 9.96 | 52.53 ± 6.70 | 59.60 ± 9.42 | 39.33 ± 26.37 | 19.60 ± 10.83 | 20.67 ± 8.74 | 20.40 ± 8.69 | 23.47 ± 10.89 |
| | | Random | 36.40 ± 12.63 | 40.27 ± 8.81 | 38.13 ± 7.91 | 41.07 ± 9.85 | 45.87 ± 9.64 | 36.53 ± 15.52 | 42.13 ± 8.09 | 45.87 ± 10.04 | 46.67 ± 10.30 | 49.47 ± 6.95 | 35.60 ± 25.32 | 13.73 ± 6.36 | 14.00 ± 7.45 | 13.33 ± 8.27 | 13.87 ± 6.95 |
| | GCN-GARNET | $L^1$D-RND | 18.67 ± 9.96 | 38.27 ± 13.89 | 44.80 ± 14.34 | 46.13 ± 16.66 | 52.00 ± 18.55 | 20.27 ± 13.89 | 49.47 ± 14.05 | 51.33 ± 11.48 | 58.00 ± 13.82 | 58.27 ± 15.21 | 18.67 ± 18.46 | 34.53 ± 18.63 | 34.67 ± 21.09 | 36.67 ± 19.42 | 38.67 ± 18.36 |
| | | Random | 17.20 ± 4.95 | 31.07 ± 9.94 | 34.27 ± 8.17 | 39.73 ± 7.48 | 40.40 ± 10.43 | 16.27 ± 5.01 | 37.60 ± 7.75 | 46.80 ± 8.31 | 47.07 ± 7.25 | 47.33 ± 8.44 | 12.27 ± 3.20 | 22.27 ± 9.56 | 28.40 ± 11.17 | 27.73 ± 10.98 | 26.13 ± 9.66 |
| | GNNGuard | $L^1$D-RND | 6.93 ± 4.40 | 10.40 ± 5.14 | 12.80 ± 3.76 | 18.00 ± 4.90 | 19.47 ± 5.88 | 4.80 ± 3.84 | 7.87 ± 5.53 | 9.87 ± 5.21 | 10.53 ± 4.93 | 11.47 ± 4.69 | 6.53 ± 4.93 | 8.00 ± 4.21 | 9.33 ± 3.90 | 10.40 ± 5.96 | 12.53 ± 5.48 |
| | | Random | 5.87 ± 4.31 | 12.00 ± 3.93 | 14.13 ± 3.89 | 17.47 ± 5.68 | 20.27 ± 4.95 | 5.60 ± 4.15 | 10.40 ± 2.85 | 11.73 ± 2.25 | 12.00 ± 2.93 | 14.00 ± 3.21 | 7.20 ± 4.13 | 11.07 ± 3.69 | 10.93 ± 3.53 | 11.33 ± 2.89 | 11.73 ± 3.20 |
| | GRAND | $L^1$D-RND | 36.80 ± 10.84 | 64.00 ± 9.97 | 69.87 ± 7.27 | 67.33 ± 7.04 | 66.13 ± 8.47 | 46.13 ± 11.45 | 61.47 ± 7.46 | 64.27 ± 8.31 | 64.13 ± 7.91 | 64.67 ± 8.16 | 34.27 ± 13.92 | 40.40 ± 16.79 | 46.27 ± 13.35 | 45.73 ± 14.12 | 45.20 ± 12.30 |
| | | Random | 32.40 ± 7.79 | 47.33 ± 6.58 | 45.87 ± 9.02 | 46.67 ± 10.60 | 50.00 ± 9.29 | 36.27 ± 7.63 | 46.67 ± 6.58 | 51.47 ± 5.10 | 52.13 ± 6.48 | 55.87 ± 6.02 | 29.87 ± 10.43 | 37.07 ± 9.68 | 32.53 ± 13.13 | 33.07 ± 13.13 | 34.40 ± 16.20 |

Table 32: **Vanilla Homophily Results.** Evaluating six adversarial attacks( **features attack variants**) on four vanilla attack victim models - Miss-classification rate (%). Higher is better. Best performance in **bold**, second best underlined

| | | Dataset | CORA | | | | | CITESEER | | | | | PUBMED | | | | | Avg. Rank |
|---|---|---|---|---|---|---|---|---|---|---|---|---|---|---|---|---|---|---|
| | | $\Delta \rightarrow$ | 1 | 2 | 3 | 4 | 5 | 1 | 2 | 3 | 4 | 5 | 1 | 2 | 3 | 4 | 5 | |
| Evasion | GCN | FGA | 27.20 ± 4.13 | 46.80 ± 6.27 | 56.40 ± 3.87 | 61.87 ± 3.81 | 64.13 ± 4.50 | 24.00 ± 2.73 | 34.27 ± 6.23 | 51.20 ± 7.51 | 52.80 ± 9.22 | 60.67 ± 3.18 | **36.80 ± 3.53** | 49.33 ± 7.24 | 56.00 ± 3.70 | 57.07 ± 2.81 | **59.20 ± 1.26** | 2.80 |
| | | NETTACK | 29.33 ± 4.94 | 49.87 ± 5.68 | 57.07 ± 5.44 | **62.13 ± 4.75** | 63.33 ± 4.51 | 25.07 ± 6.04 | 45.73 ± 8.28 | 54.13 ± 5.37 | 57.73 ± 3.84 | 59.20 ± 8.84 | 35.20 ± 2.60 | **54.27 ± 2.71** | **58.27 ± 1.83** | **59.07 ± 1.98** | 58.93 ± 1.83 | 2.00 |
| | | PGD | 29.60 ± 4.42 | 47.07 ± 4.20 | 55.87 ± 4.56 | 57.20 ± 4.46 | 59.07 ± 4.06 | 27.73 ± 5.12 | 40.53 ± 8.23 | 46.13 ± 7.11 | 52.93 ± 5.70 | 54.93 ± 6.27 | 35.07 ± 2.37 | 38.80 ± 4.33 | 43.20 ± 4.52 | 45.47 ± 6.57 | 46.27 ± 5.34 | 3.87 |
| | | PR-BCD | **31.07 ± 3.20** | **50.93 ± 6.27** | **58.80 ± 2.60** | 61.60 ± 3.14 | **66.00 ± 3.30** | **27.87 ± 4.87** | **47.60 ± 5.25** | **54.53 ± 5.97** | **59.20 ± 4.89** | **63.20 ± 3.36** | 30.13 ± 2.56 | 35.73 ± 3.61 | 42.93 ± 4.53 | 49.60 ± 4.15 | 53.33 ± 4.39 | 2.33 |
| | | SGA | 28.53 ± 4.98 | 44.53 ± 7.15 | 54.67 ± 4.94 | 58.27 ± 3.92 | 61.60 ± 4.36 | 22.67 ± 2.79 | 34.93 ± 4.89 | 46.53 ± 4.50 | 50.67 ± 6.70 | 57.20 ± 4.77 | 35.07 ± 2.71 | 40.13 ± 7.19 | 49.87 ± 7.54 | 54.67 ± 5.00 | 56.00 ± 4.96 | 4.00 |
| Poison | GCN | FGA | 29.20 ± 4.39 | 49.07 ± 5.39 | 57.47 ± 3.58 | **62.13 ± 3.58** | 64.00 ± 4.72 | 31.73 ± 6.27 | 49.87 ± 6.35 | 55.87 ± 3.42 | **61.73 ± 4.83** | 62.80 ± 3.76 | **37.20 ± 3.69** | 49.33 ± 6.58 | 56.80 ± 2.91 | 56.80 ± 2.70 | **59.07 ± 1.67** | 2.53 |
| | | NETTACK | 33.47 ± 4.69 | **52.40 ± 4.36** | **58.93 ± 2.91** | 62.00 ± 3.70 | 63.33 ± 3.35 | **34.13 ± 6.95** | **52.40 ± 5.08** | 58.13 ± 5.78 | **61.73 ± 5.90** | 63.20 ± 6.04 | 34.93 ± 2.91 | **53.87 ± 3.96** | **58.27 ± 1.83** | **58.93 ± 1.67** | 58.80 ± 1.82 | 1.67 |
| | | PGD | 30.27 ± 3.92 | 50.00 ± 4.07 | 56.13 ± 4.63 | 58.13 ± 4.69 | 59.47 ± 3.42 | 33.07 ± 6.76 | 46.40 ± 8.04 | 51.47 ± 6.12 | 56.27 ± 5.06 | 59.33 ± 5.33 | 34.80 ± 3.19 | 38.80 ± 4.26 | 42.93 ± 4.13 | 46.13 ± 6.57 | 45.73 ± 5.34 | 4.27 |
| | | PR-BCD | **34.13 ± 5.26** | 52.40 ± 5.30 | 58.00 ± 3.70 | 60.93 ± 3.01 | **64.40 ± 2.95** | 33.20 ± 4.95 | 51.87 ± 4.31 | **59.20 ± 4.26** | 60.80 ± 4.33 | **64.67 ± 5.54** | 30.67 ± 2.79 | 36.67 ± 3.68 | 43.33 ± 4.58 | 48.93 ± 3.61 | 53.20 ± 5.12 | 2.67 |
| | | SGA | 29.87 ± 5.37 | 50.53 ± 6.25 | 55.87 ± 4.44 | 60.80 ± 3.10 | 62.40 ± 4.67 | 27.47 ± 4.63 | 43.20 ± 3.53 | 51.20 ± 3.53 | 53.47 ± 6.25 | 60.67 ± 5.11 | 35.07 ± 2.91 | 40.67 ± 7.70 | 49.60 ± 7.60 | 55.33 ± 5.00 | 55.87 ± 4.63 | 3.87 |

Table 33: **Ablation study**.Evaluating two variants of $L^1$D-RND: $L^1$D-RND- Add only ( Degree effect) and $L^1$D-RND- Remove only ($\mathcal{L}_1$ effect) on PNA and GRAND uner CORA dataset. Best performance in **bold**, second best underlined

| Dataset | | $\Delta \rightarrow$ | 1 | 2 | 3 | 4 | 5 | Avg. Rank |
|---------|-----|----------------------|---|---|---|---|---|-----------|
| Evasion | PNA | Random | $29.47 \pm 11.72$ | $35.33 \pm 9.90$ | $31.87 \pm 11.17$ | $37.20 \pm 12.87$ | $39.07 \pm 10.47$ | 4.00 |
| | | $L^1$D-RND- Add only | $31.73 \pm 11.00$ | $46.27 \pm 18.00$ | $51.07 \pm 15.69$ | $\mathbf{49.07 \pm 15.34}$ | $\mathbf{53.60 \pm 11.89}$ | 1.80 |
| | | $L^1$D-RND- Remove only | $32.40 \pm 12.49$ | $\mathbf{46.80 \pm 17.64}$ | $\mathbf{53.87 \pm 16.34}$ | $47.20 \pm 14.16$ | $51.87 \pm 10.99$ | 1.80 |
| | | $L^1$D-RND | $\mathbf{33.20 \pm 8.81}$ | $41.60 \pm 12.86$ | $47.20 \pm 9.44$ | $45.20 \pm 12.14$ | $53.07 \pm 15.36$ | 2.40 |
| | GRAND | Random | $27.33 \pm 7.99$ | $41.87 \pm 7.42$ | $40.67 \pm 11.48$ | $42.93 \pm 9.47$ | $45.47 \pm 9.66$ | 4.00 |
| | | $L^1$D-RND- Add only | $\mathbf{62.80 \pm 7.92}$ | $\mathbf{70.53 \pm 8.05}$ | $69.60 \pm 10.26$ | $68.27 \pm 12.14$ | $64.93 \pm 13.69$ | 1.60 |
| | | $L^1$D-RND- Remove only | $62.80 \pm 7.92$ | $70.53 \pm 8.05$ | $69.60 \pm 10.26$ | $68.27 \pm 12.14$ | $64.93 \pm 13.69$ | 2.60 |
| | | $L^1$D-RND | $34.13 \pm 14.99$ | $63.47 \pm 7.54$ | $\mathbf{71.07 \pm 6.54}$ | $\mathbf{71.07 \pm 7.70}$ | $\mathbf{72.00 \pm 8.72}$ | 1.80 |
| Poison | PNA | Random | $36.40 \pm 12.63$ | $40.27 \pm 8.81$ | $38.13 \pm 7.91$ | $41.07 \pm 9.85$ | $45.87 \pm 9.64$ | 3.80 |
| | | $L^1$D-RND- Add only | $\mathbf{39.20 \pm 12.23}$ | $\mathbf{49.60 \pm 14.68}$ | $\mathbf{52.93 \pm 12.12}$ | $54.40 \pm 12.26$ | $54.93 \pm 13.58$ | 1.40 |
| | | $L^1$D-RND- Remove only | $34.27 \pm 12.58$ | $48.80 \pm 14.92$ | $49.33 \pm 15.17$ | $\mathbf{56.00 \pm 14.70}$ | $54.67 \pm 10.30$ | 2.40 |
| | | $L^1$D-RND | $37.47 \pm 7.80$ | $44.00 \pm 13.65$ | $48.13 \pm 11.82$ | $50.40 \pm 13.27$ | $\mathbf{57.20 \pm 13.26}$ | 2.40 |
| | GRAND | Random | $32.40 \pm 7.79$ | $47.33 \pm 6.58$ | $45.87 \pm 9.02$ | $46.67 \pm 10.60$ | $50.00 \pm 9.29$ | 4.00 |
| | | $L^1$D-RND- Add only | $\mathbf{64.93 \pm 7.36}$ | $\mathbf{68.00 \pm 7.48}$ | $66.67 \pm 10.52$ | $69.20 \pm 10.66$ | $66.93 \pm 12.14$ | 1.20 |
| | | $L^1$D-RND- Remove only | $64.93 \pm 7.36$ | $68.00 \pm 7.48$ | $66.67 \pm 10.52$ | $69.20 \pm 10.66$ | $66.93 \pm 12.14$ | 2.20 |
| | | $L^1$D-RND | $36.80 \pm 10.84$ | $64.00 \pm 9.97$ | $\mathbf{69.87 \pm 7.27}$ | $67.33 \pm 7.04$ | $66.13 \pm 8.47$ | 2.60 |

The Dataset header spans the columns 1–5 with the value CORA.