# OpenReview forum: "Adversarial Graph Neural Network Benchmarks: Towards Practical and Fair Evaluation"
_ICLR.cc/2026/Conference — Submitted to ICLR 2026_

### Official Review · Reviewer_byfG · 2025-10-27

**Soundness:** 3
**Presentation:** 3
**Contribution:** 2
**Rating:** 6
**Confidence:** 3

**Summary:**

The paper proposes a new benchmarking protocol for adversarial robustness in GNNs. The authors argue that the lack of standardized evaluation protocols in existing literature has led to inconsistent and potentially misleading conclusions about the robustness of GNNs. To address this, they propose a rigorous and unified framework and use it to re-evaluate seven popular adversarial attacks and eight defense mechanisms across six datasets. The study also introduces a simple baseline attack, L1D-RND, which achieves surprisingly competitive results,

**Strengths:**

The analysis of target node selection is particularly impactful, demonstrating that high-degree nodes are significantly more robust to attacks. The introduction of the L¹D-RND baseline is a simple but powerful methodological choice. Its strong performance against both vanilla and defended models serves as a crucial sanity check.

**Weaknesses:**

While the topic is suitable for NeurIPS/benchmarks, some framing still reads like a broader methods survey plus pipeline paper.

**Questions:**

Q1: The performance of the naive L¹D-RND baseline is surprisingly strong, especially against defended models. Do you have a hypothesis for why this simple, non-optimized method is so effective?

Q2: The paper shows that the performance ranking of attacks like Nettack and FGA changes dramatically between homophilic and heterophilic datasets. Does your analysis provide any intuition why?

---

> ### Author Response · Authors · 2025-11-20
>
> We thank the reviewer for taking the time to assess our paper and for the constructive suggestions. It's rewarding to see the recognition of key aspects of our work, e.g., that the analysis of target node selection is particularly impactful and the L1D-RND baseline is a simple yet powerful method that serves as a crucial sanity check. In the following, we will reply to the reviewers' questions/comments and have revised our manuscript accordingly, with changes highlighted in brown.
>
> ---
> ### W1. Some framing still reads like a broader methods survey plus pipeline paper.
>
>
> > While the topic is suitable for NeurIPS/benchmarks, some framing still reads like a broader methods survey plus pipeline paper.
>
>
>
> **Response:** We thank the reviewer for this comment. ICLR actually has a benchmark track, and in fact, we have selected it as our primary submission area (please see the primary area field at the top left of this page). Our main contributions, standardized evaluation methodology, large-scale experimental design, and comprehensive benchmarking, best align with this track.
>
> While our primary focus is on evaluation, our work also finds several important insights that challenge prevailing assumptions in prior studies. For example, we show that performing model selection on both surrogate and victim models, as would be done in practice, substantially changes attack performance (Table 15 and Figure 4). As reviewers acknowledge, we also find that high-degree nodes are significantly harder to attack (Table 14 and Figure 4), yet are often ignored in earlier evaluations, leading to overly optimistic conclusions. In addition, we introduce L1D-RND, a simple yet surprisingly strong and cost-efficient baseline that rivals state-of-the-art attacks.

---

> ### Author Response · Authors · 2025-11-20
>
> ### Q1. hypothesis  on why L¹D-RND
>
>
> > The performance of the naive L¹D-RND baseline is surprisingly strong, especially against defended models. Do you have a hypothesis for why this simple, non-optimized method is so effective?
>
>
> **Response:** Thank you for your insightful observation. We share your surprise at the strong performance of L1D-RND and have conducted an ablation study to better understand the phenomenon. In this ablation study, we isolate the effect of each component by turning off either addition (lines 9-15) and deletion (lines 17-23) parts. We conduct experiments with these variants of L1D-RND on one vanilla model, PNA, and one defense model, GRAND, where L1D-RND exhibits the greatest effect. The results are presented in the following table:
>
> #### Table: Evaluating two variants of  L1D-RND:  L1D-RND - Add only ( Degree effect) and  L1D-RND - Remove only ($\mathcal{L}_1$ effect) on PNA and GRAND uner Cora datase (Misclassification rate %). Best in bold and the second best in italic.
>
> | Setting | Model | Variant | Δ=1 | Δ=2 | Δ=3 | Δ=4 | Δ=5 | Avg. Rank |
> |--------|--------|---------|------|------|------|------|------|-----------|
> | **Evasion** | **PNA** | Random | 29.47 ± 11.72 | 35.33 ± 9.90 | 31.87 ± 11.17 | 37.20 ± 12.87 | 39.07 ± 10.47 | 4.00 |
> | | | L1D-RND – Add only | 31.73 ± 11.00 | *46.27 ± 18.00* | *51.07 ± 15.69* | **49.07 ± 15.34** | **53.60 ± 11.89** | 1.80 |
> | | | L1D-RND – Remove only | *32.40 ± 12.49* | **46.80 ± 17.64** | **53.87 ± 16.34** | *47.20 ± 14.16* | 51.87 ± 10.99 | 1.80 |
> | | | L1D-RND | **33.20 ± 8.81** | 41.60 ± 12.86 | 47.20 ± 9.44 | 45.20 ± 12.14 | *53.07 ± 15.36* | 2.40 |
> | **Evasion** | **GRAND** | Random | 27.33 ± 7.99 | 41.87 ± 7.42 | 40.67 ± 11.48 | 42.93 ± 9.47 | 45.47 ± 9.66 | 4.00 |
> | | | L1D-RND – Add only | **62.80 ± 7.92** | **70.53 ± 8.05** | *69.60 ± 10.26* | *68.27 ± 12.14* | *64.93 ± 13.69* | 1.60 |
> | | | L1D-RND – Remove only | *62.80 ± 7.92* | *70.53 ± 8.05* | 69.60 ± 10.26 | 68.27 ± 12.14 | 64.93 ± 13.69 | 2.60 |
> | | | L1D-RND | 34.13 ± 14.99 | 63.47 ± 7.54 | **71.07 ± 6.54** | **71.07 ± 7.70** | **72.00 ± 8.72** | 1.80 |
> | **Poison** | **PNA** | Random | 36.40 ± 12.63 | 40.27 ± 8.81 | 38.13 ± 7.91 | 41.07 ± 9.85 | 45.87 ± 9.64 | 3.80 |
> | | | L1D-RND – Add only | **39.20 ± 12.23** | **49.60 ± 14.68** | **52.93 ± 12.12** | *54.40 ± 12.26* | *54.93 ± 13.58* | 1.40 |
> | | | L1D-RND – Remove only | 34.27 ± 12.58 | *48.80 ± 14.92* | *49.33 ± 15.17* | **56.00 ± 14.70** | 54.67 ± 10.30 | 2.40 |
> | | | L1D-RND | *37.47 ± 7.80* | 44.00 ± 13.65 | 48.13 ± 11.82 | 50.40 ± 13.27 | **57.20 ± 13.26** | 2.40 |
> | **Poison** | **GRAND** | Random | 32.40 ± 7.79 | 47.33 ± 6.58 | 45.87 ± 9.02 | 46.67 ± 10.60 | 50.00 ± 9.29 | 4.00 |
> | | | L1D-RND – Add only | **64.93 ± 7.36** | **68.00 ± 7.48** | *66.67 ± 10.52* | **69.20 ± 10.66** | **66.93 ± 12.14** | 1.20 |
> | | | L1D-RND – Remove only | *64.93 ± 7.36* | *68.00 ± 7.48* | 66.67 ± 10.52 | *69.20 ± 10.66* | *66.93 ± 12.14* | 2.20 |
> | | | L1D-RND | 36.80 ± 10.84 | 64.00 ± 9.97 | **69.87 ± 7.27** | 67.33 ± 7.04 | 66.13 ± 8.47 | 2.60 |
>
>
> Our hypothesis is that the success of L1D-RND attack, especially against defended models, arises from two main factors:
>
> First is bias toward high-degree nodes. Even though L1D-RND is simple, its edge selection is not purely random. On the addition side, it connects the target node to a high-degree node (line 13 in Alg. 3). On the removal side, it prunes based on an $L_1$-norm-based influence measure (Alg. 4), which correlates with local feature aggregation. This introduces a structure-aware perturbation mechanism, despite the absence of gradient-based optimization. Our ablation shows that both components, when used independently, are sufficient to create harmful structural noise.
>
> The second factor is the mismatch between attack complexity and defense generalization. Existing defenses often assume strong adversarial priors (e.g., feature gradients, link importance). These may fail to generalize against structurally simple, diverse attacks. L1D-RND (because it doesn’t follow conventional optimization) creates perturbations that may fall outside the expected adversarial space, allowing it to bypass some defenses.
>
> The ablation table supports this: L1D-RND-Add Only and  L1D-RND-Remove Only consistently outperform the fully randomized baseline and sometimes even the full L1D-RND. In particular, Add Only is dominant in many settings, suggesting that the influence of hub-attachment alone can severely distort message-passing dynamics.
>
> We have revised the manuscript to include the results in Table 33 and the corresponding discussion in Appendix I (highlighted in brown).

---

> ### Author Response · Authors · 2025-11-20
>
> ### Q2.Nettack and FGA changes dramatically between homophilic and heterophilic datasets
>
>
> > The paper shows that the performance ranking of attacks like Nettack and FGA changes dramatically between homophilic and heterophilic datasets. Does your analysis provide any intuition why?
>
>
> **Response:**
> Thank you for this insightful question. We attribute the ranking shifts in Nettack and FGA between homophilic and heterophilic datasets to their reliance on neighbourhood label similarity. These attacks assume that manipulating local structure, adding or removing edges,  will distort a node’s aggregated neighbourhood signal. In homophilic graphs, this assumption holds, so such attacks are highly effective. However, in heterophilic graphs where neighbours often belong to different classes, this assumption breaks down, and local structural perturbations lose their impact. As a result, Nettack and FGA underperform, while structure-agnostic methods perform relatively better. This may explain the dramatic change in relative effectiveness across graph types.

---

> ### Author Response · Authors · 2025-11-27
>
> Thank you once again for your valuable time and insightful feedback, which provides valuable guidance for refining our work. If there are any further points you would like us to clarify, we would be more than happy to address them.

---

### Official Review · Reviewer_GT8F · 2025-10-30

**Soundness:** 3
**Presentation:** 2
**Contribution:** 2
**Rating:** 4
**Confidence:** 3

**Summary:**

This paper presents a large-scale benchmark study for adversarial robustness in Graph Neural Networks (GNNs). The authors re-evaluate seven commonly used adversarial attacks (e.g., Nettack, FGA, PR-BCD, SGA, GOttack) and eight defense methods under unified experimental conditions, spanning six datasets (Cora, Citeseer, Pubmed, Chameleon, Squirrel, OGB-Arxiv). The study performs over 437,000 experiments to identify inconsistencies and biases in previous literature and proposes a standardized evaluation framework for fair, reproducible comparison of adversarial GNN methods.
Notably, the paper introduces a simple baseline attack, L1D-RND, which surprisingly achieves competitive performance, revealing that many existing complex attacks might be over-claimed.

**Strengths:**

* Strong motivation and clarity of purpose. The authors clearly identify the reproducibility and fairness problems in existing adversarial GNN research.

* Massive and rigorous experimental effort. Over 400k runs under controlled conditions is impressive, and the inclusion of both evasion and poisoning settings makes the study comprehensive.

* Methodological contribution. The risk-assessment-based evaluation pipeline, incorporating model selection and random splits, significantly improves fairness and reliability.

* Insightful analysis. The findings on the effects of node degree, model selection, and dataset type (homophily vs. heterophily) are valuable and highlight overlooked pitfalls in prior works.

**Weaknesses:**

* Limited novelty beyond benchmarking. The work does not propose new algorithms or defense strategies; its main contribution lies in evaluation methodology rather than new technical innovation.

* Analysis depth on defenses is limited. While the results include many defenses (e.g., GNNGuard, RUNG), the paper does not provide deeper insights into why some defenses succeed or fail under different graph structures.

* Overemphasis on L1D-RND baseline. Although the naive attack’s competitiveness is an interesting observation, it occupies a disproportionate part of the discussion, while more meaningful interpretive analysis could be expanded.

* Scalability constraints remain. The paper points out that most existing attacks cannot scale beyond moderate graph sizes, yet it does not propose or analyze new scalable solutions.

* Writing issues in some sections. Certain parts of the experiments and tables are overly dense, making the presentation hard to follow; the narrative could be more concise and reader-friendly.

**Questions:**

See weaknesses

---

> ### Author Response · Authors · 2025-11-20
>
> We thank the reviewer for taking the time to assess our paper and for the constructive suggestions. We appreciate the reviewer’s recognition that our risk-assessment-based evaluation pipeline, incorporating model selection and random splits, enhances fairness, reliability, and the comprehensiveness of our study through large-scale, rigorous experimentation. Below, we address each of the reviewer’s comments in detail and have revised our manuscript accordingly, with changes highlighted in magenta (and violet for Weakness 4).
>
> ---
> ### W1. Limited novelty beyond benchmarking
>
> > Limited novelty beyond benchmarking. The work does not propose new algorithms or defense strategies; its main contribution lies in evaluation methodology rather than new technical innovation.
>
>
> **Response:** We thank the reviewer for allowing us to clarify the purpose of our work. We have selected "Datasets and Benchmarks" as the primary submission area, as we believe our paper’s main contributions, standardized evaluation methodology, large-scale experimental design, and comprehensive benchmarking, best align with this track.
>
> While our focus is on evaluation, we introduce novel key insights that challenge assumptions in prior work. We show that performing model selection on both surrogate and victim models, as done in practice, significantly changes attack performance (Table 30 and Figure 7). We also reveal that high-degree nodes are much harder to attack (Table 29 and Figure 7) yet are often overlooked in prior evaluations, leading to inflated performance claims. We also propose L1D-RND, a simple yet surprisingly strong and cost-efficient baseline attack that rivals state-of-the-art attacks. Finally, our results highlight that choices such as target node selection and training procedures can drastically alter conclusions about attack effectiveness. We would like to argue that these are valuable and necessary insights a benchmarking paper should provide, rather than providing a new method or algorithm whose analysis would require a separate full paper in itself.

---

> ### Author Response · Authors · 2025-11-20
>
> ### W2. In-depth analysis for defense
>
> > Analysis depth on defenses is limited ... the paper does not provide deeper insights into why some defenses succeed or fail under different graph structures.
>
> **Response:** We thank the reviewer for pointing this out. In response to this comment, we have added Tables 7 and 8 to our revised manuscript to provide an interpretation of the success and failure of defense under different settings as follows:
>
> #### Table 7: Defenses and their average success rates (evasion setting) under budget 1 across five datasets. Best in bold and the second best in italics
> |Taxonomy|Subcategory|Type|Selected Defense|Cora|Citeseer|Pubmed|Squirrel|Chameleon|
> |--------|-----------|----|---------------|----|--------|------|--------|---------|
> |Improving graph|Unsupervised||Jaccard-GCN|20.81±3.94|19.58±3.53|*28.49±7.30*|--|--|
> |Improving graph|Supervised||GARNET|18.91±2.50|20.01±3.87|16.03±1.82|12.95±7.05|13.73±5.80|
> |Improving training|||GRAND|16.26±7.33|20.47±10.29|20.40±5.30|4.38±3.00|2.28±1.03|
> |Improving training|||GCORN|22.26±2.76|19.48±3.30|OOR|18.11±8.14|15.27±9.58|
> |Improving training|||NoisyGNN|*23.73±2.42*|**26.97±4.05**|16.05±10.88|**57.38±17.40**|**45.86±21.98**|
> |Improving architecture|Adaptively weighting edges|Rule-based|GNNGuard|7.36±0.84|3.90±0.70|3.77±1.23|*31.36±6.95*|0.00±0.00|
> |Improving architecture|Adaptively weighting edges|Probabilistic|RobustGCN|**26.76±4.88**|*21.96±4.89*|**31.40±9.20**|26.30±12.28|*25.00±10.93*|
> |Improving architecture|Adaptively weighting edges|Robustagg.|ElasticGNN|22.47±4.02|21.33±3.04|22.17±4.77|15.56±6.22|12.97±4.44|
> |Improving architecture|Adaptively weighting edges||RUNG|18.64±2.21|18.61±2.51|OOR|1.65±0.74|1.27±0.73|
>
> #### Table 8:  Defenses and their average success rates (poison setting) under budget 1 across five datasets. Best in bold and the second best in italics
> |Taxonomy|Subcategory|Type|Selected Defense|Cora|Citeseer|Pubmed|Squirrel|Chameleon|
> |--------|-----------|----|---------------|----|--------|------|--------|---------|
> |Improving graph|Unsupervised||Jaccard-GCN*|24.03±4.26|26.72±6.06|*28.95±7.30*|--|--|
> |Improving graph|Supervised||GARNET|20.51±2.32|21.51±3.65|16.45±1.83|21.92±9.53|26.11±9.24|
> |Improving training|||GRAND|20.78±6.61|23.54±9.26|20.55±5.82|9.52±3.59|4.81±1.58|
> |Improving training|||GCORN|22.66±2.71|19.97±3.08|OOR|19.90±8.70|16.49±9.59|
> |Improving training|||NoisyGNN|*26.07±2.00*|**29.46±3.18**|15.98±10.86|**61.77±17.79**|**49.80±21.90**|
> |Improving architecture|Adaptively weighting edges|Rule-based|GNNGuard|8.22±1.25|4.45±0.91|5.06±0.71|31.36±6.95|0.00±0.00|
> |Improving architecture|Adaptively weighting edges|Probabilistic|RobustGCN|**30.15±5.76**|27.19±7.10|**31.19±9.10**|*33.71±11.92*|*31.58±10.94*|
> |Improving architecture|Adaptively weighting edges|Robust agg.|ElasticGNN|25.67±4.91|*27.27±5.05*|21.18±4.46|25.14±8.85|22.89±7.64|
> |Improving architecture|Adaptively weighting edges||RUNG|18.81±2.20|18.95±2.27|OOR|13.63±5.29|12.32±2.67|
>
> Similarity-based pruning methods such as GCN-Jaccard and GNNGuard collapse under heterophily because neighbouring nodes are inherently dissimilar. For instance, on CHAMELEON, all attack models produce a misclassification rate of 0.00 under GNNGuard as the attack budget increases, since GNNGuard prunes nearly all edges in heterophilous graphs (refer Table 24). Training-based defenses that rely on feature smoothness, such as GRAND, also deteriorate sharply under heterophily. In the evasion setting, GRAND drops from 19.05% on homophily graphs to 3.33%  on heterophily graphs, a 471.40% relative decrease. In the poisoning setting, it falls from 21.63% to 7.17%, corresponding to a 201.56% relative decrease, revealing its sensitivity to structure–feature mismatch. Defenses like RUNG and GCORN also become unstable when homophily is low. RUNG decreased from 18.63% to 1.47% in evasion and from 18.88% to 12.98% in poisoning. GCORN also drops from 20.88% to 16.70% in evasion and 21.32% to 18.20% in poisoning, highlighting their struggles to generalize beyond homophilous settings.
>
> In contrast, defenses that make fewer homophily assumptions such as RobustGCN, ElasticGNN, and to some extent, GARNET exhibit more stable behavior across diverse datasets. In the poisoning setting, GARNET improves from 19.47% to  24.02%, yielding an 18.96% gain. RobustGCN remains steady, shifting only slightly from 26.71% to  25.66% in evasion and increasing from 29.52% to  32.65% in poisoning. Meanwhile, ElasticGNN, although experiencing some degradation from 21.99% to 14.27% in evasion and 24.71% to  24.02% in poisoning still performs more reliably than heavily homophily-dependent defenses. These patterns collectively confirm that defenses grounded in rigid smoothing or homophily assumptions tend to fail on heterophily graphs, whereas structurally flexible defenses remain more resilient.
>
> We have revised our manuscript, Appendix F.2 (highlighted in magenta), to incorporate this analysis on defense models.

---

> ### Author Response · Authors · 2025-11-20
>
> ---
> ### W3. analysis  on  L1D-RND baseline
>
> > Overemphasis on L1D-RND baseline. Although the naive attack’s competitiveness is an interesting observation, it occupies a disproportionate part of the discussion, while more meaningful interpretive analysis could be expanded.
>
> **Response:** Thank you for the review. We actually use L1D-RND as an ersatz to argue that existing attacks may not be justified in terms of their computational costs. In a sense, this naive baseline allows us to point out deficiencies of existing methods. Our goal was to use it as a diagnostic tool that exposes how little additional benefit more elaborate attacks sometimes bring once computation is taken into account.
>
> Based on your review, we will move some of the detailed numbers to the appendix, and reallocate space to more interpretive analysis. In particular, we will focus the main text on what the stronger attacks are actually changing in the explanation subgraphs, how these changes relate to misclassification, and when the extra computational cost yields qualitatively different perturbations rather than small quantitative gains over the naive strategy.

---

> ### Author Response · Authors · 2025-11-20
>
> ### W4. new scalable solutions
>
> > Scalability constraints remain. The paper points out that most existing attacks cannot scale beyond moderate graph sizes, yet it does not propose or analyze new scalable solutions
>
>
> **Response:** We thank the reviewers for highlighting this point. As mentioned before, our work focuses on establishing a unified benchmarking protocol. Conducting this benchmark already required over 437,000 experiments within a standardized pipeline, the paper has reached the 9-page limit with a 29-page appendix. Therefore, we leave designing new scalable attacks or defenses as future work, as they require an in-depth systematic analysis that we cannot provide for the above reasons.
>
> In response to this comment, we nonetheless provide a detailed analysis of each adversarial attack’s runtime by breaking down the algorithms. Our goal is to provide empirical data that can inform graph learning researchers.  We break attack algorithms into three main stages according to their implementations provided by DeepRobust[1]: Surrogate, Pre-attack and Attacks. Surrogate stage includes the time required to train and set up the surrogate models used in the attack. Pre-attack involves operations: computing logits of target nodes and normalizing the adjacency matrix in NETTACK and GOttack; performing project gradient descent training and projected randomized block coordinate descent in PGD and PR-BCD, respectively; retrieving subgraphs in SGA. Finally, in Attacks phases, attack algorithms find the best edges to flip according to the surrogate loss of all potential edges in NETTACK and selected potential edges in GOttack, while PR-BCD and PGD perform a Bernoulli sample to select the optimal edge to flip. We provide a time complexity breakdown of three main stages of adversarial attacks for an attack budget of 1 across three datasets (Cora, Pubmed, and OGBN-Arxiv) as follows:
> ####  Table: Cora
> | Method| Surrogate|Pre-attack| Attack|
> |-|-|-|-|
> |FGA| _0.73066_ | _0.00011_ |0.00531|
> |NETTACK| 1.23075 | 0.16365 | 0.20430 |
> |PGD (NA)| 0.98343 | 3.69364 | 0.04686 |
> |PRBCD (NA)| 1.43114 | 4.31843 | 0.06399 |
> |SGA| 1.62636 | 0.32616 | _0.00323_ |
> |GOttack| 1.16423 | 0.08035 | 0.07113 |
> |L1D-RND| **0.00000** | **0.00006** | **0.00213** |
> ####  Table: Pubmed
> | Method| Surrogate | Pre-attack | Attack |
> |-|-|-|-|
> | FGA| _1.86278_ | _0.00045_ | 0.27881 |
> | NETTACK| 4.45833 | 1.43834 | 6.53812 |
> | PGD (NA)| 2.75710 | 155.74790 | 2.74592 |
> | PRBCD (NA)| 2.57097 | 3.29684 | 0.06547 |
> | SGA| 10.47330 | 0.33145 | _0.00329_ |
> | GOttack| 1.91390 | 0.91947 | 2.03039 |
> | L1D-RND| **0.00000** | **0.00007** | **0.00387** |
>
> #### Table: OGBN-Arxiv
> | Method| Surrogate | Pre-attack | Attack |
> |-|-|-|-|
> | FGA| OOM | OOM | OOM |
> | NETTACK| 12.54029 | 362.42469 | 894.00637 |
> | PGD (NA)| OOM | OOM | OOM |
> | PRBCD (NA)   | 7.86793 | 12.93282 | 0.34241 |
> | SGA| _3.85193_ | _0.50283_ | _0.01772_ |
> | GOttack| 12.80845 | 326.28377 | 288.88327 |
> | L1D-RND| **0.00000** | **0.00010** | **0.13041** |
>
> *We omit standard deviation here. The completed results are shown in Table 11 of the revised manuscript.*
>
> Due to fast gradient-based updates, FGA spends significantly less time on Pre-attack and Attack stages on Cora and Pubmed datasets and achieves the second fastest on Cora and Pubmed. However, FGA needs to maintain a dense adjacency matrix to compute the gradient, which requires more than 100GiB on OGBN-Arxiv, causing Out-Of-Memory.
>
> On Cora, Pubmed and OGBN-Arxiv, SGA is one of the most efficient attacks, as it limits the pre-computation to small subgraphs, resulting in minimal in Pre-attack and Attack stages. In contrast, PGD's runtime on Cora and Pubmed is dominated by project gradient descent training in Pre-attack stage, accounting for 96% of total runtime on PUBMED. In addition, PGD encounters the same memory constraints as FGA on OGB-Arxiv. Instead of project gradient descent, PR-BCD utilizes projected randomized block coordinate descent, which has been shown to reduce the dominant effect of Pre-attack stage to 55% on Pubmed.
> Methods such as Nettack and GOttack are significantly slower, particularly on larger datasets. Pre-attack and Attack stages dominate the runtime in Nettack due to performing optimization to select the best edge to flip over all potential edges. GOttack's Pre-attack and Attack stages' runtime has shown improvements over Nettack due to reducing the search space by filtering potential edges. Its efficiency is due to its simple random edge selection strategy, which avoids surrogate training and iterative optimization entirely.
>
> We have revised and updated our manuscript to include this analysis in Appendix F.3 (highlighted in violet). We believe that the insights uncovered through our analysis shed light on important limitations of current methods and point to promising directions for developing scalable approaches in future work.
>
> [1] Li, Yaxin, et al. "Deeprobust: A pytorch library for adversarial attacks and defenses."

---

> ### Author Response · Authors · 2025-11-20
>
> ### W5. Writing issues in some sections
>
> > Writing issues in some sections. Certain parts of the experiments and tables are overly dense, making the presentation hard to follow; the narrative could be more concise and reader-friendly
>
>
> **Response:** Thank you very much for the review. We agree that the tables have grown larger and heavier as the number of results has increased. We believe that with our scope and hundreds of thousands of experiments, this could not be avoided. However, for this reason, we are implementing an adversarial graph learning benchmark website (modelled after the OGB style) to present all results more effectively. We will also think about alternative ways of organizing the tables to improve readability.

---

> ### Author Response · Authors · 2025-11-27
>
> Thank you once again for your insightful feedback and valuable time. If there are any further points you would like us to clarify, we would be more than happy to address them. If you feel that your comments and concerns have been addressed, we would kindly appreciate it if you could consider updating your scores.

---

### Official Review · Reviewer_voAP · 2025-10-31

**Soundness:** 3
**Presentation:** 2
**Contribution:** 2
**Rating:** 6
**Confidence:** 5

**Summary:**

The authors are interested in the subject of adversarial attacks for Graph Neural Networks. Specifically, the work tackles the challenge of fair comparison and evaluation of the current available GNNs and the proposed defense methods, by proposing a unified experimental setup, taking into account different factors.

**Strengths:**

- The tackled problem of GNN’s adversarial robustness is important to ensure better adoption of these methods.
- Some methods and evaluation consider different settings than the baselines, creating therefore some difference in terms of performance which is not due to the method itself but rather to hyper-parameters - unifying therefore the experimental setup is of great gain to the community.

**Weaknesses:**

- The authors considers the adversarial accuracy (or attacked accuracy) as the main point of comparison. While this is already a good element, I believe that in some specific settings, one should also take into account the complexity of the proposed defense method. In this perspective, I would suggest adding a Table of comparison in terms of complexity or training/evaluation time of each method. Such direction could give some credits to other methods that are rather aiming to have a simpler and yet effective approaches to defending adversarial robustness (for instance NoisyGNN [1])
- While a number of hyper-parameters and other elements are considered in the paper, I think important missing element to ensure the claimed and desired fairness is the initialization. As previously studied in [2], changing the initialization can have a great impact of the performance of a method. I was wondering if it’s possible to discuss such direction and how you ensured fairness in this aspect.
- The paper focuses on node classification with the topological attacks. While it’s already a big and important part of current research, I was wondering if there was a plan to extend to node feature-based adversarial attacks (in which different methods have been proposed [3])?

- [Minimal] The presentation format of the results could be enhanced. For instance through visualization. While I liked the table with all the results, it’s not very easy to compare all the methods and understand the ranking. By providing some visualizations (for instance a Radar-Chart plot), it could be easier to see the difference between defenses when subject to an attack for each datasets.

Typically, I would be very interested in seeing the discussion regarding the initialization and the time complexity in the revised manuscript for the rebuttal. I would also expect the authors to better cern their perspective on extending their framework and evaluation to the context of feature-based attacks.

—

[1] A Simple and Yet Fairly Effective Defense for Graph Neural Networks. - AAAI 2024.

[2] If You Want to Be Robust, Be Wary of Initialization. - NeurIPS 2024.

[3] Graph Neural Networks with Adaptive Residual. - NeurIPS 2021.

**Questions:**

- Could you provide clarification element regarding ensuring fairness when taking into account initialization?
- Can you provide benchmark regarding the performance in terms of complexity, to better understand the trade-off performance/complexity?
- Are there any plans to extend the study for node feature-based adversarial attacks?
- Could you also provide performance of the different methods in terms of certifications to better understand their theoretical performance?

---

> ### Author Response · Authors · 2025-11-20
>
> We thank the reviewer for taking the time to review our paper and for offering valuable suggestions. We are pleased that the reviewer describes our experimental setup as a significant contribution to the community. In the following, we will reply to the reviewer's questions and comments and revise our manuscripts to incorporate the reviewer’s feedback in green.
>
> ---
> ### W1 & Q2.  complexity of the defense methods
>
> > The authors consider adversarial accuracy (or attacked accuracy) as the main point of comparison. While this is already a good element, I believe that in some specific settings, one should also take into account the complexity of the proposed defense method. In this perspective, I would suggest adding a Table of comparison in terms of complexity or training/evaluation time of each method. Such direction could give some credits to other methods that are rather aiming to have a simpler and yet effective approaches to defending adversarial robustness (for instance NoisyGNN [1])
>
> > Can you provide benchmark regarding the performance in terms of complexity, to better understand the trade-off performance/complexity?
>
> **Response:** We appreciate the reviewer’s suggestion to incorporate the runtime analysis of defense models. In response to this comment/question, we provide a runtime comparison (in seconds) of defense on five datasets, Cora, Citeseer, Pubmed, Squirrel, and Chameleon in the table below:
> #### Table: Average total time to train defense models (in seconds). Smaller is better. Best time in **bold**, second best is *italic*.
> | **Defense**   | Cora            | Citeseer        | Pubmed           | Squirrel         | Chameleon        |
> |---------------|-------------------|--------------------|---------------------|---------------------|---------------------|
> | ElasticGNN    | 5.03 ± 2.78       | 3.92 ± 2.03        | 8.57 ± 5.27         | 10.67 ± 2.79        | 9.72 ± 5.05         |
> | GCN-GARNET    | 2.56 ± 2.51       | 3.87 ± 3.66        | 25.81 ± 9.39        | 3.45 ± 2.14         | 3.79 ± 1.56         |
> | GCN-Jaccard   | _1.24 ± 0.68_     | 1.58 ± 0.79        | -                   | -                   | -                   |
> | GNNGuard      | 1.91 ± 0.47       | 1.62 ± 0.59        | 4.39 ± 0.84         | 5.42 ± 0.63         | 2.37 ± 0.54         |
> | GRAND         | 6.46 ± 3.00       | 7.50 ± 2.39        | 93.48 ± 40.46       | 28.94 ± 23.13       | 5.42 ± 6.92         |
> | RobustGCN     | 3.95 ± 2.99       | **0.87 ± 0.12**    | _5.54 ± 1.67_       | _2.24 ± 0.56_       | **1.09 ± 0.22**     |
> | GCORN         | 54.99 ± 8.24      | 29.04 ± 7.89       | 4605.11 ± 727.34    | 130.53 ± 134.07     | 15.40 ± 5.34        |
> | RUNG          | 130.52 ± 5.34     | 106.62 ± 10.01     | 4265.07 ± 2279.45   | 46.06 ± 20.67       | 7.69 ± 2.78         |
> | NoisyGNN      | **0.77 ± 0.28**   | _1.29 ± 0.49_      | **2.13 ± 0.50**     | **1.78 ± 1.35**     | _1.70 ± 0.68_       |
>
> Overall, NoisyGNN consistently achieves the fastest training times on most datasets, with particularly low values on Cora, Pubmed, and Squirrel. Its efficiency can be attributed to its lightweight architecture and minimal computational overhead, which allows it to scale well even on larger graphs. The second-fastest model is generally RobustGCN, which performs competitively on Citeseer, Pubmed, and Chameleon, likely due to its optimized graph convolutional operations that reduce redundant computations. In contrast, RUNG is the slowest model by a wide margin, especially on large datasets such as Pubmed and Citeseer. This significant slowdown is perhaps because of its complex training procedure and use of robust aggregation mechanisms. Other models, such as GCORN and GRAND, also have high training times on large graphs.
>
> We revised our manuscript to include the discussion in Appendix F.4 (highlighted in green) and Table 12.
>
> In addition, thank you for bringing up NoisyGNN. We learned about NoisyGNN during the submission process and already began running the experiments. The results on homophily datasets are in Table 23 and on heterophily datasets are in Table 26. We also updated Tables 13 and 17 with NoisyGNN information.

---

> ### Author Response · Authors · 2025-11-20
>
> ### W2 & Q1. Important missing element to ensure the claimed and desired fairness is the initialization
>
> > While a number of hyper-parameters and other elements are considered in the paper, I think important missing element to ensure the claimed and desired fairness is the initialization. As previously studied in [2], changing the initialization can have a great impact of the performance of a method. I was wondering if it’s possible to discuss such direction and how you ensured fairness in this aspect.
>
> > Could you provide clarification element regarding ensuring fairness when taking into account initialization?
>
> **Response:** Thank you for raising this important point. We fully agree that the choice of model weight initialization can significantly influence robustness, as demonstrated in [4]. Incorporating initialization schemes into the hyper-parameter search would indeed be desirable for achieving stronger fairness evaluation, but given the already large search space, which required us to already run more than 437,000 experiments, we chose to reuse the initialization procedures recommended by the original paper. We made this choice for two reasons: 1)In practice, most end users follow the parameter initialization procedures recommended by the original paper.2) Including a search over alternative initialization methods would have required months of computation, which was not feasible for our study.
>
> While hyper-parameter tuning on initialization methods is beyond the feasible scope of this benchmark release, we have taken steps to acknowledge and mitigate this limitation as we detail below.
>
> First, we revised our manuscript to explicitly acknowledge this limitation in Appendix B, adding the following discussion:
>
> *While our benchmark explores a comprehensive range of hyper-parameters, we acknowledge that model weight initialization can also influence robustness outcomes. Ennadir et al. (2024b) highlights that different initializations may lead to noticeably different robustness levels. Due to the computational cost, over 437,000 additional experiments, our current benchmark utilized the initialization used by the original papers and does not perform hyper-parameter tuning on initialization. We therefore report the specific initialization scheme used for each model, leaving a full exploration of initialization robustness for future work.*
>
>
> Second, we revised Table 17 to include the initialization scheme used for every model, shown below:
>
> | Models                                                                 | Initialization Scheme      |
> |-------------------------------------------------------------------------|-----------------------------|
> | GCN, GCN_surrogate, SGC, RobustGCN, GNNGuard, NoisyGNN                 | Glorot (Xavier) Initialization[1] |
> | GIN, GSAGE, PNA, ElasticGNN, GRAND, RUNG                               | Kaiming (He)[2] Initialization    |
> | GCORN                                                                  | Orthogonal Initialization [3]     |
>
> Finally, we updated Section C to include a new discussion on the importance of initialization and its potential impact on robustness:
>
> *Initialization plays a non-trivial role in determining the stability and robustness of GNN models. Consistent with findings in recent literature, different initializations may lead to different robustness levels, even under identical training settings. While our benchmark uses commonly adopted default initializers for each architecture, we recognize that further investigation into initialization sensitivity may reveal additional insights into model and defense behavior.*
>
>
> [1] Glorot, Xavier, and Yoshua Bengio. "Understanding the difficulty of training deep feedforward neural networks." Proceedings of the thirteenth international conference on artificial intelligence and statistics. JMLR Workshop and Conference Proceedings, 2010.
>
> [2] He, Kaiming, et al. "Delving deep into rectifiers: Surpassing human-level performance on imagenet classification." Proceedings of the IEEE international conference on computer vision. 2015
>
> [3] Saxe, Andrew M., James L. McClelland, and Surya Ganguli. "Exact solutions to the nonlinear dynamics of learning in deep linear neural networks." arXiv preprint arXiv:1312.6120 (2013).
>
> [4] Ennadir, Sofiane, et al. "If you want to be robust, be wary of initialization." Advances in Neural Information Processing Systems 37 (2024): 23796-23823.

---

> ### Author Response · Authors · 2025-11-20
>
> ### W3 & Q3.  Feature-based attack
>
> > The paper focuses on node classification with the topological attacks. While it’s already a big and important part of current research, I was wondering if there was a plan to extend to node feature-based adversarial attacks (in which different methods have been proposed [3])?
>
> > Are there any plans to extend the study for node feature-based adversarial attacks?
>
> **Response:** We appreciate the reviewer’s suggestion regarding extending the benchmark to node features–based adversarial attacks. In this work, focusing solely on structural attacks already required over 437,000 experiments within a unified and standardized pipeline, and the paper has reached the 9-page limit with a 29-page appendix. Nevertheless, we fully agree that benchmarking feature-based attacks is an important future direction.
>
> **To demonstrate feasibility, we have already extended our experimental setup and conducted feature-attack variants of FGA, NETTACK, PGD, and SGA on Cora, Citeseer, and Pubmed**(PGD on Pubmed is currently in progress). These results, now shown in Table 32, confirm that our framework naturally generalizes to feature perturbations.
>
> We have revised Appendix Section B to acknowledge the limitation and highlighted this as a future work direction as follows:
>
> *Although this work focuses on topological attacks, our experimental pipeline and open-source code have been extended to support feature-based attacks. We therefore conduct feature-attack variants of FGA, NETTACK, PGD, PR-BCD, and SGA on GCN across homophily datasets. The results show that NETTACK and FGA consistently yield the highest misclassification rates in both evasion and poisoning settings, while SGA and PGD perform substantially worse on PUBMED. We also observe the same trend as in structural attacks: high-degree nodes remain significantly more robust (11.92% average success rate) than low-degree nodes (94.85%). Despite these results, a full benchmark of feature-based attacks is beyond the scope of the current paper, and we acknowledge this as an important direction and leave it as future work, building upon our setting.*
>
> ---
> ### W4. Visualization
>
> > [Minimal] The presentation format of the results could be enhanced. For instance through visualization. While I liked the table with all the results, it’s not very easy to compare all the methods and understand the ranking. By providing some visualizations (for instance a Radar-Chart plot), it could be easier to see the difference between defenses when subject to an attack for each dataset.
>
>
> **Response:**
> Thank you very much for this great suggestion.  We have now generated radar plots of the misclassification rates for all experimental settings. Misclassification rates of defense models against each adversarial attack under budget 1 on each dataset are shown as radar plots in Figures 5 and 6. In addition, we add radar plots to show the average performance of each adversarial attack across vanilla and defended classifiers in Figures 3 and 4.
>
> ---
> ### Q1. Fairness in initialization
>
> **Response:** Please refer to our answer to W2.
>
> ---
> ### Q2. Performance in terms of complexity
>
> **Response:** Please refer to our answer to W1.
>
> ---
> ### Q3. Feature-based adversarial attack
>
> **Response:** Please refer to our answer to W3.
>
> ---
> ### Q4.  Theoretical performance
>
> > Could you also provide performance of the different methods in terms of certifications to better understand their theoretical performance?
>
> **Response:** We thank the reviewer for raising the issue. We agree that certification provides an important theoretical perspective on robustness. However, most existing defense methods in our benchmark do not provide certified robustness modules or code, and many lack the guarantees or assumptions required to compute certificates in a comparable manner. As a result, it is not feasible to produce fair, standardized certification results across all methods within our current framework.
>
> We therefore acknowledge certification as an important but orthogonal direction, and we consider a systematic benchmark of certified defenses to be outside the scope of this work. We have added this point to Appendix B of the manuscript as follows:
>
> *Our benchmark does not include an evaluation of certified robustness. While certification offers valuable theoretical guarantees, most defense methods in our study do not provide certification modules or compatible implementations, making it infeasible to run a fair and standardized certification comparison. We therefore consider certification an important but orthogonal direction, and we leave a systematic benchmark of certified defenses to future work.*

---

> > ### Author Response · Authors · 2025-11-25
> > **PGD feature-based attack on PUBMED results are now ready**
> >
> > Thank you once again for your insightful feedback and valuable time. Regarding W3 and Q3 about feature-based attacks, the experiments of PGD on PUBMED are now ready, and we have updated Table 32 in the Appendix with the completed results. If there are any further points you would like us to clarify, we would be more than happy to address them.

---

> > > ### Comment · Reviewer_voAP · 2025-11-26
> > >
> > > Sorry for the small confusion in where I posted my original feedback regarding the rebuttal, as I was reading the other reviewer's comment.
> > >
> > > I would like to note that I deeply appreciate the author's rebuttal and provided elements, I am happy with all the answers and see that the manuscript is very interesting and could provide some additional value to the research community focusing on Adversarial robustness.
> > >
> > > I therefore keep my positive score, and keep monitoring the other reviewer's feedback. Thank you again for taking all my request and answering my question during the rebuttal period.

---

> > > > ### Author Response · Authors · 2025-11-26
> > > >
> > > > Thank you very much. We appreciate your comments, which provide valuable guidance for refining our work, and we will incorporate the related changes into the article.

---

### Official Review · Reviewer_w8mN · 2025-11-04

**Soundness:** 3
**Presentation:** 3
**Contribution:** 3
**Rating:** 6
**Confidence:** 4

**Summary:**

This paper presents a comprehensive and rigorous benchmark for evaluating adversarial attacks and defenses on Graph Neural Networks (GNNs). The authors identify critical issues in the current literature, such as inconsistent evaluation protocols and biased experimental setups, which lead to unreliable or incomparable conclusions. To address this, they propose a standardized framework that incorporates realistic model selection, diverse target node sampling strategies, and evaluation across multiple random data splits.
Within this framework, the authors conduct a large-scale re-evaluation of seven prominent attack methods and eight defense mechanisms across six diverse graph datasets (including homophilic, heterophilic, and large-scale graphs) under both poisoning and evasion settings.

**Strengths:**

1.	The paper’s decision to focus on creating a fair and standardized benchmark directly addresses a fundamental issue hindering progress in adversarial GNN research. This methodological contribution is highly valuable, as it provides the community with a tool to filter out unreliable results and foster more reproducible science.
2.	The experimental framework itself is a key strength. Its design is exceptionally rigorous, incorporating crucial elements like proper model selection and diverse target node sampling that are often overlooked. This approach ensures that the comparisons are not only fair but also reflect more realistic application scenarios.
3.	The discovery that target node selection and victim model tuning can drastically alter outcomes is a major finding. The paper provides compelling evidence that many prior works may have systematically overestimated attack effectiveness. This insight is powerful and forces a necessary re-evaluation of the community's collective understanding of GNN fragility.

**Weaknesses:**

1.	In the paper, it seems that the evaluation of adaptive attacks is somewhat limited. While including an adaptive variant of one attack is a good step, a truly robust defense should be tested against attackers that are specifically designed to circumvent its core mechanism. It may help that the authors could discuss the scope of their adaptive evaluation and acknowledge that stronger, tailored attacks might exist for some of the tested defenses.
2.	In the experiments, the analysis on the large-scale OGB-ARXIV dataset seems to be constrained by computational resources. This is understandable, but it does mean that conclusions about scalability are based on a limited set of successful runs. At this time, this might temper the strength of these specific claims. I would encourage the authors to be more explicit about this limitation in the main text.
3.	The paper reports that many attacks failed on the large graph due to timeouts, which is an important finding. However, the reasons for these failures are not deeply analyzed. It would be more insightful to understand if the bottleneck is memory, a specific computational step, or another factor. A more detailed breakdown of these failure modes would provide valuable lessons for designing future scalable algorithms.

**Questions:**

1.	I am curious about your findings on adaptive attacks[1]. Did you observe if making an attack "adaptive" (i.e., aware of the defense) had any impact on its general attack strength? For instance, does an attack become less effective overall when it is modified to bypass a specific defense? A brief comment on this potential trade-off would be very interesting.
2.	The findings on the OGB-ARXIV dataset are very insightful. Beyond reporting the timeouts, do you have any further analysis on the specific bottlenecks that prevent methods from scaling? Understanding whether the primary limitation is memory usage or computational time for specific operations would be very helpful for future algorithm design.
3.	The results clearly show that high-degree nodes are inherently more robust to the attacks you tested. What implications does this have for the design of future defense mechanisms? Does it suggest that defenses should focus their efforts on protecting low-degree or fringe nodes, rather than applying a uniform protection strategy across the entire graph?

[1]Felix Mujkanovic, Simon Geisler, Stephan Günnemann, and Aleksandar Bojchevski. Are defenses for graph neural networks robust? Advances in Neural Information Processing Systems 35 (NeurIPS 2022), 2022.

---

> ### Author Response · Authors · 2025-11-20
>
> We thank the reviewer for taking the time to review our paper and for offering valuable suggestions. We are pleased that the reviewer recognizes the importance of our work in addressing a fundamental issue hindering progress in adversarial GNN research. Here, we address each point raised by the reviewer as follows and revise our manuscript to incorporate the reviewer’s feedback in violet in the updated PDF.
>
> ---
> ### W1.  The scope of their adaptive evaluation
> > In the paper, it seems that the evaluation of adaptive attacks is somewhat limited. While including an adaptive variant of one attack is a good step, a truly robust defense should be tested against attackers that are specifically designed to circumvent its core mechanism. It may help that the authors could discuss the scope of their adaptive evaluation and acknowledge that stronger, tailored attacks might exist for some of the tested defenses.
>
> **Response:** We thank the reviewer for this valuable observation. We agree that evaluating defenses against attacks specifically crafted to circumvent their core mechanisms is important for assessing true robustness [1]. In this work, we included an adaptive variant of PR-BCD, with results reported in Tables 27/28. An intriguing result from these tables is that the adaptive version is not always better than the non-adaptive version in both homophilic and heterophilic settings.
> Nonetheless, we agree that stronger, defense-tailored adaptive attacks may provide a more reliable lower bound on the robustness of defenses. To address this, we have updated Section 4.1 (highlighted in violet) to clarify the scope of our adaptive evaluation and acknowledge the importance of evaluating defenses against adaptive attacks.
>
> [1] Felix Mujkanovic, Simon Geisler, Stephan Günnemann, and Aleksandar Bojchevski. Are defenses for graph neural networks robust? Advances in Neural Information Processing Systems 35 (NeurIPS 2022), 2022.
>
> ---
> ### W2.  More explicit about this limitation in the main text.
>
> > In the experiments, the analysis on the large-scale OGB-ARXIV dataset seems to be constrained by computational resources. This is understandable, but it does mean that conclusions about scalability are based on a limited set of successful runs. At this time, this might temper the strength of these specific claims. I would encourage the authors to be more explicit about this limitation in the main text
>
> **Response:**
> Thank you for raising this. We would like to clarify that the limitation was not due to insufficient computational resources. Our experiments were conducted on a compute cluster equipped with multiple NVIDIA RTX A40 GPUs, each with 48 GB of ECC GDDR6 memory, 10,752 CUDA cores, 336 third-generation Tensor cores, and 84 second-generation RT cores. These GPUs are optimized for large-scale training and inference, with PCIe Gen4 support and over 690 GB/s of memory bandwidth, making them well-suited for high-throughput graph neural network workloads.
>
> As noted in our paper, the core limitation lies in the scalability of current adversarial attack algorithms, not in hardware. Most existing attacks could not process OGB-ARXIV due to impractically long runtimes, excessive memory consumption, or failure to converge. In fact, many approaches struggled even with mid-sized graphs. For example, running Nettack on the 19K nodes in Pubmed dataset already takes 1753 seconds.
>
> We included OGB-Arxiv to illustrate this gap. The inability of these methods to scale to even one million-node graphs raises concerns about their real-world practicality. Thus, the partial results on OGB-Arxiv are not a shortcoming of our study, but rather a key finding that motivates the need for more scalable adversarial techniques in graph learning.

---

> > ### Author Response · Authors · 2025-11-27
> >
> > Thank you once again for your valuable time and insightful feedback, which provides valuable guidance for refining our work. If there are any further points you would like us to clarify, we would be more than happy to address them.

---

> ### Author Response · Authors · 2025-11-20
>
> ---
> ### W3 & Q2.  Deep analysis of attack failures on a large graph.
>
> > The paper reports that many attacks failed on the large graph due to timeouts, which is an important finding. However, the reasons for these failures are not deeply analyzed... A more detailed breakdown of these failure modes would provide valuable lessons for designing future scalable algorithms.
>
> > The findings on the OGB-Arxiv dataset are very insightful. Beyond reporting the timeouts, do you have any further analysis on the specific bottlenecks that prevent methods from scaling? ....
>
>
> **Response:** We thank the reviewer for the thoughtful suggestion. To better understand the computational cost of different adversarial attack methods, we break attack algorithms into three main stages according to their implementations provided by DeepRobust[1]: Surrogate, Pre-attack and Attacks. Surrogate stage includes the time required to train and set up the surrogate models used in the attack. Pre-attack involves operations: computing logits of target nodes and normalizing the adjacency matrix in NETTACK and GOttack; performing project gradient descent training and projected randomized block coordinate descent in PGD and PR-BCD, respectively; retrieving subgraphs in SGA. Finally, in Attacks phases, attack algorithms find the best edges to flip according to the surrogate loss of all potential edges in NETTACK and selected potential edges in GOttack, while PR-BCD and PGD perform a Bernoulli sample to select the optimal edge to flip. We provide a time complexity breakdown of three main stages of adversarial attacks for an attack budget of 1 across three datasets (Cora, Pubmed, and OGBN-Arxiv) as follows:
>
>
> ####  Table: Cora
> | Method        | Surrogate | Pre-attack | Attack |
> |--------------|-----------|------------|--------|
> | FGA          | _0.73066_ | _0.00011_ | 0.00531 |
> | NETTACK      | 1.23075 | 0.16365 | 0.20430 |
> | PGD (NA)     | 0.98343 | 3.69364 | 0.04686 |
> | PRBCD (NA)   | 1.43114 | 4.31843 | 0.06399 |
> | SGA          | 1.62636 | 0.32616 | _0.00323_ |
> | GOttack      | 1.16423 | 0.08035 | 0.07113 |
> | L1D-RND   | **0.00000** | **0.00006** | **0.00213** |
>
>
> ####  Table: Pubmed
> | Method        | Surrogate | Pre-attack | Attack |
> |--------------|-----------|------------|--------|
> | FGA          | _1.86278_ | _0.00045_ | 0.27881 |
> | NETTACK      | 4.45833 | 1.43834 | 6.53812 |
> | PGD (NA)     | 2.75710 | 155.74790 | 2.74592 |
> | PRBCD (NA)   | 2.57097 | 3.29684 | 0.06547 |
> | SGA          | 10.47330 | 0.33145 | _0.00329_ |
> | GOttack      | 1.91390 | 0.91947 | 2.03039 |
> | L1D-RND   | **0.00000** | **0.00007** | **0.00387** |
>
> #### Table: OGBN-Arxiv
> | Method        | Surrogate | Pre-attack | Attack |
> |--------------|-----------|------------|--------|
> | FGA          | OOM | OOM | OOM |
> | NETTACK      | 12.54029 | 362.42469 | 894.00637 |
> | PGD (NA)     | OOM | OOM | OOM |
> | PRBCD (NA)   | 7.86793 | 12.93282 | 0.34241 |
> | SGA          | _3.85193_ | _0.50283_ | _0.01772_ |
> | GOttack      | 12.80845 | 326.28377 | 288.88327 |
> | L1D-RND   | **0.00000** | **0.00010** | **0.13041** |
>
> *We omit standard deviation here. The completed results are shown in Table 11 of the revised manuscript.*
>
> Due to fast gradient-based updates, FGA spends significantly less time on Pre-attack and Attack stages on Cora and Pubmed datasets and achieves the second fastest on Cora and Pubmed. However, FGA needs to maintain a dense adjacency matrix to compute the gradient, which requires more than 100GiB on OGBN-Arxiv, causing Out-Of-Memory.
>
> On Cora, Pubmed and OGBN-Arxiv, SGA is one of the most efficient attacks, as it limits the pre-computation to small subgraphs, resulting in minimal in Pre-attack and Attack stages. In contrast, PGD's runtime on Cora and Pubmed is dominated by project gradient descent training in Pre-attack stage, accounting for 96% of total runtime on PUBMED. In addition, PGD encounters the same memory constraints as FGA on OGB-Arxiv. Instead of project gradient descent, PR-BCD utilizes projected randomized block coordinate descent, which has been shown to reduce the dominant effect of Pre-attack stage to 55% on Pubmed.
> Methods such as Nettack and GOttack are significantly slower, particularly on larger datasets. Pre-attack and Attack stages dominate the runtime in Nettack due to performing optimization to select the best edge to flip over all potential edges. GOttack's Pre-attack and Attack stages' runtime has shown improvements over Nettack due to reducing the search space by filtering potential edges. Across all datasets, L1D-RND is the fastest method and achieves close to zero seconds in all three stages. Its efficiency is due to its simple random edge selection strategy, which avoids surrogate training and iterative optimization entirely.
>
> We have revised and updated our manuscript to include this analysis in Appendix F.3 (highlighted in violet)
>
> [1] Li, Yaxin, et al. "Deeprobust: A pytorch library for adversarial attacks and defenses."

---

> ### Author Response · Authors · 2025-11-20
>
> ### Q1.  Findings on adaptive attacks
>
> > I am curious about your findings on adaptive attacks[1]. Did you observe if making an attack "adaptive" (i.e., aware of the defense) had any impact on its general attack strength? For instance, does an attack become less effective overall when it is modified to bypass a specific defense? A brief comment on this potential trade-off would be very interesting
>
> **Response:**
> Thank you for the review. We had similar questions and had prepared tables 27 and 28 for that purpose. Adaptive attacks are theoretically stronger; however, the tables show that PR-BCD (NA, the non-adaptive version) consistently outperforms PR-BCD on homophilic datasets at higher attack budgets. On heterophilic datasets, the adaptive’s performance is mixed and often worse. Hence, the expectation that adaptive will perform better is not consistently met, especially on the homophilic datasets. Another aspect is that, depending on the victim model, the adaptive versions may be much more costly than the non-adaptive ones. Therefore,  the added cost may not always be justified.
>
> ---
> ### Q2. Findings on the OGB-ARXIV
>
> **Response:** Please refer to our answer to W3.
>
> ---
> ### Q3.  Implications does this have for the design of future defense mechanisms?
>
> > The results clearly show that high-degree nodes are inherently more robust to the attacks you tested. What implications does this have for the design of future defense mechanisms? Does it suggest that defenses should focus their efforts on protecting low-degree or fringe nodes, rather than applying a uniform protection strategy across the entire graph?
>
> **Response:** We thank the reviewer for this insightful observation. We agree that our results reveal a clear trend: low-degree nodes tend to be substantially more vulnerable, whereas high-degree nodes exhibit stronger inherent robustness under adversarial attacks. This suggests that future defense mechanisms may benefit from prioritizing protection for low-degree nodes rather than applying a uniform strategy across the entire graph. In addition, the importance of low-degree nodes varies across application domains, critical in some settings but less so in others. Consequently, defenses may choose either to focus protection on low-degree nodes or to maintain uniform protection across all nodes, depending on the domain.
>
> We have incorporated this discussion into our conclusion in Section 6.2 (highlighted in violet). We hope that by bringing this empirical pattern to light, our benchmark can help guide the community toward more targeted and context-aware defense strategies.

---

### Author Response · Authors · 2025-11-30

Dear chairs,

We sincerely appreciate the time and effort you are investing in reviewing our submission, especially given these challenging circumstances. To support your evaluation, we have included below a brief overview of the main changes made during the rebuttal period.

We believe we have addressed all reviewer comments and concerns by adding additional experimental analyses and providing further clarifications.

- - -
1. Reviewer **w8mN** (rating 6) asked for the performance of adaptive attacks, limitations about computational resources, analysis of why attacks fail on a large graph, and the implications of our results for the design of future defenses.

>We provided adaptive versions of attacks when available (e.g., PR-BCD), and the adaptive attacks do not consistently outperform their non-adaptive counterparts. The computational limitations are not due to a lack of resources but stem from the attack complexity. We analyzed the time complexity breakdown to identify bottlenecks that prevent attacks from scaling. We also explained that future defenses could benefit from defending low-degree nodes.


- - -
2. Reviewer **voAP** (rating 6) asked us to include the time complexity of the defense methods. The reviewer also mentioned the importance of weight initialization in the models’ robustness and asked for feature-based attacks (in addition to our structural attacks) and certification for defenses.

> We added the time complexity and initialization schemes of the models. We also added results on feature attacks. A concrete improvement is that we have added radar plots of attacks/defenses as the reviewer suggested.

- - -
3. Reviewer **GT8F** (rating 4) asked for an analysis on why some defenses fail and asked us to reduce the discussion of the naive L¹D-RND baseline. In particular, the reviewer mentioned that our work does not contribute a new algorithm.

> We provided a new table of defense families with their performances and analyzed why some defenses fail. We also pointed out that the new algorithm comment may not apply to us because our "primary submission area" is the "Datasets and Benchmarks", and we believe that our paper’s main contributions (standardized evaluation methodology, large-scale experimental design, and comprehensive benchmarking) provide a strong contribution to the field.

- - -
4. Reviewer **byfG** (rating 6) raised questions on the effectiveness of L¹D-RND and further analysis on Nettack and FGA to understand why these models have diverging performances on homophilic and heterophilic datasets.

> We responded with a new ablation study on the L¹D-RND by randomly switching off certain parts of the algorithm. This analysis showed that connections to high-degree nodes have an important contribution to the attack performance. We also provided our hypothesis on why Nettack and FGA have diverging performances on homophilic and heterophilic datasets

---

We note that the reviewers have not raised any issues that endanger the validity of our experiments. The ratings have been quite positive, and we welcome any new questions from the AC, if necessary.

---

### Meta-Review · Area_Chair_K53S · 2026-01-07

**Summary:**

This paper presents a well-motivated benchmark for evaluating adversarial attacks and defenses on GNNs. Reviewers appreciated the scale of the experimental effort, the standardized evaluation protocol, and the insightful findings on target node selection, model tuning, and the surprising effectiveness of simple baseline attacks. However, the contribution is somewhat incremental, given the limited novelty beyond benchmarking and the existence of prior benchmarks on graph learning robustness. In addition, the analysis of adaptive attacks, defense mechanisms, and scalability on large graphs remains limited.

**Reviewer Concerns:**

Reviewers have concerns about the limited novelty beyond benchmarking. In addition, the analysis of adaptive attacks, defense mechanisms, and scalability on large graphs remains limited.

**Reviewer Scores:**

Reviewers keep their marginal scores after rebuttal.

---

### Decision · Program_Chairs · 2026-01-26

Reject